# Habitat openness and squamate color evolution over deep time

Jonathan Goldenberg [1,2] ✉, Karen Bisschop [3,4], Joshua W. Lambert [5], Michaël P. J. Nicolaï [2], Rampal S. Etienne [5], Liliana D'Alba [2,6] & Matthew D. Shawkey [2]

While the ecological roles of colored integument have been extensively studied, what regulates global patterns of color variation remains poorly understood. Here, using a global dataset of 1249 squamates, we evaluate whether and how six key eco-environmental variables and their interactions shaped the evolutionary history of their coloration. We show that only habitat openness consistently associates with brightness evolution, with brighter integuments favored in open habitats, possibly for enhanced heat reflection. Furthermore, brightness evolution rates likely track $\delta^{18}O$ (a temperature proxy) changes and increase during global aridification phases, such as those in the Miocene and Pliocene. This trend may be due to the establishment of an arid climate that promoted habitat openness shifts, ultimately inducing adaption to new niches. Our findings suggest that a single environmental variable is associated with color variation in the largest extant tetrapod order.

A central aim of ecology and evolution is to understand how species respond to rapid environmental changes. This has led many studies to evaluate how present conditions influence the ecology and evolution of species[1–3]. Yet, analyzing only current conditions may mask long-term evolutionary drivers, as the Earth's climate has undergone major climatic shifts due to new orogenic and ocean current formations and cataclysmic events that promoted species turnover[4] and mass extinctions[5], respectively. Fortunately, recent advances in bioinformatics and paleontological discoveries have enabled the development of tools to deduce ancient climates and reconstruct species' ecologies[6–8]. A changing climate has a critical impact on the biotic environment because it induces animals to shift their distribution or to adapt[9]. Two ways species may cope with climate change are through phenotypic plasticity and genetic adaptation[10]. However, the former likely acts faster than the latter allowing for evolutionary rescue by 'buying time' for adaptation to occur[11]. Thus, phenotypic evolutionary reconstructions can provide insights into how species may respond to future climate scenarios[12].

Among organismal phenotypes, coloration is one of the most visible, and its ecological function has been extensively studied, particularly in terms of concealment, mimicry, social signaling, and sexual selection[13]. While thermoregulation has received less attention, it may become even more relevant due to anthropogenic climate change[14,15]. Notably, in the context of thermoregulation, brightness rather than hue emerges as a crucial metric because it directly affects heat absorption[16]. Coloration itself can be influenced by multiple abiotic factors, including Ultraviolet (UV) exposure driven by latitude and altitude distributions[17], humidity levels[18], and temperature fluctuations[16], as well as biotic factors such as body size[19], circadian rhythm[20], intraspecific competition[21], and vegetation cover[22]. For instance, darker species are expected to be favored at higher latitudes to maximize heat gain in colder environments[16]. Surprisingly, despite this extensive body of research, the mechanisms driving global patterns of color variation are fervently debated, with contrasting results persisting even within closely related taxa[14].

Squamates are one such group for which the function and evolution of the colored integument has been extensively studied.

[1]Division of Biodiversity and Evolution, Department of Biology, Lund University, Lund, Sweden. [2]Evolution and Optics of Nanostructures group, Department of Biology, Ghent University, Ghent, Belgium. [3]Terrestrial Ecology Unit, Department of Biology, Ghent University, Ghent, Belgium. [4]Laboratory of Aquatic Biology, KU Leuven Kulak, Kortrijk, Belgium. [5]Groningen Institute for Evolutionary Life Sciences, University of Groningen, Groningen, The Netherlands. [6]Naturalis Biodiversity Center, Leiden, The Netherlands. ✉e-mail: jonathan.goldenberg@ugent.be

Molecular data suggest squamates originated at the interface between the Triassic and Jurassic ~ 202 Ma[23] - but a recent fossil discovery from the late Triassic of England pushes the crown age of Squamata to 232 Ma[24]. They radiated globally in the late Cretaceous ~ 80 Ma[23], and with over 11500 species (The Reptile Database[25], http://www.reptile-database.org, accessed January 12, 2023) it is the largest order of extant vertebrates and exhibits an exceptional ecological diversity[26]. Here, to gain an understanding of the mechanisms driving their color brightness evolution and to specifically evaluate whether selected drivers are consistent across taxa, despite the opposing predictions found in the literature[14], we compiled a global dataset of squamates' life-history traits and ecological information known to influence brightness[19,27]. This dataset includes habitat openness, latitudinal and altitudinal distributions (which correlate with environmental factors such as temperature, photoperiod, and UV exposure), body mass, concealment, polymorphism, and circadian rhythm. It covers 1372 aquatic and terrestrial species from 25 families representing major squamate groups. Despite covering only 12% of the species in the order, the database has a wide enough sampling to capture different ecological strategies, and tempo and mode of evolution.

We show that only habitat openness is associated with brightness evolution, specifically that brighter integuments are favored in open habitats, possibly to better reflect and thereby reduce the incoming light. Family-level analyses indicate that body mass and altitudinal distribution also influence brightness but only in groups with relatively small body sizes, suggesting stronger evolutionary pressures in small species. Other family-specific strengths of selections are latitude and interactions between some of the abovementioned factors, likely

reflecting the life history strategies of species. Moreover, brightness evolution rates likely track foraminiferal $\delta^{18}O$ fluctuations, a measure of fractionation of $^{18}O$ to $^{16}O$ and an indicator of temperature change over time[28], presumably due to the establishment of cooler and drier climates which promoted landscape shifts[29], ultimately inducing species to adapt to new habitats. Because $\delta^{18}O$ fluctuations are indicators of past temperature variations, their correlation with brightness evolution rates suggests that shifts in habitat openness, driven by temperature changes[28], played a significant role in shaping the evolution of dorsal brightness. Thus, while habitat openness itself does not directly select for brightness, it promotes environmental conditions that favor brighter integuments, possibly through the physiological stress and predation risks in open environments. This finding underscores the utility of foraminiferal $\delta^{18}O$ as a proxy for habitat openness shifts and its relevance for understanding evolutionary dynamics in response to environmental change over broad temporal and spatial scales.

## Results

### Identifying patterns of color variation and evolution

To visualize how coloration is distributed across the globe, we initially plotted dorsal brightness variation for 1251 species for which distribution data are available[30] (Fig. 1). We found that brighter species are generally found in open habitats such as grasslands, deserts, and prairielands, and darker species in forested regions. In contrast to our expectations that darker species are favored at higher latitudes to maximize heat gain, we saw no substantial changes in brightness across a wide latitudinal gradient.

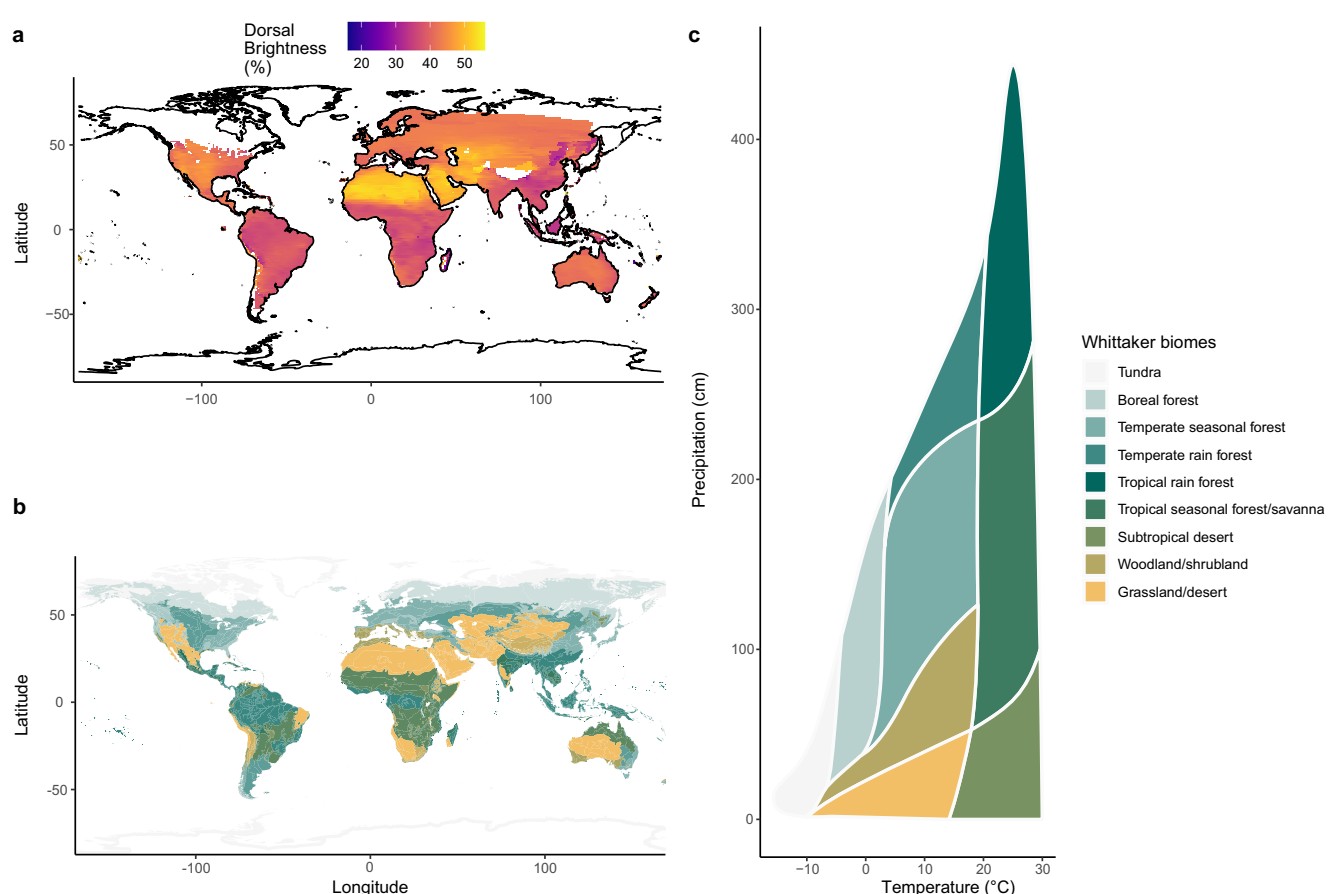

**Fig. 1 | Squamate dorsal brightness association with biomes. a** Global variation of dorsal brightness in squamates shows regional clustering, where open areas such as deserts, grasslands, and steppes host brighter species than closed environments such as forests (**b**). Darker tonalities in (**a**) indicate regional areas with darker species, whereas brighter ones indicate brighter species assemblages. See methods for brightness classification. Tonalities in (**b**) follow the Whittaker's biomes classification (**c**). Species' distribution from Roll et al.[30] and maps produced with the R packages ggplot[77] and plotbiomes[93].

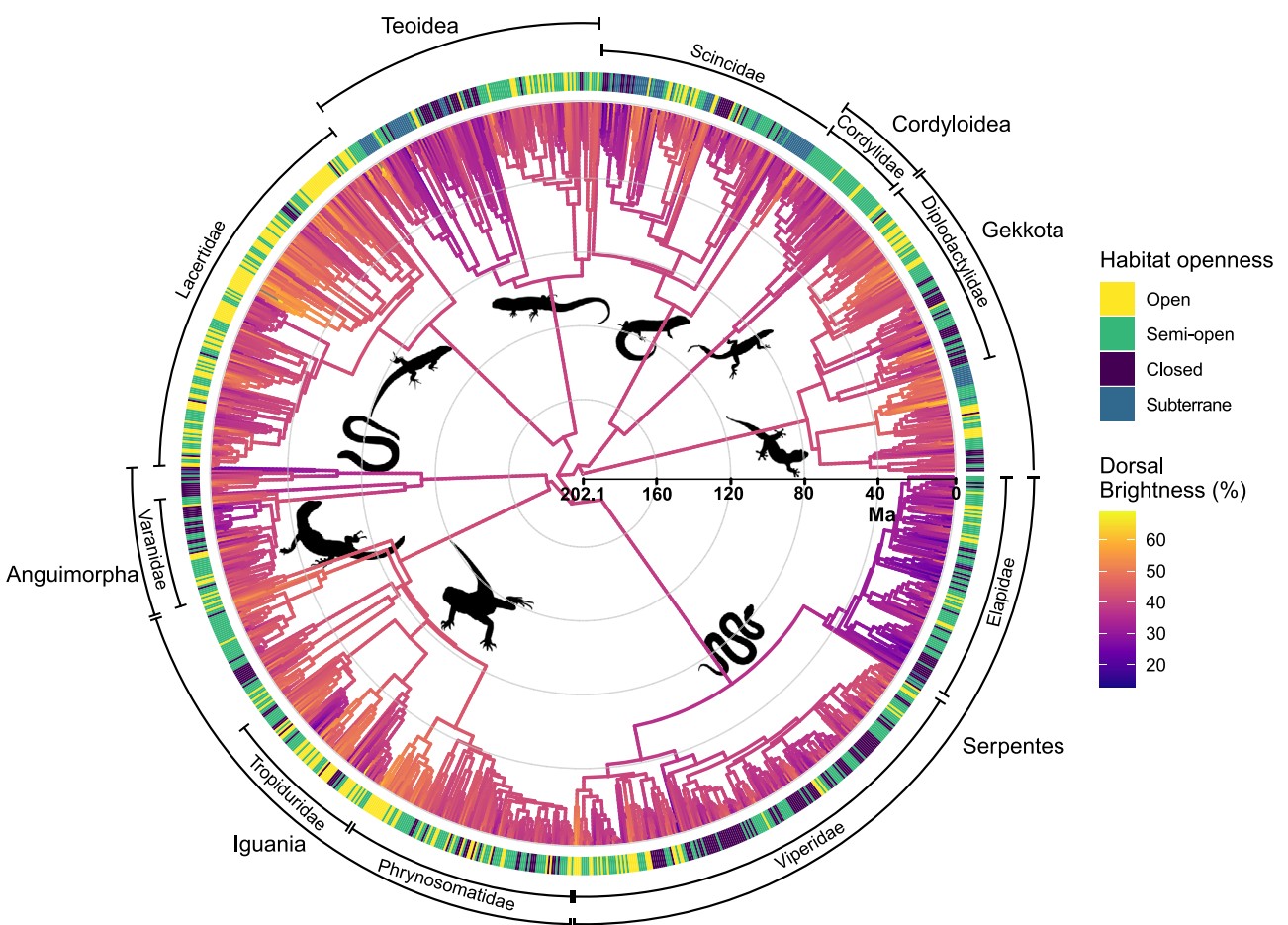

**Fig. 2 | Evolutionary reconstruction of color brightness.** The ancestral state estimation of squamates brightness shows phylogenetic clustering and associations between bright species and open habitats and dark species and closed habitats under an OU model of evolution (best-supported model – see text). Branch colors reflect the relative brightness of species. Light exposure, driven by habitat openness, is categorized according to the canopy density or behavioral habits (in the case of a fossorial, subterranean lifestyle), which regulate the amount of light reaching species within their described environment. See methods for a full description. Visualization and annotation of tree performed with the R package ggtree[81]. Silhouettes from phylopic.org.

We then reconstructed the ancestral state estimates of color brightness of squamates under an Ornstein-Uhlenbeck (OU) model (AIC: Brownian motion (BM) = 8756.22; OU = 8424.152; Early Burst (EB) = 8435.44), and, because brightness clusters with different biomes across the globe (Fig. 1), we also visualized how habitat openness is distributed across the phylogeny (Fig. 2). We found clustering of both brightness evolution and habitat openness, with an overall strong phylogenetic signal for brightness at squamate (Pagel's lambda ($\lambda$) = 0.75, $p << 0.01$) and at clade (e.g., Elapidae: $\lambda = 0.48$, $p = 0.01$; Lacertidae: $\lambda = 0.82$, $p << 0.01$; Varanidae: $\lambda = 0.58$, $p < 0.01$) levels. The estimated brightness of the squamate root was relatively high at 40% (fixed optimum) and following major diversification events in the late Cretaceous - early Paleogene (70–40 Ma) groups increased (e.g., Iguanidae 48% [33,64], Gekkota 44 % [29,58]) and decreased (e.g., Elapidae 29% [20,37], Teoidea 36% [20,53]) their brightness levels.

**Ecological and environmental variables associated with brightness**
Next, we evaluated the relationship between ecological and environmental variables and the brightness of species by using phylogenetic comparative methods under a Bayesian framework. As eco-environmental pressures are likely clade-specific, we conducted analyses both at the order level, which includes all species of all families and at the family level. We performed analyses for those nine families for which we had at least 50% of the species represented in the phylogeny, or more than 100 species, to have sufficient statistical power and to avoid inferring spurious patterns. Of the 1321 species for which we measured brightness levels, we only considered 1249 species due to the latest taxonomic reclassifications.

First, we ran the model across the entire phylogeny (hereafter total model) (Fig. 3) and found that (1) body mass is negatively correlated with brightness (Supplementary Fig. 1), (2) latitude is positively correlated with brightness (but only from the tropics until the interface between subtropical and temperate regions) (Supplementary Fig. 2), (3) brighter integuments tend to be selected at low elevations (Supplementary Fig. 3), (4) diurnal species are not substantially brighter than those with nocturnal habits (Supplementary Fig. 4), and (5) brighter integuments are favored in open habitats compared to closed habitats, whilst subterranean habitats are less accurately predicted (i.e., with larger credible intervals; Fig. 4). Furthermore, we also detected that cryptic species in open areas are overall the brightest (Supplementary Fig. 5a), and negative interactions between altitude and habitat openness, where for example open habitats and low elevations contain brighter species than closed habitats at low elevations (Supplementary Fig. 5b).

Family-level analyses also show that the brightness of species is consistently influenced by habitat openness: dark colors are selected in closed environments and bright ones in open areas (Fig. 3). Moreover, we also found complex interaction effects between body mass and habitat openness in Tropiduridae in which smaller species in open

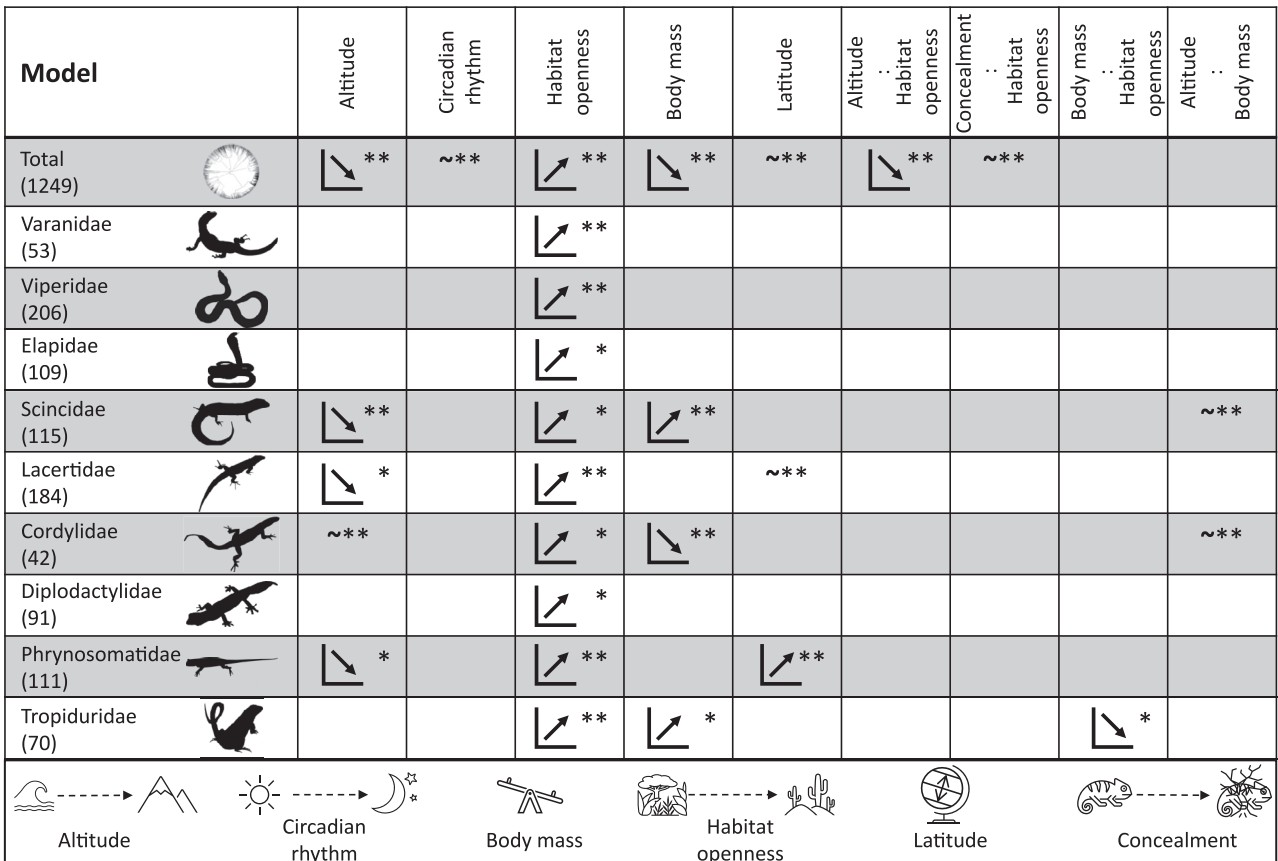

**Fig. 3 | Associations between eco-environmental variables and brightness.** Here, we show the summary output of our models at global and family levels. Two asterisks (**) indicate strongly supported parameters (variable present in all mostly supported models (Deviance Information Criterion (DIC) < 5) and has a cumulative Akaike weight of > 0.75); One asterisk (*) less strongly supported (variable present in any of the mostly supported models (DIC < 5) and has a cumulative Akaike weight of > 0.75). An increasing trend indicates a positive relationship between the variable of interest and brightness, and vice versa a negative one. "~" displays no clear trend between the variable of interest and brightness. Altitude: category from low-to-high altitudes; Circadian rhythm: category from day-to-night activity patterns; Habitat openness: category from closed-to-open habitats (fossorial species are not considered here); Body mass: continuous from small-to-large species; Latitude: category from low-to-high latitudes; Concealment: category conspicuous or cryptic. In all analyses, we included polymorphism (binary state: Yes, No) as a random effect to account for intraspecific color variation. For visualization purposes we omitted non-supported interactions such as between latitudinal distribution and body mass. Numerical values in parentheses show the number of species analyzed within each model. Empty cells denote a lack of significant correlation. Silhouettes from phylopic.org.

areas are darker than in closed areas - and as body mass increases this relationship reverses - (Supplementary Fig. 13). Also, in Cordylidae and Scincidae smaller species at low elevations tend to be brighter (Supplementary Figs. 14, 15). Overall, body mass, latitude, and altitude influence the brightness of species but only in those families where species have a relatively small size (i.e., Scincidae, Cordylidae, Tropiduridae, Phrynosomatidae, Lacertidae). Furthermore, lower altitudes likely select for brighter species in Scincidae (Supplementary Fig. 6); body mass positively (Tropiduridae, Supplementary Fig. 8; Scincidae, Supplementary Fig. 10) and negatively (Cordylidae, Supplementary Fig. 9) influences brightness; and Lacertidae are darker (Supplementary Fig. 11) and Phrynosomatidae generally brighter (Supplementary Fig. 12) at higher latitudes. Surprisingly, Varanidae, Viperidae, Elapidae, and Diplodactylidae did not show any association with any of these variables.

## Brightness evolution rates and climatic changes

To understand how quickly color brightness evolves in response to changing habitats we evaluated seven families for which we had at least 50% of the species represented in the phylogeny from the previously selected nine. Here, we applied more stringent selection criteria than before because evolutionary rates are highly sensitive to incomplete phylogenetic sampling and limited data. We found significant positive correlations between $\delta^{18}O$ and rates of color brightness evolution

(Supplementary Fig. 17), as well as higher optimum θ values for color brightness evolution in open habitats (Supplementary Fig. 18), which associate with the aridification phases during the Miocene-Pliocene and Eocene-Oligocene. As another line of evidence, we recovered highly similar patterns when mapping changes in evolutionary rates over time in response to variations in $\delta^{18}O$ for our baseline trait, under different simulated datasets produced by adding noise (ranging from $\sqrt{10}$ to $\sqrt{200}$), and accounting for measurement errors (Fig. 5 and Supplementary Fig. 35). Specifically, all families followed the same late burst patterns when examining only simulated data (Supplementary Fig. 35). We also found similar patterns when accounting for measurement errors, but interestingly, few deviations emerged. For Varanidae, we observed an early burst pattern in the baseline model (i.e., no noise, only measurement error) as well as under a simulated low-noise condition. In Phrynosomatidae, the baseline and all simulated data, except one with moderate-high noise, followed early-burst patterns (Supplementary Fig. 35). In 92% of cases (77 out of 84 models), following a long-equilibrated, static period, the tempo of brightness evolution increased at the interface between the Miocene and Pliocene (Fig. 5a, cand Supplementary Fig. 35), paralleling the cooling and aridification period observed from fluctuations in $\delta^{18}O$ over the past 95 Myr[28]. Similarly, in the remaining 8% of cases high rates were also found following two rapid aridification phases during the Eocene-Oligocene. To visualize this better, we zoomed into these epochs

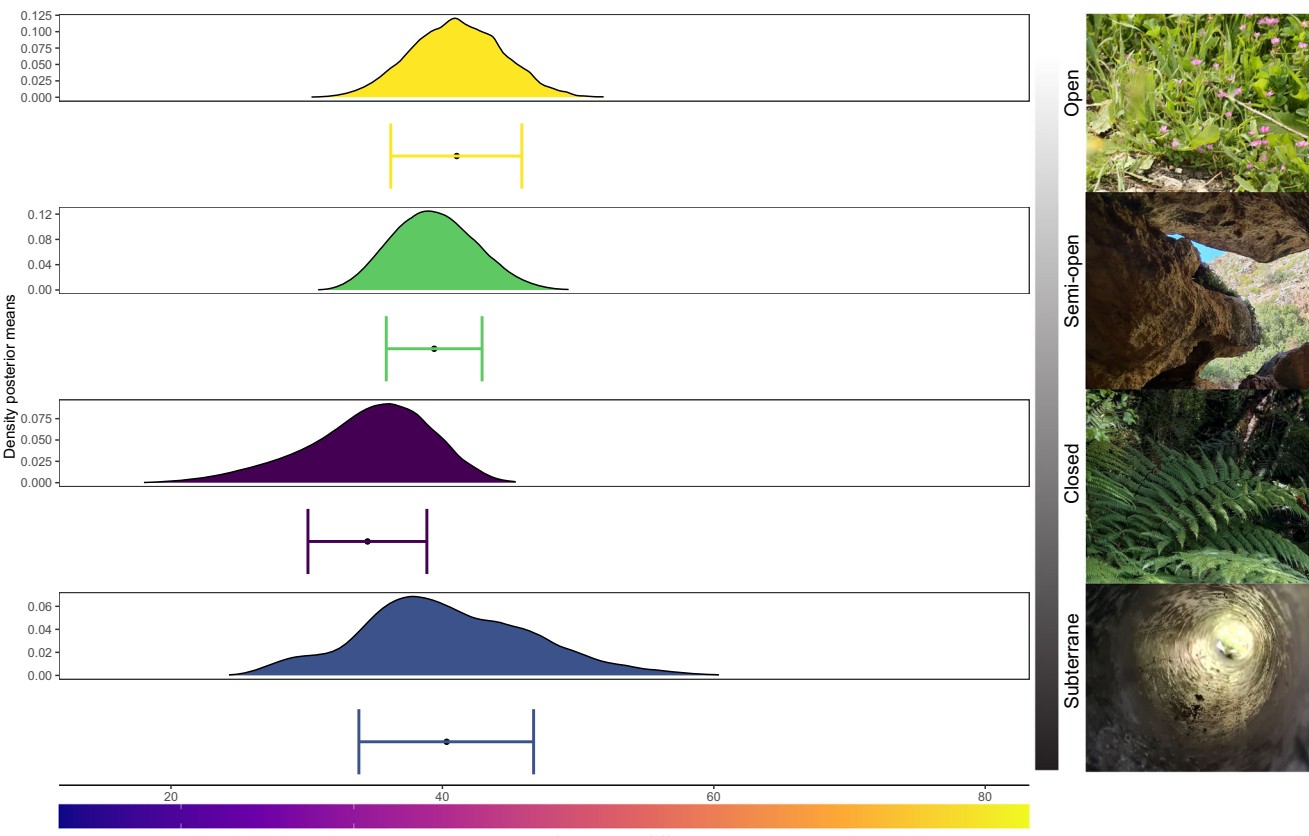

**Fig. 4 | Habitat openness predicts dorsal color brightness.** Predicted association between habitat openness and dorsal brightness of squamates. Dots indicate mean predicted values. Bars represent 95% credible intervals, and curves show posterior means density plots. The denser the vegetation, the darker the species get. Sample size (*n*): Open = 297; Semi-open = 596; Closed = 271; Subterrane = 85. Representative vegetation images by JG. Colored density plots designate different vegetation types as in Fig. 2. Vertical gray bar represents the light exposure gradient from open to subterrane habitats.

(Fig. 5b, d), and discovered that most rates increased starting in the mid-Miocene, possibly compounded by the establishment of the cooler climate during the Pliocene (5.3 Ma ca.), followed by a further temperature decrease during the quaternary glaciation (2.5 Ma ca.). In addition, the early-burst patterns in Phrynosomatidae and Varanidae were associated with the rapid cooling following the Azolla and Eocene-Oligocene events. These events likely promoted habitat openness shifts, ultimately inducing species to adapt to new vacant niches. We then compared the rates of brightness evolution between groups (Fig. 5e, Supplementary Figs. 19–25 and Supplementary Table 1) and found that they were highest for Phrynosomatidae (mean rate ($\sigma^2$): 6.02 ± 0.81 standard error (SE)) and Viperidae (mean $\sigma^2$: 4.48 ± 0.44 SE) and lowest for Cordylidae (mean $\sigma^2$: 1.91 ± 0.41 SE) and Diplodactylidae (mean $\sigma^2$: 3.21 ± 0.48 SE).

## Model evaluations

Squamate level analyses: To rule out dataset-construction biases, we then assessed the robustness of our model outputs by repeating the same analyses on different subsets. First, we included only those species for which we collected five or more images (i.e., 1033 species) to apply a more conservative brightness acquisition process, and the results did not change from the total model (Supplementary Table 2). Then, we ran the analyses without fossorial species (i.e., *n*: 1164) because brightness levels in subterranean environments, which are dark and resource-limited, are likely determined by factors other than thermoregulation and crypsis. Again, the model predicts the same patterns as the total one (Supplementary Table 2) but with higher support. This is evident from the smaller credible intervals when comparing the models "without fossorial" vs. "with fossorial" (our total

model) species across different habitats: Open (41% [35,44] vs. 41% [36,46]), Semi-Open (39% [36,41] vs. 39% [36,43]), and Closed (34% [29,37] vs. 36% [31,39]). These smaller credible intervals confirm the higher uncertainties of brightness prediction in fossorial species (Fig. 4).

Phylogenetic composition: We also evaluated if the phylogeny itself may influence the outcome. We tested this with vipers and cordylids by contrasting the models previously produced for these families (Fig. 3) with new ones using the phylogeny of Alencar et al.[31] and Stanley et al.[32] respectively. Once again, we retrieved the same patterns (Supplementary Table 2): brightness of species is consistently affected by habitat openness, with dark integuments favored in closed canopy environments and bright ones in open areas.

Image selection: We also assessed how image selection may affect the outcome by bootstrapping three times the brightness values, using the subset with five or more images. Results firmly supported the patterns reported above (Supplementary Table 2).

Next, we performed 25000 comparisons of brightness values of images within and among species to estimate variations between and within species. As expected, we found that the mean ($t = 60.92$, df = 193676, $p \ll 0.01$) and variance ($F = 1.38$, num df = 99836, denom df = 97506, $p \ll 0.01$) are significantly different between groups (Supplementary Fig. 26), which indicates that differences in brightness are smaller within than between species.

Smoothing parameters: For the evolutionary rate analyses we evaluated the robustness of our models by fitting nine different smoothing parameters (lambda: $\lambda$), apart from our reference $\lambda = 1$, to the data spanning from $\lambda = 0.5$–$0.9$ to $\lambda = 2$–$5$, and found that the rates of evolution are consistent (Supplementary Figs. 27–33).

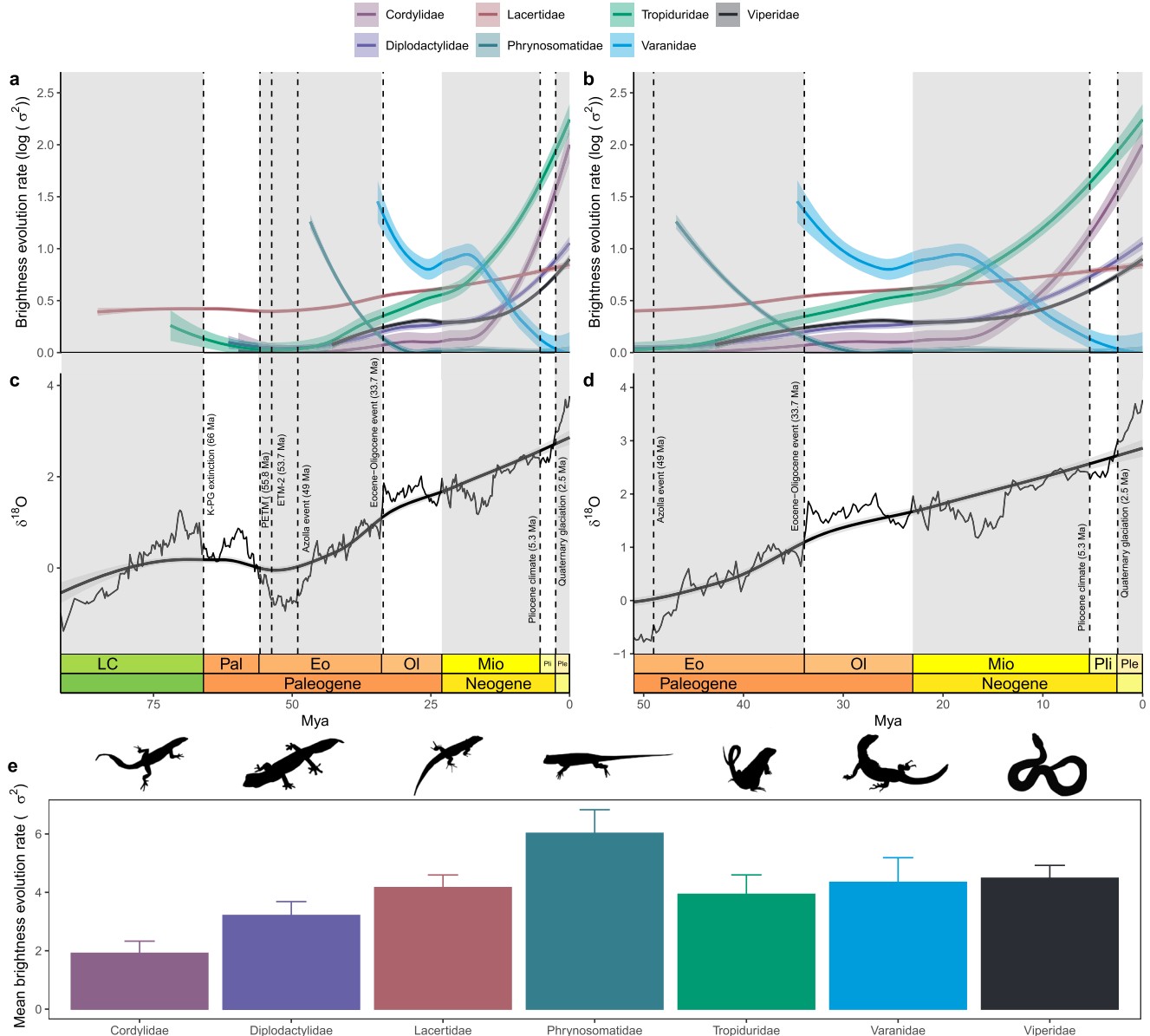

**Fig. 5 | Rates of brightness evolution in squamates.** On the left panels (**a**, **c**) rates of color brightness evolution ($\sigma^2$) in log scale for each clade, since their roots, are plotted against time and compared to variations of $\delta^{18}O$ (our proxy for habitat openness change; data from Gaskell et al.[28]). On the right panels (**b**, **d**) data are zoomed into the Eocene – Pleistocene interface. All displayed rates account for measurement errors. Dashed lines indicate major global climate shift events induced by geological or environmental processes. Epochs, periods, and their acronyms and color codes follow the International Geological Time Scale as set by the International Commission on Stratigraphy (ICS). For visualization purposes we omitted individual data points, which we report in Supplementary Fig. 16.

Correlation analyses between rates of color brightness evolution and $\delta^{18}O$ across focal families are shown in Supplementary Fig. 17. Shaded areas around solid curves (i.e., loess lines) show 95% C.I. Bottom panel (**e**) shows mean rates of color brightness evolution ($\sigma^2$) + standard errors for each focal clade. Individual data points are not included, as the rates represent model-based estimates derived from the analysis of extant trait data across the trees. For visualization purposes, we only show the positive standard errors. Sample size (*n*): Cordylidae = 42; Diplodactylidae = 91; Lacertidae = 184; Phrynosomatidae = 111; Tropiduridae = 70; Varanidae = 53; Viperidae = 206. Output produced with the R packages deeptime[94], phytools[87], RPANDA[88], and ggplot[77]. Silhouettes from phylopic.org.

Evolutionary models: To detect whether climate change through time has shaped the rate of evolution of color brightness, we compared a single rate model, which is independent of habitat openness, to multi-regime OU, single optimum OU, and BM models where the rate regimes depend on changes in habitat openness. We found that the OU multi-regime model has overall the highest support (wAIC across families = 0.66–1.00), indicating that habitat openness has an important role in shaping the evolution rate of color brightness. Finally, as we found that the OU phylogenetic structure fits the data better, we evaluated how such a phylogenetic structure influences the output of the comparative model.

Results predict the same patterns as the total model (Supplementary Table 2).

Altogether, these results are robust against perturbations in both phylogeny and coloration data. Even though there is a large body of literature using photographic material[33,34], our sensitivity analyses show that potential issues related to using non-standardized material are not qualitatively affecting any of the results.

## Discussion

Coloration evolves under multiple competing selective forces[19] from diverse ecological and environmental factors[35], which likely vary from

one clade to the next. This makes it difficult to isolate drivers of global patterns of color variation. For example, due to melanins' unique properties[14], dark integuments should be favored in cold environments to maximize heat retention[16], but also favored in hot and humid environments to provide antimicrobial or UV protections, or for camouflage[36]. Given such opposing predictions, it is remarkable that we have here identified a single factor associated with global patterns of its evolution. Habitat openness modulates the temperature of the environment and determines the amount of light that reaches organisms[37,38] which ultimately influences their social signaling and sexual selection[39], prey-predator relationship through fine-tuned concealment levels[40], locomotion types and thus habitat exploitation[41], and thermoregulation[42]. Because all these factors are intertwined with the colored integument, habitat openness has a critical influence on its ecology and evolution at a global scale.

Indeed, our analyses suggest that the rapid evolution of brighter and darker integuments may have allowed species to radiate into new open and closed habitats, respectively, that arose during global cooling phases, observed, for example, at the interface between the Miocene and Pliocene. This phase, marked by the substantial increase of $\delta^{18}O$[28] and decrease of $CO_2$[29] during the mid-late Miocene, promoted aridity at the global level, leading to large-scale shifts in landscapes favoring the establishment of $C_4$ (i.e., plants that thrive best in dry climates) over $C_3$ (i.e., plants that thrive best in wet climates) vegetation[29]. Such vast changes produced new vacant niches in which species could radiate, and our results suggest that integumentary brightness is a key phenotype that helped species adapt to these novel conditions. This is because integumentary brightness modulates heat absorption, where, for example, brighter integuments generally provide thermal advantages in hotter, open area conditions by reflecting incoming heat[16].

To further understand how coloration may shift across geographical and environmental gradients we considered palaeoclimatological patterns. Major geological (e.g., volcanic activities) and environmental (e.g., algal blooms) changes induced profound shifts in Earth's climate and overall ecosystem structure[43], promoting new ecological opportunities, ultimately leading to adaptive radiation[44]. For example, changing habitat composition[45], morphological evolution[46], resource availability[47], reduced predation pressure[48], and mass extinctions[49] have all been key triggers for adaptive radiation. Moreover, geologically sudden events such as the interface between the Miocene and the Pliocene followed by the Quaternary glaciation may explain patterns of punctuated equilibria in the fossil record[50]. Our findings follow such a pattern in that long evolutionary stasis in brightness is followed by a rapid burst in variation associated with new environmental conditions (Fig. 5), a trend exhibited across all families under different, simulated noise levels (Supplementary Fig. 35). Although most families followed this pattern even when accounting for measurement errors, Varanidae and Phrynosomatidae partially deviated, showing potential early adaptations and static rates. These early burst and static patterns were observed in only seven out of the 84 examined evolutionary rate models, which are possibly driven by the rapid temperature decrease following the Azolla (49 Mya ca.) and the Eocene-Oligocene extinction (33.5 Mya ca.) events in Phrynosomatidae and Varanidae respectively. Whether these patterns represent a biological signature or reflect other factors remains uncertain. Potential biases, such as measurement error and noise in rate estimates, may contribute to the observed trend, making it difficult to fully disentangle true evolutionary signals from artifacts. While our analyses attempt to account for these biases, some residual effects remain, highlighting the need for more robust simulations and sensitivity tests. While this study focused on extant species, well-preserved squamate fossil records could provide an opportunity to empirically describe their colored integument, thereby calibrating trait reconstructions and offering further insights into the early evolution of these families.

Remarkably, the observed rapid evolution in the mid-late Miocene - early Pliocene has been detected in different systems and phenotypes spanning from plants[51] and animals[52] to even fungi[53], suggesting that shifting landscapes in this period had a profound impact on shaping current global diversity[54].

Nevertheless, selective forces shaping phenotypic evolution are seldom widely consistent. We analyzed the data at order and family levels and found that some increased their brightness (e.g., Iguanidae, Gekkota) at the root of the group, while others (e.g., Elapidae, Teoidea) decreased it, and at different rates, possibly because of different life-history strategies that evolved in response to regional eco-environmental conditions. Such contrasting signatures at family or higher taxonomic levels decrease the likelihood of identifying unified mechanisms regulating global patterns of color variation across a vast phylogenetic group using traditional ecogeographic rules[14]. In fact, in contrast to our expectations, we did not detect any consistent effect of brightness variation for any sampled eco-environmental parameter besides habitat openness. The lack of influence of the interaction between concealment type (either cryptic or conspicuous) and habitat openness at the family level, where integumentary color should match the light environment for crypsis[55], was particularly unexpected but suggests that camouflage may be less significant than other factors. Similarly, although diurnal and nocturnal habits are linked to various morphological and environmental adaptations[56], our results indicate that both diel activities shape color brightness in similar ways. Instead, other factors such as habitat openness and other examined environmental conditions likely have a more pronounced impact on the evolution of dorsal brightness in squamates. Moreover, we also performed analyses without accounting for fossorial species because conditions below ground differ significantly from those above, potentially influencing coloration through selection pressures such as production costs[57], antimicrobial[58] and heavy metal[59] protection. Given their numerous abundance across squamates[25], focusing on the selection pressures driving color variation of fossorial species, such as many skinks[56], will be an exciting avenue for future research.

We found that the brightness of species in families with relatively small body sizes is also negatively influenced by altitude (Scincidae) and positively (Scincidae and Cordylidae) and negatively (Tropiduridae) by body mass, suggesting contrasting selection pressures within these small-bodied groups compared to larger organisms. Smaller organisms heat and cool faster than larger organisms[60], and are thus more rapidly susceptible to surrounding climate conditions. Neglecting behavioral adaptations[61], it follows that colder or hotter environments found, for example, along an altitudinal gradient have a major influence on the colored integument of smaller species for thermoregulatory purposes, as larger species maintain a more constant temperature via their mass[62]. Similarly, as body size modulates the heat transfer within an organism and its surroundings[60], our findings indicate that smaller species, due to their rapid heat exchanges, evolved targeted adaptations (e.g., larger and brighter or smaller and brighter) to cope with specific ecological pressures. Surprisingly, the integumentary brightness of front-fanged snakes (Elapidae), vipers, varanids, and diplodactylids could not be explained by any parameter but habitat openness. Unlike other families, these groups have remarkably variable ecologies, ranging from fossorial, arboreal, terrestrial, and aquatic habitats to small species like *Drysdalia coronoides* (18 cm) and large like *Ophiophagus hannah* (> 5 m), as well as diurnal, cathemeral, nocturnal rhythms[63,64]. This ecological variability promotes specialized adaptations at smaller phylogenetic scales, such as at the genus level, thereby potentially eclipsing general patterns[14].

Our results suggest that brightness variation of squamates is associated with habitat openness and its underlying selective forces, and also elucidate the complexity and trade-offs between the mechanisms driving their color variation across different phylogenetic levels. This emphasizes the importance of examining patterns of

phenotypic evolution at different taxonomic scales to underpin the forces regulating global trait adaptations. By leveraging the database presented here, future research can gain deeper insights into how species may respond to changing environmental conditions, providing information for predicting species' responses to future climate scenarios[65], helping to inform conservation strategies, and our understanding of biodiversity dynamics.

## Methods

### Dataset construction and species selection

To assess the evolution of dorsal brightness in squamates, defined here as the mean of the RGB values[19], we compiled a species-level dataset with ecological and environmental variables known to influence the colored integument. The collection included data acquisition from the literature (i.e., habitat openness, body mass, latitudinal and altitudinal distributions, and circadian rhythm) and visual analyses (i.e., concealment level and color polymorphism), which are all explained in detail below. For the visual analyses, we examined 10638 images (mean per species x̄: 7.76 ± 3.51 SD). Moreover, we acquired brightness information through analytical methods (see below). We selected one of the most complete genetic-based species-level phylogenies available to date for squamates[23], which is a time-calibrated tree based on 52 genes for 4162 species because we prioritized phylogenetic accuracy over sampling. We initially collected information on 1372 taxonomically unambiguous species (12 % of the order) from 25 families covering major groups and capturing different ecological strategies and evolutionary paths; but as images of aquatic species ($n = 33$) could not be consistent due to different light conditions, and we could not retrieve image data for 18, we ultimately analyzed 1321 species. Our aim in this study was to provide evidence for broad-scale patterns. As such, further investigations are needed to underpin specific trends within and across genera of the same clades. In all cases, we only examined adults within naturally occurring habitats, distributions, and coloration. In addition, we did not collect information for rapidly changing color species such as chameleons[66], agamas[67], or anoles[68] to preclude non-examined factors such as stress conditions. The dataset, with all references, is available at the supplied repository.

Habitat openness is a key variable regulating temperature and the amount of light penetrating the vertical layers[37,38], ultimately affecting species' coloration for thermoregulation, concealment, and social and sexual signaling[39,42,69]. Following descriptions in the literature, we identified six major habitats in which species occur that distinctively influence ambient light and temperature conditions. (1) "Open" habitats in which species entirely or mostly dwell in relatively low vegetations (i.e., ≤ 5 m), e.g., grasslands, deserts, plains, tundra, forest clearcuts, or any other area free of tree cover; (2) "Closed" habitats in which species entirely or mostly live in mid-high vegetations (i.e., > 5 m; arboreal species are confined to this group), e.g., forests, woodlands or any other area with tree cover; (3) "Semi-open" where species occur on both "Closed" and "Open" habitats (rupicolous and partially arboreal species fall within this group) and/or where high vegetation is not fully closing the canopy, e.g., edge effect of the forest; (4) "Fossorial" comprises species that entirely or mostly spend their time underground or in caves; (5) "Marine" indicates marine species; and (6) "MarineInter" includes marine and intertidal species. We used the cutoff of 5 m height to distinguish between "Open" and "Closed" habitats because woody plants, differently from shrubs, are usually 5 m tall or higher[70].

Body mass has a significant influence in modulating the heat exchange between an organism and its surrounding environment, and likely interacts with the colored integument for regulating the body temperature[14,60]. Thus, we collected published body mass data[71].

Following thermally dependent predictions that darkly colored species will be favored in cold environments to maximize heat retention[16], we compiled latitudinal and altitudinal distributions.

Latitude follows the categorical classification system of Goldenberg et al.[19] as very little information is available on density distribution across species ranges. Moreover, as latitude is linked with solar radiation it indirectly accounts for broad temperature variations. For altitude we provided four measures. A categorical one which classifies ranges as "Low" ($x \leq 500$ m), "Low-Medium" ($x \leq 1000$ m), "Medium-High" ($x > 500$ m $\lor$ $x > 1000$ m), "High" ($x > 1000$ m), "All" (all the range) following established threshold classes[19] (see Supplementary Fig. 34 for altitude distribution). And three continuous, i.e., (i) minimum elevation range (m); (ii) maximum elevation range (m); and (iii) difference between maximum and minimum range (m). As our models performed better with categorical variables (DIC: categories = 8052.05; continuous = 8512.52), we used this classification system to report and discuss our results.

Circadian rhythm can also have substantial influence on the coloration of organisms for thermoregulation, concealment, and social and sexual signaling[72–74]. We identified "diurnal", "nocturnal", and "cathemeral" species, and because it is challenging to differentiate between crepuscular and cathemeral species, we merged the two strategies into cathemeral.

Concealment likely interacts with habitat openness, where open habitats select for brighter species to maximize concealment with the surrounding, whereas closed ones for dark for opposing reasonings. We characterized species as "Cryptic", "Conspicuous", or "Uncertain" by evaluating the collected images following the established classification of Arbuckle and Speed[75].

We also scored the presence or absence of color polymorphism to account for intraspecific variation by visual examination. As this qualitative classification may be subject to observation bias, five independent researchers scored a subset of 100 species, and as we found a strong positive association between scores (r: 0.92, $p << 0.001$), only one continued the classification.

Finally, as direct spectrophotometry measurements of live squamates are logistically challenging, we retrieved brightness levels from 10638 images from peer-reviewed articles, field guides, Google Images, documentaries, and our own data. We removed images that were clearly over-or-under-exposed. To account for lighting and setup conditions, we took all the precautions described in Goldenberg et al.[19], such as selecting multiple images per species, and excluding over- or underexposed regions from the focal area. Supported by previous research on squamates that demonstrated strong positive correlations among UV, Vis, and NIR[19], this workflow supports the use of mean RGB values from the dorsum as a proxy for brightness. We measured the brightness of species from the selected images with the interactive, custom-made plugin MacroBright v.0.1[19] in ImageJ v. 1.53q which guides the user to select the region of interest from which the researcher can obtain the brightness level.

### Descriptive analyses

Visualizing how an examined trait is distributed across the globe can provide important ecological clues. As such, we visualized how brightness is spatially distributed across our sampled species in relation to major biomes. We obtained species distribution data from Roll et al.[30] merged them with brightness measures, and plotted the spatial variation by using "rgdal"[76] and "ggplot"[77] in R v.4.1.3[78]. For visualizing major terrestrial biomes we made use of the spatial data of Olson et al.[79].

### Ancestral brightness estimation

To investigate the evolutionary history of integumental brightness, we estimated ancestral states of color brightness by comparing the BM, OU, and EB model of evolution in "phylolm"[80] and visualized them through "ggtree"[81]. We compared evolutionary models using the widely established difference in AIC values, following the thresholds

$\Delta AIC \leq 2$ (models equally supported), $2 < \Delta AIC < 10$ (considerable support for the model with lower AIC), $\Delta AIC \geq 10$ (significant support for lower AIC model). Furthermore, as we found clustering of brightness in the spatial visualization where major biomes likely select for different colored integuments, we annotated the habitat openness associated with each species on the figure.

### Associations between eco-environmental variables and brightness

To examine the probability that a species will display a specific brightness in response to our selected eco-environmental variables, we performed multiple Markov chain Monte Carlo Generalized Linear Mixed Models ("MCMCglmm"[82]) while accounting for phylogeny using the pedigree command. We initially constructed a global model where all species were included. Then, as different clades may be subjected to contrasting selection forces, we performed identical models at the family level. For statistical power and to avoid false evolutionary patterns[83], we selected nine families for which we had at least 50% of the focal species represented in the phylogeny or more than 100 species. In all cases, we set the brightness as the response variable and habitat openness, altitude, latitude, body mass, circadian rhythm, and their interactions as fixed effects. Moreover, we included the interaction between habitat openness and concealment level as a fixed effect. To account for intraspecific color variation within and among sexes, we indicated polymorphism as a random effect. We verified which variables better explained variation in the system using the dredge function in "MuMIn"[84] ranking by Deviance Information Criterion (DIC). We defined variables as strongly supported if they were present in all mostly supported models (DIC < 5) and had a cumulative Akaike weight of > 0.75[19]; less strongly supported if they were present in any of the mostly supported models (DIC < 5) and had a cumulative Akaike of > 0.75. We ran all models setting 100000 MCMC iterations, burnin = 40000 and thin = 20. As we did not have a prior knowledge of how brightness is associated with different eco-environmental variables and because all models converged (see below), we defined the default priors[85], and family (a character vector describing the trait class) was set as "gaussian". To assess convergence, we ran five models to verify if they converged to the same posterior distribution. The Gelman and Rubin criterion shows that the point estimate values of the variables of interest are all confined between 1 and 1.01, and the multivariate psrf is 1. The results of the Gelman and Rubin criterion, coupled with trace plots, indicated that our model successfully converged. We followed the default behavior of MCMCglmm, which incorporates measurement error by specifying the variance structure as $VG \otimes D$[85]. In this structure, D is a diagonal matrix representing the measurement error variances, and VG is a scalar fixed at 1.

### Evolution rates and association with $\delta^{18}O$

Driven by the consistent brightness association with habitat openness, we investigated the evolution rate of integumental brightness and how it relates to published variations of foraminiferal $\delta^{18}O$[28], here used as a proxy for habitat openness change over time[29]. To measure the evolution rates we used the recently developed penalized-likelihood method *multirateBM*[86] in "phytools"[87]. For statistical power and to avoid potential biases in estimating evolutionary rates, we applied more stringent selection criteria than previously, and we examined only seven families for which we had at least 50% of the focal species represented in the phylogeny. We then evaluated the robustness of our evolutionary rate analyses by fitting nine different smoothing parameters (lambda: $\lambda$) to our reference $\lambda = 1$, spanning from $\lambda = 0.5-0.9$ to $\lambda = 2-5$ using "phytools"[87]. As another line of evidence, we evaluated how evolutionary rates change over time in response to $\delta^{18}O$ using *fit_t.env* and *plot.fit_t.env* in "RPANDA"[88]. This allowed us to examine the effects under different levels of simulated noise and measurement error, broadly defined as low ($\sqrt{10}$, $\sqrt{50}$), moderate ($\sqrt{100}$, $\sqrt{150}$), and high ($\sqrt{200}$), given their established importance in phylogenetic comparative methods[89]. We also evaluated whether the observed pattern of rates of color brightness evolution are linked with changes in habitat openness by fitting multi-regime OU and BM models (i.e., changes in rates depending on the mapped habitat openness) and comparing them to a single rate model (i.e., independent of habitat openness) by simulating stochastic character maps on the phylogenetic trees using *make.simmap* in "phytools"[87].

### Statistical power evaluation

Evolutionary patterns, as in all statistical analyses, can be influenced by the input data. To hamper false evolutionary patterns, we then examined the robustness of our models by repeating the same analyses on different subsets. First, we included only those species for which we collected five or more images (i.e., 1033 species) to apply a more conservative brightness acquisition process. Then, we ran the analyses without fossorial species (i.e., 1164) because integumental brightness in subterranean environments, which are dark and resource-limited[90], is likely shaped by other selective forces than thermoregulation and crypsis. We also bootstrapped our brightness values to assess how image selection may affect the results. From the subset for which we analyzed five or more images, we randomly selected three images per species, extracted the brightness values, and then analyzed the data as previously described. We repeated this bootstrap process three times. We then fitted an OU model of evolution to evaluate its performance against a BM one. We extracted the $\alpha$ value using "phylolm"[80] and rescaled the tree using the *rescale* function in "geiger"[91]. The tree was then incorporated into the MCMCglmm. We further evaluated if the phylogeny itself may affect the findings. We tested this on vipers and cordylids by comparing the models previously analyzed using the phylogeny of Zheng and Wiens[23] against the ones using Alencar et al. and Stanley et al. phylogenies[31,32]. Finally, to estimate how brightness varies among and within species, we performed 25000 comparisons contrasting two images that were either from the same or different species.

### Reporting summary

Further information on research design is available in the Nature Portfolio Reporting Summary linked to this article.

## Data availability

All data associated with this manuscript, including all raw data[23,28,31,32], are available on Zenodo: https://doi.org/10.5281/zenodo.14866945[92].

## Code availability

All R-scripts and the associated files necessary to reproduce all figures and outputs are available through our repository on Zenodo: https://doi.org/10.5281/zenodo.14866945[92].

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

## Acknowledgements

We thank all students that helped us collect color brightness data, especially Florian Van Hecke, Bram De Vilder, Daphne Hessmann, June De Groote, Elise Nieuwenhuijze, Dido De Naeyer, Luciane Marie Maessen, and Vasiliki M. Oikonomaki. We further thank the Feiner-Uller lab and Charlie Cornwallis for productive discussions. Wenner-Gren Foundations (UPD2022-0061 to J.G.) Research Foundation Flanders (FWO, 12T5622N to K.B., GOG2217N to M.D.S., and GOA7921N to M.D.S. and L.D.). Belgian American Educational Foundation (B.A.E.F., to M.P.J.N.). Bijzonder Onderzoeks Fonds UGent (BOF, to M.P.J.N.). Human Frontiers Science Program (RGP0047 to M.D.S.). Air Force Office of Scientific Research (FA9550-1-18-0447 to M.D.S.). European Office of Aerospace Research and Development (FA8655-23-2-7041 to M.D.S.)

## Author contributions

J.G. conceptualized, lead the analyses, and drafted the manuscript. J.G. and K.B. collected the data. K.B., J.W.L., and M.P.J.N. supported the analyses. J.G., K.B., J.W.L., and R.S.E. designed the methods. R.S.E., L.D., and M.D.S supervised the work. All authors contributed to the writing of the manuscript and the revision process.

## Funding

## Competing interests

The authors declare no competing interests.
