## [Transparent Peer Review file · Nature Communications]

Habitat openness and squamate color evolution over deep time

Corresponding Author: Dr Jonathan Goldenberg

Version 0:

Reviewer comments:

Reviewer #1

(Remarks to the Author)

Reviewer report for NCOMMS-23-56634

Key results – The manuscript presents a compelling and novel results on the evolution of colour brightness variation in global squamates. The main result, that brightness correlates closely with environmental variation, both spatially, temporally, and across clades, provides cogent answers to a relevant and hitherto mostly unexplored biological question. First, the evolution of these brightness features is visualised across different biomes, showing a clear prevalence of brighter colourations in open habitats, but not clearly across wide latitudinal gradients. Second, it is mapped across a phylogenetic tree to show the phylogenetic clustering and historical trends across the evolution of squamates. Further analyses evaluate the relations between brightness and a series of habitat variables, finding that vegetation structure is the strongest predictor of colour brightness evolution, as also shown previously in the geographical and phylogenetic visualisations. Finally, the manuscript reconstructs brightness evolution through time in a sample of squamate clades, showing how it tracks together with variations in climatic conditions as predicted by the variations in oxygen isotopes, which the authors take to represent an evolutionary response to variations in vegetation structure.

Validity – I believe that the study takes on an interesting biological question and does a good job at addressing it from a variety of different angles. I particularly appreciated how phylogenetic, ecological and spatial patterns are integrated in a compelling interpretation of the history of colour brightness in squamates. However, I do have some concerns with the interpretation of some of the results, due to the insufficient justification for the use of your proxy for vegetation structure. I do not take issue with the results themselves, here, but with the choice to use (and here I quote your own text) 'variations in foraminiferal $\delta^{18}O$ as a proxy of vegetation structure change over time'. I went and checked the article you used as a reference, Herbert et al. (2016). While the article does say that there were changes in the vegetational compositions in different periods due to shifting climates, there is little that explicitly addresses vegetation structure itself beyond mentioning that there was a transition between wet and dry climate plants. In my opinion, here there is a leap in logic that should at least be better expanded upon. These estimates are based on variations of sea temperatures and levels through time, which in turn are estimated based on variations in oxygen isotope concentrations. It just does not seem to be strong enough of an association to be used as a proxy of vegetation structure, not without a clearer and stronger supporting explanation. To be clear, I am not saying that I think these interpretations are invalid, just that they should be better explained to the reader across the text. Additionally, I do have some concerns with the sampling strategy used in the article (see Data & Methodology section).

Significance – This work represents a valuable scientific contribution to a relatively unexplored field. Specifically, it is significant because a) it addresses a series of biological questions that have only been taken into scarce consideration by the existing literature, and b) it does so in Squamates, a study system which is well-suited for this type of study, having an impressive taxonomic diversity, global distribution, and a long and complex evolutionary history. The way these questions are addressed uses well-established analytical approaches which could allow their extension to broader taxonomic categories. Thus, I believe that this article will be a valid addition to the existing literature on the topic.

Data & methodology – The data used in this article appears to be generally well justified in both its acquisition and its uses, with some exceptions that need addressing. In particular, I have some issues with the naming of certain variables, and their representation within the article. Additionally, I am adding some notes and observations related to data collection and methodology. Below, I will list the things that, in my opinion, would benefit from your consideration for the improvement of this work:

-vegetation structure: the name of this variable is not entirely appropriate to indicate the categories that you use in your work, specifically in the light of one of those categories, 'Subterranean'. What does living underground have specifically to do with vegetation? A lizard can be fossorial in a forest as well as in an open plain. I believe that these categories could be better framed as 'habitat openness', with subterranean being the least open of all categories. Additionally, the Open/Closed category would probably be more understandable if it were named simply 'semi-open'.

-Related to the previous point, on a more methodological note, while you did run analyses excluding fossorial (subterranean) species, I wonder whether you considered examining the effect that fossoriality specifically has on brightness variation (against all other categories). Fossorial species are very interesting in that they probably do not conform to the same rules as other species (i.e., thermoregulation is less likely to affect their colouration), and their numerical representation across squamate clades is very large. Writing a bit more on that in your discussion would be beneficial to contextualising your findings.

-I noticed that you included a coding of the activity patterns of lizards in your analyses, and that in the results, you mentioned that diurnal species are brighter than the others. It honestly surprised me that not much was included about this in the discussion, as it seems like a very relevant result. For example, the relationship between lifestyle and activity patterns has been discussed by recent papers (i.e., Slavenko et al., 2022), and it would be great if this type of analysis were to be contextualised in the light of colour evolution.

-Taxon sampling: While you do provide a good justification for the numerical sampling of taxa in your analyses (that is, what proportion of each clade was considered sufficient), the taxonomic (and, relatedly, geographic) diversity within each of the clades included in your work is not very well represented. While the representation within genera is quite good, there are very few genera per clade included. Additionally, there is a disproportionate numerical representation of some groups against others. This also means that many clades were essentially completely excluded from your analyses, biasing the results towards very few select genera. I am aware that taxon sampling is a difficult thing to do (and also limited by the information available), and sampling philosophy can be subjective, but I would have at least expected to see a better representation of genera across the phylogeny of each group. Using an example from your own data (on skinks because I am more familiar with this group), when more than one-fourth (33/116) of your sample of skinks belongs to the genus *Plestiodon*, and about one-fifth (21/116) by the genus *Chalcides*, how much are your results really saying about skinks in general? How much about *Plestiodon* and *Chalcides*? There are 168 skink genera, only 25 of which are included in your dataset, with most of the dataset being represented by species belonging to five or six of them. In my opinion, this is a methodological weakness that should at least be acknowledged and addressed within the article's methods and discussion.

Analytical approach – The study takes a valid and adequate analytical approach to its analyses, both statistically and in terms of data interpretation. While I would in some cases use different approaches to the same statistical/phylogenetic problems, I do not find any particular issue with any aspect of the analyses. I particularly appreciated the model validation included in the text. It does show how much careful consideration went into this article, to ensure the validity of analyses and evaluate their potential pitfalls. The phylogenetic analyses employed here appear to be sound, but I would like to see some more information about the alternative models of character evolution that you tested, and why you decided to use an OU early-burst model specifically.

Clarity and context – This article is relatively well contextualised within the existing literature, but it could use some major improvements in its clarity and stylistic choices. I will proceed to comment on each section in order:

Introduction: mostly well contextualised and clear, it sets the theoretical stage for the analyses that follow. I would have liked to see the questions/hypotheses or at least an aim of the paper to be more clearly spelled-out. And I am unsure whether you should summarise the results at the end of the introduction (that should be the abstract's role, but feel free to ignore this if that is in the journal's style). Instead, you should write what you did to address your hypotheses.

Results: a bit unclear, as it reads more like a mix of methods and results, which could confuse a reader trying to parse the text. This also means that there is quite a lot of redundancy between materials/methods and the results. This redundancy impacts clarity. The results section should be also better sectioned in smaller paragraphs to improve clarity.

Discussion: The discussion is generally well written, presents coherent arguments, and is clear in most of its parts. However, some of the results (i.e., the global relationship with activity pattern/circadian rhythm) are incompletely or not at all discussed, as the article chooses to selectively focus on some of the results instead of others. Additionally, the part of the discussion (lines 295-306) that focuses on the observed interaction between body size and altitude is very opaque, and its logic is not very clear. I read the paragraph several times, and I am not really sure of what you are trying to say with it. If your patterns are not at all consistent across taxonomic groups, perhaps it does not make much sense to try to explain them as a whole.

Materials/methods: No particular comment on this part, except what already stated in other sections of this report.

Figures: See specific comments below.

Suggested improvements – For general suggestions about the article, see each of the sections above. Below, I have listed some specific suggestions for parts of the text which require improvement.

Line 75: As this is the first time you mention the ratio of stable isotopes of oxygen, you should define it briefly.

Lines 86-88: Beyond the expected relationship between brightness and vegetation structure, it's really interesting that you found brightness not to correlate with latitudinal variation. Any reason why you think that may be the case?

Fig. 2: It is a bit hard to visualise the brightness scale on the ancestral reconstruction image. Perhaps a different (starker) scheme would be appropriate. The labelling of the different clades on the tree could be graphically accomplished better. Also, the separation between the clades is a bit hard to see. One possible suggestion is using alternating, thicker bands of two different colours above each of the clades.

Line 123: At least 50% of the focal species? What do you exactly mean by that? What are these 'focal species'?

Line 120: 'using phylogenetic comparative methods' is a bit vague. Please say which specific operation you performed.

Additionally, I am unsure whether this part of this section should go into the methods instead. Mentioning the operations that you performed outside of the methods is fine, but I think that the choices of which groups were analysed and why definitely belongs to the methods and not the results.

125-126: So, you lost more than 70 species due to reclassifications? That sounds like a lot of species lost only due to reclassification. I would have expected a loss due to species being absent from the phylogeny, not due to reclassification. If

possible, add example citations on the greater reclassifications efforts that have caused this taxonomic reduction.

Fig. 3: It's not a bad figure per se, but perhaps I would advocate removing the icons at the bottom of the image, or at least placing them at the top next to the relevant variables. They are a bit confusing in the context of the figure, otherwise. If space is an issue, you could try switching the X and the Y axis of the figure, making this operation easier.

150: Why 'surprisingly'?

Line 165 (caption): Empty cells denote a lack of significant correlation.

Fig. 4: This figure has some issues and would benefit from being made smaller and more informative. For example, the confidence intervals of the different categories could be illustrated a bit better. I also imagine that there will be some outliers, which are not shown here but would be really easy to display (depriving a reader of that information would diminish the informative value of the figure). A basic ggplot R script with `geom_boxplot` would fix these issues. In that case, I recommend keeping the colour exclusively in the confidence boxes. Moreover, there is an issue with the similarity between the white-to-black dorsal brightness bar on the Y axis, and an identical bar on the X axis that is used to indicate vegetation structure. As the categories are not explicitly continuous, it is questionable to use the variation in a continuous colour bar to indicate them. I also am unsure whether `OpenClosed` (an unfortunate category name – again, wouldn't 'semi-open' be better?) and especially `Subterrane` really represent vegetation, at least in the way they are displayed in the images. Is being in a subterranean cave 'vegetation' structure? Looks to me like this category is not about vegetation at all, but about habitat structure or type. See previous comments about this point.

Line 168: 95% confidence intervals.

Line 173: Again, it is unclear to me what do you exactly mean by 'focal species'.

Line 179: Reiterating the point from the general evaluation of the article, I am unconvinced that the variation in O2 isotopes means what you take it to mean, or at least, that the evidence you present is very solid. Yes, overall vegetation structure saw increases or decreases in certain periods, but it seems to me like an oversimplification of the real trends. Could any control factor be introduced or at least considered for latitude or (reconstructed) biomes?

Lines 204-210: This is, again, something that would be more suitable as a part of the methods section.

Line 213: It is a good thing that you checked for phylogenetic bias in these two groups, but at that point you might as well extend these analyses to all the groups. The fact that the models for those two groups are not affected by phylogeny does not imply that all the other groups will be.

Line 203: The Model Evaluation section lists a series of supplementary analyses that you performed to evaluate how robust your models are, which is a laudable effort, but I believe that its structure necessitates improvement. First, the choice to keep this section in a single paragraph makes it hard to keep track of the text. It needs to be rewritten in a more focused way that separates in at least a couple of distinct paragraphs. Additionally, I am unsure whether parts of this section wouldn't be better included in the methods. In my view, and I will leave it to the editors to weigh in on this matter, the best way to make this part work would be to include the evaluation in the methods sequentially after each original version of the model and keeping the same sequence in the results. For example, in the methods I would write something like: 'to assess the evolution of dorsal brightness in squamates, we did X. Then, to validate it, we did Y for these groups and these other groups etc...', and in the results 'The result of X was x, which was validated by doing Y.'

Lines 401-404: Model comparison should not be included in the methods but in the results. Additionally, it would be best practice to include (in the methods) a minimum threshold for model selection (under which two compared models are considered to perform not significantly better than the other: for an example, see Bergmann & Morinaga, 2019).

Lines 404-406: If you found clustering of brightness where biomes select for different colours, why was this not explicitly tested statistically?

Lines 415-419: I am admittedly a bit unfamiliar with this modelling approach, but why did you account only for interactions between two of the fixed effects, instead of all or more combinations of them, at least as a preliminary step in your model construction?

Lines 407-432: Sometimes it is unclear which functions are associated with which packages.

Lines 447-463: Here I would like to put the accent on the unnecessary redundancy between this part of the methods and the results section. As I wrote above, this section, and not the results, is where these methodological notes should belong.

Lines 295-298: I think that this part of the discussion should be explained more clearly. If the trends are all over the place depending on the clade, I do not see how you can draw a consistent unitary explanation from them. To me it reads like a circular argument.

Lines 308-312: Maybe add some citations to this information. Also, the double parentheses in the last sentences make reading difficult. Perhaps break the sentence in two.

Reviewer recommendation: The article presents valuable findings that provide novel insights on a relatively unexplored topic. However, it is my opinion that significant changes and clarifications are required to grant its acceptance. If the relevant alterations are made, I would be happy to see this paper published in the journal.

References

- Bergmann, P. J., & Morinaga, G. (2019). The convergent evolution of snake-like forms by divergent evolutionary pathways in squamate reptiles. *Evolution*, 73(3), 481-496.
- Herbert, T. D., Lawrence, K. T., Tzanova, A., Peterson, L. C., Caballero-Gill, R., & Kelly, C. S. (2016). Late Miocene global cooling and the rise of modern ecosystems. *Nature Geoscience*, 9(11), 843-847.
- Slavenko, A., Dror, L., Camaiti, M., Farquhar, J. E., Shea, G. M., Chapple, D. G., & Meiri, S. (2022). Evolution of diel activity patterns in skinks (Squamata: Scincidae), the world's second-largest family of terrestrial vertebrates. *Evolution*, 76(6), 1195-1208.

Reviewer #2

(Remarks to the Author)

General Comments

This paper analyses a large dataset of 1249 squamates, and is particularly interested in the link between brightness evolution and environmental covariables, such as habitat, altitude, and vegetation structure, which is argued to be the single most important driver of brightness evolution. The paper is well written and well structured, and of possible broad interest to the community.

I do not have any background in squamate evolution, so I will not discuss the biological conclusion, but rather focus on the statistical analyses carried in this work.

The analyses that are carried on are well described, and the reader can easily find information on which exact methods were used in the analyses. I found the associated Rmd file particularly useful, and I could run part of it on my machine without any problem, which makes this work reproducible.

The analyses use well documented statistical methods, and seem to be robust to a number of variations. However, as detailed below, I have a number of concern regarding the analyses, and their power to back up some of the claims of the paper. Potential concerns include : multiple testing, model selection, accounting for measurement error, lack of formal tests for some associations, and possible (not very consequential) code errors. As I'm not an expert in these organisms or kind of studies, some of the concerns I raise might be irrelevant, but I included them in case it might be useful.

Below are detailed some of my concerns. Attached is also a modified version of the "Squamata_Evo_Analyses.Rmd" file provided by the authors, with a few comments (marked as R chunk named "COMMENT").

I thank the authors for their interesting and well-thought work, and the Associate Editor for trusting me with a review.

Detailed Comments

* l.85: "We found that brighter species are generally found in open habitats such as grasslands, deserts, and prairielands, and darker species in forested regions..."

This is just a visual finding, comparing the maps, isn't it ? Could a formal test, coming from the geostatistics literature, back up this claim ? I don't find the visual inspection very compelling. However formal tests using phylogenetic comparative methods are carried on afterward, so maybe this is enough.

* Fig.1: It is hard to tell in some regions if the squamates are very bright, or if white is just missing data ? (E.g. in Alaska and other Northern regions.) The contrast of the purple gradient is not very good in other regions, making the visual inspection difficult.

* l.105: "estimated brightness of the squamate root was relatively high at 40%"

How is this information exploited ? The model used does not include any of the covariates that are later found to be important, so how reliable is this estimate ? See also potential concern about the model of evolution being used (below).

* l.127-137: all the supplementary figures seem to be issued from simple linear regression, where only one predictor is included on each analysis, and all other ignored (l.407-620 of the "Squamata_Evo_Analyses.Rmd" file). Could this be a problem in the interpretation of the result ? This means that the effect that is shown is the one that is *not* controlled for all other covariates. But we could imagine a situation where, if only predictor "x" is included, then it has a significant effect, but if both "x" and "z" are included, then "x" has no effect anymore ? Would it be possible, and maybe more informative, to use the predictions from the full model ? Or maybe I misread the code ?

* Fig.4: same remark.

* Fig.S6: could CIs be included ?

* Fig.3: There are $10 \times 9 = 90$ test represented here (and more were performed, as non-supported interactions were omitted). Did you correct for multiple testing ?

* l.176: "We found that following a long-equilibrated, static period, in the interface between the Miocene and Pliocene, the tempo of brightness evolution dramatically increased (Fig. 5a, c), paralleling the cooling and aridification period observed from fluctuations in $\delta^{18}O$ over the past 95 Myr31."

Visually comparing temporal curves is known to be misleading in some cases. I wonder how compelling this analysis is to back up the claim.

Maybe a model that includes a formal relationship between $\delta^{18}O$ and BM rate could be better suited here.

See e.g.

Clavel, J. & Morlon, H., 2017. Accelerated body size evolution during cold climatic periods in the Cenozoic. Proceedings of the National Academy of Science, 114(16): 4183-4188.

With function "fit_t_env" from package RPANDA.

As the relationship between climate change and evolution seems to be particularly important to the paper (see e.g. introduction "A changing climate has a critical impact..."), visual comparisons of curves might not be sufficient evidence.

* I.228: "Finally, to detect whether climate change through time has shaped the rate of evolution of color brightness and the retrieved result is not due to late burst patterns..."

I am not really convinced by this multiple regime analysis.

First, you test regimes defined by vegetation only, but maybe other covariables, defining other regimes, would lead to better fits ?

Second, in the code, it seems that you only test BM1, BMM and OUM, but not OU1, i.e. a single optimum OU. The preference for OUM could only be due to OU being better than BM (see comments on model selection below). Indeed, for the first family tested, Cordylidae, it looks like OU1 is better supported than OUM. (See COMMENT 3 in the attached script.)

I don't see how these analyses can exclude "late burst patterns".

* I.401-406:

- The fits actually use the package "geiger" ("fitContinuous") and not phytools.

- The fits are done without taking account any measurement errors, which are known to be important in PCM. See e.g. SILVESTRO, Daniele, KOSTIKOVA, Anna, LITSIOS, Glenn, PEARMAN, Peter B. y SALAMIN, Nicolas. Measurement errors should always be incorporated in phylogenetic comparative analysis. Münkemüller, Tamara. 2015, 340–346. (6). ISSN: 2041-210X.

Measurement error can be estimated e.g. with package "phylolm".

- The ancestral reconstruction is actually done with the BM, and not the best fitting OU model (function "fastAnc" ignores the "model" parameter, see COMMENT 2 in the attached script).

* I.408-432:

- Is there a reason to choose MCMCglm over a frequentist method, e.g. phylolm ?

- MCMCglm includes the phylogeny, but can only account for a BM (when using the "gaussian" family, if I'm not mistaken). But the previous analysis seemed to show that the OU (with measurement error) was better supported. Could using a BM structure in the regression lead to biased results ?

- I'm not familiar enough with MCMCglm to know which model exactly was fitted. Do you include any measurement error (i.e. individual non phylogenetic variation) in addition to the phylogenetically structured errors ?

- How did you choose for which interactions to include ? There might be a "multiple testing" issue here, as you fit many models in the script, with possible bias in the one you include (or exclude). In your script, I.218-385, 6 models with various interaction structures are fitted, before the dredge function is called. Would the dredge function on another model give different results ? Should the "richest" model be used first, before being dredged ?

- Why is polymorphism treated as random effect, and not a fixed one ? Could other effect be moved from "fixed" to "random" ? How did you choose ?

- Is there a reason for the specific threshold you chose (DIC > 5, wAIC >0.75) ?

Reviewer #3

(Remarks to the Author)

This is an excellent paper that makes use of a very large data set on integument brightness to capture how brightness has evolved in different habitats and climates in squamates. The dataset is indeed impressive, and the question is highly relevant as past adaptive change relative to environmental change might inform how organisms might respond to the current wave of climate change.

Overall, I very much enjoyed reading the paper and its scope and impact are well worth publishing.

My main comment is about terminology and expression – throughout the authors are not very careful with respect to the fact that they are reporting relationships and not causality. They also assign selective agency to environmental factors that are unlikely to be direct agents, rather they covary with the actual selective mechanism. This I think needs to be tightened up before acceptance.

Line 62: provide reference for this statement: 'known to influence'

Line 69 and onwards: I think there is an argument missing that transitions from correlation to causation. How do you know that vegetation brightness 'selects' for brighter integuments? I would argue that the selective agent is perhaps a predator or a physiological process (CT max) that reduces reproductive fitness and survival of darker individuals in these environments.

73: similarly to the above argument, latitude is not an environmental pressure (and I assume the authors are implying a selective pressure) – latitude summarises a whole raft of environmental variation across the globe. Many traits vary with latitude, and this gives us cause to look at the exact environmental variables that change with latitude and then investigate the biological mechanisms.

86: you did not articulate that expectation in the introduction.

101: I wonder if 'clustering' is the right word.. is it not 'correlate'?

118: here the authors have the terminology correct – it is an association and not a causation

Lien 199: here the terminology is not correct, it is a relationship, it is unclear what the actual cause is

132: same argument, 'open habitats' are not the selective agent. The expression in 130 would also work here; brighter integument was selected for in open habitats'

Fig 3 – this is a very helpful figure!

246: it would be helpful to mention the mechanisms behind these selective advantages

Discussion

The discussion is quite long and in parts repetitive (e.g. repeated mention that bright integuments reduce the absorption of irradiation). It could be shortened and focused on the 3-4 key results.

The Introduction nicely sets out how views of past adaptations can inform future responses to climate change, but the discussion does not strongly return to this concept.

314: again, it is not vegetation structure but the impact it has on temperature that is the selective force, as articulated in line 248

Version 1:

Reviewer comments:

Reviewer #1

(Remarks to the Author)

I have reviewed the revised version of the manuscript, as well as the response to my comments and those of the other reviewers, and find the article is now satisfactory to grant ready acceptance for publication. My comments have been thoroughly addressed and responded to in a clear manner, and I have no further comments to add. I am keen to see this excellent article added to the literature on the evolution of colour variation in squamates.

(Remarks on code availability)

I have reviewed the code and it runs smoothly in a standard R environment.

Reviewer #3

(Remarks to the Author)

I have read the revised version and the authors have addressed most of my concerns by tightening up their arguments and language. The manuscript is now in very good shape.

I only have three small comments for the authors to consider:

61-69: this is a single sentence and too long and complex. Can you please edit it into 2 or even 3 sentences.

74: I think the bright integument reflects incoming light rather than heat

76: my comment is again that latitude itself is not a selective pressure, the strength of selection might vary with latitude

(Remarks on code availability)

Reviewer #4

(Remarks to the Author)

This manuscript examines the evolution of brightness in squamates. The main conclusions are that habitat openness has been the primary driver of the evolution of color integuments and that this openness was influenced by aridification during the Miocene-Pliocene transition. I have not been involved in the earlier rounds of the review process and I'm not a squamates specialist. Overall, the manuscript is well-written, although I believe several details are missing from the

Materials and Methods section (as noted by one of the previous reviewers), such as whether the data were transformed, what the package implementations were, etc. However, I do have some concerns regarding the conclusions, which may be overstated or driven by the statistical analyses. These issues need to be addressed before recommending publication.

Scaling of the Variables:

It is not specified whether the predictors in the (Bayesian) regression were scaled, nor is it clear if the response variable (brightness) was log-transformed in both the regression and model fits. This is problematic because such transformations (e.g., using the logarithm of the studied trait) can affect model selection and the estimated parameters. For instance, in the regression analysis, if the predictors are not scaled, the interpretation of the estimated coefficients becomes difficult (note that the coefficients and their HPDs are not reported in the results or in the supplementary information – a summary table of these values, in addition to Figure 3, would help in interpreting the results and making comparisons in future studies). Furthermore, if the predictors were not scaled, the default priors used may bias the estimation of coefficients (and thus the relationships) depending on the scales of the predictors.

Relationship Between the Rate of Evolution and Past Climatic (Habitat Opening) Changes:

One of the main results of this paper is that changes in environments (habitat openness), as reflected in fluctuations in climate (foraminifera $\delta^{18}O$ curve), have driven brightness evolution. To support this, the authors reconstruct branch-specific rates of evolution using a method that models the evolution of the rates as a geometric Brownian motion (GBM) (i.e., the log of the rate evolves as BM), and then perform correlation analyses with the $\delta^{18}O$ curve. However, I believe that the observed pattern of increasing rates could be driven by two factors, and the conclusions may be overstated.

First, the best-fit model was an Ornstein-Uhlenbeck (OU) process. This suggests either there is substantial measurement error in the brightness color proxy (see below) or that trait evolution was somehow constrained (the alpha value is not small, as suggested in the responses to the review, because it depends on the depth of the tree. Instead, for a 202 million-year-old tree, an alpha of 0.01 means that the signal halves after ~69 million years). If the traits are measured with error or follow an OU process, the rates in the more recent (shorter) branches will appear higher than they are, either to compensate for variance contributed by the errors or to account for relative changes in a shorter period in a stationary process such as OU. Second, the GBM model used has a variance that scales with time, and as a result, the average of log-normally distributed rates might be biased toward increasing over time. Together, these effects can lead to a spurious increase in rates near the present.

In addition to the fact that spurious correlations can arise when time series are studied with improper tools, other periods of global cooling (e.g., the Eocene-Oligocene transition) do not show an increase in rates in Fig. 5. This suggests that either habitat openness was not a driver during these other periods, or that the observed pattern is due to potential biases, such as those discussed above.

As one of the previous reviewers advised, there is a model in the RPANDA package (Clavel & Morlon, 2017, PNAS) that explicitly models the rates of evolution as a function of an external (e.g., climatic) curve. This model also allows for the joint estimation of measurement error and can thus account for possible bias toward inferring higher rates near the present.

Measurement Errors in Model Fit:

It is not specified whether measurement errors (ME) were addressed in the analyses, or how they were handled. If no estimates of ME are available, it is generally possible to infer them jointly during model fitting (Silvestro et al., 2015, MEE). The "phyloilm", "Geiger", or "RPANDA" packages allow for the estimation of measurement errors. This is important because ME can influence model selection and parameter inference (see the comment above on the increase in rates near the tips).

Specific Comments:

- L. 79: Although it is not incorrect per se, I think "fractionation" would be a more precise term ("a measure of fractionation of ^{18}O to ^{16}O ") as it describes the process of changing the relative isotopic abundances.
- L. 226-227: It is not valid to compare differences in the DIC criterion (or likelihoods in general) that were computed on different datasets (e.g., with and without fossorial species) and use these differences as compelling evidence for higher support. If the predictors were standardized (see comment above), then the absolute values of the coefficients could be interpreted as effect sizes.
- L. 446-449: This is a misleading way of interpreting AIC differences, because for some nested models, the maximum possible AIC "delta" is much lower than these ranges. For instance, the support of the BM model over the EB or OU models cannot reach a delta AIC of >2 if BM is the true process and there is sufficient statistical power. I recommend that either the best model be used in downstream analyses, or that only the few best models be compared. While such guidelines are often used, they may be misleading for future studies that follow your protocol.
- L. 458-486: The citation and statement are incorrect. Most models of trait evolution (as opposed to diversification models, which the citation refers to) are based on the covariances between species that depend on their shared evolutionary history. Limited phylogenetic sampling—if sample size is sufficient—will not bias the results as it would for diversification models.
- L. 488-491: Please cite the package implementations used for fitting the various models.
- L. 505: "using the 'rescale' function".
- Fig. 2: Specify how ancestral states were reconstructed and which model was used.
- Fig. 4: Are these the marginal means? Were they computed after accounting for the covariates?
- Fig. 5: The figure shows confidence intervals (CIs) that are negative for the evolutionary rate, which cannot be correct, as

rates cannot be negative. Proper CIs must be calculated (e.g., 95% percentile of bootstrapped datasets), but see the comment above regarding the methods used to assess the link between rates of evolution and the d18O curve.

(Remarks on code availability)

Version 2:

Reviewer comments:

Reviewer #4

(Remarks to the Author)

I have read the revised version, and the authors have addressed some of my concerns, but not all of them. I am still not convinced by the claim that the rate of evolution follows climatic changes (or habitat opening). At the very least, I think the current analyses are not conclusive, as they fail to rule out potential biases.

First, the authors' argument that support for a multi-optimum OU model is consistent with constraints rather than ME is insufficient. This conclusion would only be valid if ME was incorporated into the analysis. Furthermore, support for the OU (or OUM) model also suggests that increased rates near the present might be spurious. An OU process is stationary, meaning that closely related species could exhibit as much divergence as more distantly related ones (i.e., all species are constrained to evolve around an optimum). If rates are estimated from data generated under such a process, an apparent increase in rates might simply reflect the need for more changes to explain similar levels of divergence in more recently diverged species, not a true signal of rate increase, especially if the data is estimated with some uncertainty.

Second, in my previous reviews, I suggested that, in the absence of uncertainty estimates for brightness, the effect of ME on the observed increase in rates could be assessed by modeling it as a nuisance term (as in Silvestro et al. 2015). The authors explain that such a term was used for the regression analysis, but they do not provide evidence that this effect was considered in the analysis of evolutionary rates. I demonstrate with a simple code (see below) that I can replicate the pattern of increasing rates at the tips, similar to the one found by the authors—including the last figures shown in the response—by modeling a process with a constant rate and adding random noise at the tips (i.e., ME). This noise affects the estimation of rates in both RPANDA and Phytools. When estimating ME in RPANDA, this effect can be partly compensated for, although more simulations would be needed to assess the uncertainty.

Finally, the authors state that “the observed lack of increased rates during periods such as the Eocene-Oligocene transition in Fig. 5 can be explained by the evolutionary timing and diversification of the examined squamates.” However, the increase in temperature during the late-Oligocene/early-Miocene began around ~25 Ma. Most of the crown ages of the clades considered date from >40 Ma (and >80 Ma for some). It seems unjustified, and not clearly discussed, to rule out the possibility that habitat opening could have left a significant imprint before this time (i.e. colder periods such as the Oligocene ~34-23Ma). Several studies have evidenced the possibility to detect early increases in rates.

In summary, without further evidence showing that the observed pattern cannot be explained by simple biases, I would remain cautious about the conclusions regarding the role of habitat opening in promoting the evolution of color brightness.

```
library(RPANDA)
library(phytools)
set.seed(1)
data(Cetacea)
data(lnfTemp)
tree <- pbtree(n=200, scale=40) # ~ Viperidae sample size
trait0 <- rTraitCont(tree, sigma = 0.1) # original data simulated under BM
errors <- mnorm(Ntip(tree)) # vector of errors
trait1 <- trait0 + errors*sqrt(0.01) # higher error
trait2 <- trait0 + errors*sqrt(0.05) # lower error

# fit the model to contaminated data
noisy1 <- fit_t_env(tree, trait2, env_data=lnfTemp, scale=TRUE)
plot(noisy1, lty=2)

noisy2 <- fit_t_env(tree, trait1, env_data=lnfTemp, scale=TRUE)
lines(noisy2, lty=1)

# original BM data
original <- fit_t_env(tree, trait0, env_data=lnfTemp, scale=TRUE)
lines(original, col="red")

# try to estimate the ME? Not perfect but still better...
```

```
reconME1<-fit_t_env(tree, trait1, env_data=lnfTemp, scale=TRUE, error=NA)
lines(reconME1, lty=1, col="green")
```

```
reconME2<-fit_t_env(tree, trait2, env_data=lnfTemp, scale=TRUE, error=NA)
lines(reconME2, lty=2, col="green")
```

```
legend("topleft", legend=c("Original data (true constant rate)", "noisy data ~0.01 SD", "noisy data ~0.05 SD",
"reconstructed with estimated error for noisy data (0.01 SD)",
"reconstructed with estimated error for noisy data (0.05 SD)"),
lty=c(1,1,2,1,2), col=c("red", "black", "black", "green", "green"))
```

```
# apparent rate increase when data are simulated with an OU process
traitOU1 <- rTraitCont(tree, sigma = 0.15, model="OU", alpha = 0.02)
traitOU2 <- rTraitCont(tree, sigma = 0.15, model="OU", alpha = 0.05)
```

```
# original BM data
dataou1<-fit_t_env(tree, traitOU1, env_data=lnfTemp, scale=TRUE)
dataou2<-fit_t_env(tree, traitOU2, env_data=lnfTemp, scale=TRUE)
```

```
plot(dataou1, col="purple", ylim=c(0.001,0.022))
lines(dataou2, lty=2, col="purple")
lines(original)
```

```
legend("topleft", legend=c("Original data (BM constant rate)", "Data generated under OU"),
lty=c(1,1), col=c("black", "purple"))
```

```
# Estimate rates with phytools
mult_phytools <- multirateBM(tree, trait1, lambda=1)
plot(mult_phytools$sig2~node.depth(tree), xlab="node depth", ylab="estimated rate")
```

(Remarks on code availability)

Version 3:

Reviewer comments:

Reviewer #4

(Remarks to the Author)

I have reviewed the revised version and the authors' responses to my previous comments. I appreciate the authors' efforts to address the issues raised in the previous submission. However, I believe the additional simulations do not resolve the problem (although it improves a bit on the previous version); instead, they illustrate the difficulty of ruling out the possibility that the observed association with recent climate cooling is spurious.

The additional simulations (Fig. 5, S35) involved adding noise to the empirical data. This approach means that any bias in rate estimates due to noise or measurement error (ME) in the empirical data may be exaggerated in the simulated datasets. Indeed, as shown in Figure S35, the rates estimated toward the tips from data contaminated with extreme noise levels are higher than those estimated from the raw data. This observation supports the comments from the previous review that noise could generate the observed pattern of rate increases near the tips. However, this does not provide evidence that the rates estimated from the empirical data are not influenced by such noise. The simulations in Figure S35 demonstrate that with higher noise levels, the rate estimates near the present are considerably higher than those for data simulated without or with low levels of noise, even after attempting to control for noise by estimating an ME term. While estimating the ME does tend to reduce the effect of noise, it also reveals that there remains a bias that could spuriously support a climate effect. In other words, without such a bias, we would expect the curves in the right panel of the figure to overlap.

Given that the only climate association proposed is the recent global cooling starting from the late Miocene, and considering that noisy data can produce a similar pattern, along with the difficulty of controlling for such noise, I maintain that the current results are likely artefactual – or at least a spurious association cannot be ruled out. A more robust set of simulations and analyses aimed at reducing or controlling for the detected biases would be necessary to convincingly demonstrate an effect of recent climatic changes on the rate of evolution.

(Remarks on code availability)

Response to reviewers to manuscript NCOMMS-23-56634

Former title: “Vegetation structure shapes color evolution of squamates over deep time”

Revised title: "Habitat openness shapes color evolution of squamates over deep time”

by Jonathan Goldenberg, Karen Bisschop, Joshua W. Lambert, Michaël P.J. Nicolai, Rampal S. Etienne, Liliana D’Alba, Matthew D. Shawkey

Overview

List of terms used in this document:

Original article: the submitted original article

Revised article: the revised article incorporating the revisions below

We thank the reviewers for their insightful comments and appreciation. We have thoroughly addressed all of the concerns. We initially address the major concerns by outlining how we incorporated the major comments. We then systematically respond to each specific comment provided. All modifications in the revised article are indicated in red for easy reference.

General major changes

Dear Reviewers, we incorporated and addressed all your comments and we specifically placed special emphasis on clarifying our methods and analyses. For example, in the revised article we undertook additional analyses to evaluate which evolutionary model fits best our data and we addressed statistical approaches as suggested by Reviewer #2. Moreover, we clarified terminologies and our rationale as suggested by Reviewers #1 and #3. We also made text and figure adjustments to improve clarity. One important text adjustment relates to changing “vegetation structure” to “habitat openness” as suggested by Reviewer #1. Notably, all these changes/additions strengthened our results without altering their outcome. We trust that by addressing these points and better highlighting how our hypotheses align with the existing literature we have effectively addressed any concerns.

Reviewer #1

Comment: Key results – The manuscript presents a compelling and novel results on the evolution of colour brightness variation in global squamates. The main result, that brightness correlates closely with environmental variation, both spatially, temporally, and across clades, provides cogent answers to a relevant and hitherto mostly unexplored biological question. First, the evolution of these brightness features is visualised across different biomes, showing a clear prevalence of brighter colourations in open habitats, but not clearly across wide latitudinal gradients. Second, it is mapped across a phylogenetic tree to show the phylogenetic clustering and historical trends across the evolution of squamates. Further analyses evaluate the relations between brightness and a series of habitat variables, finding that vegetation structure is the strongest predictor of colour brightness evolution, as also shown previously in the geographical and phylogenetic visualisations. Finally, the manuscript reconstructs brightness evolution through time in a sample of squamate clades, showing how it tracks together with variations in climatic conditions as predicted by the variations in oxygen isotopes, which the authors take to represent an evolutionary response to variations in vegetation structure.

Response: Thank you for evaluating our work. We are happy you found our study compelling and novel.

Comment: Validity – I believe that the study takes on an interesting biological question and does a good job at addressing it from a variety of different angles. I particularly appreciated how phylogenetic, ecological and spatial patterns are integrated in a compelling interpretation of the history of colour brightness in squamates. However, I do have some concerns with the interpretation of some of the results, due to the insufficient justification for the use of your proxy for vegetation structure. I do not take issue with the results themselves, here, but with the choice to use (and here I quote your own text) ‘variations in foraminiferal $\delta^{18}\text{O}$ as a proxy of vegetation structure change over time’. I went and checked the article you used as a reference, Herbert et al. (2016). While the article does say that there were changes in the vegetational compositions in different periods due to shifting climates, there is little that explicitly addresses vegetation structure itself beyond mentioning that there was a transition between wet and dry climate plants. In my opinion, here there is a leap in logic that should at least be better expanded upon. These estimates are based on variations of sea temperatures and levels through time, which in turn are estimated based on variations in oxygen isotope concentrations. It just does not seem to be strong enough of an association to be used as a proxy of vegetation structure, not without a clearer and stronger supporting explanation. To be clear, I am not saying that I think these interpretations are invalid, just that they should be better explained to the reader across the text. Additionally, I do have some concerns with the sampling strategy used in the article (see Data & Methodology section).

Response: We understand your concerns. To address this, we have expanded the rationale behind it. In L78-90 of the revised text we write: “Moreover, brightness evolution rates track foraminiferal $\delta^{18}\text{O}$ fluctuations, a measure of the relative abundance of ^{18}O to ^{16}O and an indicator of temperature change over time²⁶, and dramatically increased between the Miocene and Pliocene, presumably due to the establishment of a cooler and drier climate which promoted landscape shifts²⁷, ultimately inducing species to adapt to new habitats. Because $\delta^{18}\text{O}$ fluctuations are indicators of past temperature variations, their correlation with brightness evolution rates suggests that shifts in habitat openness, driven by temperature changes²⁷, played a significant role in shaping the evolution of dorsal brightness. Thus, while habitat openness itself does not directly select for brightness, it promotes environmental conditions that favor brighter integuments, possibly through the physiological stress and predation risks in open environments. This finding underscores the utility of foraminiferal $\delta^{18}\text{O}$ as a proxy for habitat openness shifts and its relevance for understanding evolutionary dynamics in response to environmental change over broad temporal and spatial scales.”. Moreover, by following your suggestion of using “habitat openness” instead of “vegetation structure” makes the logic stronger.

Comment: Significance – This work represents a valuable scientific contribution to a relatively unexplored field. Specifically, it is significant because a) it addresses a series of biological questions that have only been taken into scarce consideration by the existing literature, and b) it does so in Squamates, a study system which is well-suited for this type of study, having an impressive taxonomic diversity, global distribution, and a long and complex evolutionary history. The way these questions are addressed uses well-established analytical approaches which could allow their extension to broader taxonomic categories. Thus, I believe that this article will be a valid addition to the existing literature on the topic.

Response: Thank you again for endorsing our work.

Comment: Data & methodology – The data used in this article appears to be generally well justified in both its acquisition and its uses, with some exceptions that need addressing. In particular, I have some issues with the naming of certain variables, and their representation within the article. Additionally, I am adding some notes and observations related to data collection and methodology. Below, I will list the things that, in my opinion, would benefit from your consideration for the improvement of this work:

-vegetation structure: the name of this variable is not entirely appropriate to indicate the categories that you use in your work, specifically in the light of one of those categories, 'Subterrane'. What does living underground have specifically to do with vegetation? A lizard can be fossorial in a forest as well as in an open plains. I believe that these categories could be better framed as 'habitat openness', with subterrane being the least open of all categories. Additionally, the OpenClosed category would probably be more understandable if it were named simply 'semi-open'.

Response: We agree and changed throughout the text from "vegetation structure" to "habitat openness" and "openclosed" to "semi-open"

Comments: -Related to the previous point, on a more methodological note, while you did run analyses excluding fossorial (subterrane) species, I wonder whether you considered examining the effect that fossoriality specifically has on brightness variation (against all other categories). Fossorial species are very interesting in that they probably do not conform to the same rules as other species (i.e., thermoregulation is less likely to affect their colouration), and their numerical representation across squamate clades is very large. Writing a bit more on that in your discussion would be beneficial to contextualising your findings.

Response: Very interesting point. We have added the following in the discussion, L326-331:

"Moreover, we also performed analyses without accounting for fossorial species because conditions below ground differ significantly from those above, potentially influencing coloration through selection pressures such as production costs⁶³, antimicrobial⁶⁴ and heavy metal⁶⁵ protection. Given their numerous abundance across squamates²⁴, focusing on the selection pressures driving color variation of fossorial species, such as many skinks⁶², will be an exciting avenue for future research."

Comment: -I noticed that you included a coding of the activity patterns of lizards in your analyses, and that in the results, you mentioned that diurnal species are brighter than the others. It honestly surprised me that not much was included about this in the discussion, as it seems like a very relevant result. For example, the relationship between lifestyle and activity patterns has been discussed by recent papers (i.e., Slavenko et al., 2022), and it would be great if this type of analysis were to be contextualised in the light of colour evolution.

Response: Following the statistical considerations raised by Reviewer #2, the tendency of diurnal species for being brighter than nocturnal ones diminished. This is most likely due to other covariates being included in the analyses and the variation between different modalities of activity periods is diminished. We expanded on this in the discussion using the suggested citation (L322-326):

"Similarly, although diurnal and nocturnal habits are linked to various morphological and environmental adaptations⁶¹, our results indicate that both diel activities shape color brightness in similar ways. Instead, other factors such as habitat openness and other examined environmental conditions likely have a more pronounced impact on the evolution of dorsal brightness in squamates."

Comment: -Taxon sampling: While you do provide a good justification for the numerical sampling of taxa in your analyses (that is, what proportion of each clade was considered sufficient), the taxonomic (and, relatedly, geographic) diversity within each of the clades included in your work is not very well represented. While the representation within genera is quite good, there are very few genera per clade included. Additionally, there is a disproportionate numerical representation of some groups against others. This also means that many clades were essentially completely excluded from your analyses, biasing the results towards very few select genera. I am aware that taxon sampling is a difficult thing to do (and also limited by the information available), and sampling philosophy can be subjective, but I would have at least expected to see a better representation of genera across the phylogeny of each group. Using an example from your own data (on skinks because I am more familiar with this group), when more than one-fourth (33/116) of your sample of skinks belongs to the genus *Plestiodon*, and about one-fifth (21/116) by the genus *Chalcides*, how much are your

results really saying about skinks in general? How much about Plestiodon and Chalcides? There are 168 skink genera, only 25 of which are included in your dataset, with most of the dataset being represented by species belonging to five or six of them. In my opinion, this is a methodological weakness that should at least be acknowledged and addressed within the article's methods and discussion.

Response: Rather than methodological weakness we see this as a methodological limitation. As the main signature of the study, that brighter species are favored in open habitats rather than closed habitats, is consistent across different family and clades, and against phylogenetic structures, we are confident that the limited sampling does not significantly influence the outcome. Specifically for skinks, we acknowledge the low representation. However, as our sampling effort aimed to capture broad temporal scale patterns we focused on collecting data across major groups. As an example, Chalcides and Plestiodon share a common ancestor around 70 Ma (Zheng and Wiens 2016), and considering the crown Scincidae arose 97 Ma, we trust we capture a broad ecological history even by analyzing these two genera (besides the other 23). We reiterate that our aim is to provide evidence for broad scale patterns and our Figure 1 shows that we sampled across all biogeographical realms. We fully agree that further investigations are warranted to underpin specific trends within and across genera of the same clades. We added such considerations in methods, L379-381: **“Our aim in this study was to provide evidence for broad scale patterns. As such, further investigations are needed to underpin specific trends within and across genera of the same clades.”**.

Comment: Analytical approach – The study takes a valid and adequate analytical approach to its analyses, both statistically and in terms of data interpretation. While I would in some cases use different approaches to the same statistical/phylogenetic problems, I do not find any particular issue with any aspect of the analyses. I particularly appreciated the model validation included in the text. It does show how much careful consideration went into this article, to ensure the validity of analyses and evaluate their potential pitfalls. The phylogenetic analyses employed here appear to be sound, but I would like to see some more information about the alternative models of character evolution that you tested, and why you decided to use an OU early-burst model specifically.

Response: The choice was statistically supported and we report this in the results, L110-112: “... under an Ornstein-Uhlenbeck (OU) model (AIC: Brownian motion (BM) = 8756.22; OU = 8424.152; Early Burst (EB) = 8435.44)...”. As the OU model of evolution had a significantly lower AIC, OU was the best supported model for character evolution.

Comment: Clarity and context – This article is relatively well contextualised within the existing literature, but it could use some major improvements in its clarity and stylistic choices. I will proceed to comment on each section in order:

Introduction: mostly well contextualised and clear, it sets the theoretical stage for the analyses that follow. I would have liked to see the questions/hypotheses or at least an aim of the paper to be more clearly spelled-out. And I am unsure whether you should summarise the results at the end of the introduction (that should be the abstract's role, but feel free to ignore this if that is in the journal's style). Instead, you should write what you did to address your hypotheses.

Response: While in our revised article we indeed kept the summary of the results at the end of the introduction following the journal style, we expanded on the aim of the study and in line L61-64 we write: **“Here, to gain understanding of the mechanisms driving their color brightness evolution and to specifically evaluate whether selected drivers are consistent across taxa despite the opposing predictions found in the literature¹⁴, we compiled a new global dataset of squamates' life-history traits and ecological information...”**.

Comment: Results: a bit unclear, as it reads more like a mix of methods and results, which could confuse a reader trying to parse the text. This also means that there is quite a lot of redundancy

between materials/methods and the results. This redundancy impacts clarity. The results section should be also better sectioned in smaller paragraphs to improve clarity.

Response: We worked on improving the clarity of this section following your suggestions below.

Comment: Discussion: The discussion is generally well written, presents coherent arguments, and is clear in most of its parts. However, some of the results (i.e., the global relationship with activity pattern/circadian rhythm) are incompletely or not at all discussed, as the article chooses to selectively focus on some of the results instead of others. Additionally, the part of the discussion (lines 295-306) that focuses on the observed interaction between body size and altitude is very opaque, and its logic is not very clear. I read the paragraph several times, and I am not really sure of what are you trying to say with it. If your patterns are not at all consistent across taxonomic groups, perhaps it does not make much sense to try to explain them as a whole.

Response: To improve the logic discussing body size, we clarify in L333-336: “We found that brightness of species in families with relatively small body sizes is also negatively influenced by altitude (Scincidae) and positively (Scincidae and Cordylidae) and negatively (Tropiduridae) by body mass, suggesting contrasting selection pressures **within these small-bodied groups compared to larger organisms.**” This clarification makes clearer that smaller organisms are subjected to different selective pressure than larger ones and we develop further on this in the following sentences. As for activity pattern, we addressed this in our previous answer.

Comment: Materials/methods: No particular comment on this part, except what already stated in other sections of this report.

Response: thank you – addressed.

Comment: Figures: See specific comments below.

Response: We worked on these suggestions – see below.

Comment: Suggested improvements – For general suggestions about the article, see each of the sections above. Below, I have listed some specific suggestions for parts of the text which require improvement.

Response: Thank you for these – we carefully addressed them.

Comment: Line 75: As this is the first time you mention the ratio of stable isotopes of oxygen, you should define it briefly.

Response: Addressed - L78-79 now reads: “Moreover, brightness evolution rates track foraminiferal $\delta^{18}\text{O}$ fluctuations, **a measure of the relative abundance of ^{18}O to ^{16}O** and...”

Comment: Lines 86-88: Beyond the expected relationship between brightness and vegetation structure, it’s really interesting that you found brightness not to correlate with latitudinal variation. Any reason why you think that may be the case?

Response: We touch on this topic in the discussion when for example we state “Such contrasting signatures at family or higher taxonomic level decrease the likelihood of identifying unified mechanisms regulating global patterns of color variation across a vast phylogenetic group using traditional ecogeographic rules¹⁴. In fact, in contrast to our expectations, we did not detect any consistent effect of brightness variation for any sampled eco-environmental parameter besides **habitat openness.**” Specifically for latitude, it is a similar outcome to another study from Slavenko et al. (2019, Glob. Ecol. Biogeogr.) where the authors showed that body size evolution is not driven by consistent spatial patterns.

Comment: Fig. 2: It is a bit hard to visualise the brightness scale on the ancestral reconstruction image. Perhaps a different (starker) scheme would be appropriate. The labelling of the different

clades on the tree could be graphically accomplished better. Also, the separation between the clades is a bit hard to see. One possible suggestion is using alternating, thicker bands of two different colours above each of the clades.

Response: Thank you for these stylistic suggestions! Following yours and Reviewer #2 comment, we consistently used a starker palette for color brightness. See below the revised image.

Comment: Line 123: At least 50% of the focal species? What do you exactly mean by that? What are these ‘focal species’?

Response: We removed “focal” for clarity. At least 50% is to ensure “sufficient statistical power and to avoid inferring spurious patterns”.

Comment: Line 120: ‘using phylogenetic comparative methods’ is a bit vague. Please say which specific operation you performed. Additionally, I am unsure whether this part of this section should go into the methods instead. Mentioning the operations that you performed outside of the methods is fine, but I think that the choices of which groups were analysed and why definitely belongs to the methods and not the results.

Response: We now specified (L133): “by using phylogenetic comparative methods under a Bayesian framework”. All details related to the specific operations are provided in the methods. The journal follows the introduction, results, discussion, and methods layout. To help a reader following the narrative we included minimal background information in the results. As this is a style choice, we kept the original structure of this paragraph on this occasion.

Comment: 125-126: So, you lost more than 70 species due to reclassifications? That sounds like a lot of species lost only due to reclassification. I would have expected a loss due to species being absent from the phylogeny, not due to reclassification. If possible, add example citations on the greater reclassifications efforts that have caused this taxonomic reduction.

Response: In the supplementary material we provide full access to our database with all species listed according to Zheng and Wiens and latest Reptile Database classification. As our sampling started from the phylogeny of Zheng and Wiens (L372-374), species cannot be absent from the phylogeny.

Comment: Fig. 3: It's not a bad figure per se, but perhaps I would advocate removing the icons at the bottom of the image, or at least placing them at the top next to the relevant variables. They are a bit confusing in the context of the figure, otherwise. If space is an issue, you could try switching the X and the Y axis of the figure, making this operation easier.

Response: We tried to place the icons on top of the figure but it was then highly confusing. As also Reviewer #3 found this "a very helpful figure!" we kept it in its original form.

Comment: 150: Why 'surprisingly'?

Response: Because differently from other families they did not show any association with any of the examined variables but habitat openness, which is an unexpected outcome.

Comment: Line 165 (caption): Empty cells denote a lack of significant correlation.

Response: Added!

Comment: Fig. 4: This figure has some issues and would benefit from being made smaller and more informative. For example, the confidence intervals of the different categories could be illustrated a bit better. I also imagine that there will be some outliers, which are not shown here but would be really easy to display (depriving a reader of that information would diminish the informative value of the figure). A basic ggplot R script with `geom_boxplot` would fix these issues. In that case, I recommend keeping the colour exclusively in the confidence boxes. Moreover, there is an issue with the similarity between the white-to-black dorsal brightness bar on the Y axis, and an identical bar on the X axis that is used to indicate vegetation structure. As the categories are not explicitly continuous, it is questionable to use the variation in a continuous colour bar to indicate them. I also am unsure whether OpenClosed (an unfortunate category name – again, wouldn't 'semi-open' be better?) and especially Subterranean really represent vegetation, at least in the way they are displayed in the images. Is being in a subterranean cave 'vegetation' structure? Looks to me like this category is not about vegetation at all, but about habitat structure or type. See previous comments about this point.

Response: Following yours and the Reviewer #2 statistical considerations we completely changed the visualization of this. We incorporated your suggestions of category name changes as indicated above.

Comment: Line 168: 95% confidence intervals.

Response: As we work in a Bayesian framework, these are credible, not confidence intervals.

Comment: Line 173: Again, it is unclear to me what do you exactly mean by 'focal species'.

Response: Similarly, we removed "focal" from the revised version.

Comment: Line 179: Reiterating the point from the general evaluation of the article, I am unconvinced that the variation in O2 isotopes means what you take it to mean, or at least, that the evidence you present is very solid. Yes, overall vegetation structure saw increases or decreases in certain periods, but it seems to me like an oversimplification of the real trends. Could any control factor be introduced or at least considered for latitude or (reconstructed) biomes?

Response: In line with a similar comment from Reviewer #2, we rephrased the paragraph clarifying that the observed trends are statistically supported (Fig. S17, S18). To do so we present the statistical result at the beginning of paragraph (L189-192). As our comparative analyses showed a strong,

consistent effect only for habitat openness, we specifically focused on the evolution rates of color brightness in different habitat types versus $\delta^{18}\text{O}$ rates.

Comment: Lines 204-210: This is, again, something that would be more suitable as a part of the methods section.

Response: Similarly to our previous answer, we prefer to keep the current structure to help readers follow the narrative.

Comment: Line 213: It is a good thing that you checked for phylogenetic bias in these two groups, but at that point you might as well extend these analyses to all the groups. The fact that the models for those two groups are not affected by phylogeny does not imply that all the other groups will be.

Response: We could not perform such additional analyses because the phylogeny included in Zheng and Wiens relies on the latest ones available for each examined group. For example, we checked the supplementary materials and code of the suggested work of Slavenko et al. (2022) on diel activity patterns of skinks. The phylogeny used is the one of Zheng and Wiens pruned to the examined taxa. Given that we analyzed 17 different models with different phylogenetic structures and compositions, we trust the results are robust.

Comment: Line 203: The Model Evaluation section lists a series of supplementary analyses that you performed to evaluate how robust your models are, which is a laudable effort, but I believe that its structure necessitates improvement. First, the choice to keep this section in a single paragraph makes it hard to keep track of the text. It needs to be rewritten in a more focused way that separates in at least a couple of distinct paragraphs. Additionally, I am unsure whether parts of this section wouldn't be better included in the methods. In my view, and I will leave it to the editors to weigh in on this matter, the best way to make this part work would be to include the evaluation in the methods sequentially after each original version of the model and keeping the same sequence in the results. For example, in the methods I would write something like: 'to assess the evolution of dorsal brightness in squamates, we did X. Then, to validate it, we did Y for these groups and these other groups etc...', and in the results 'The result of X was x, which was validated by doing Y.'

Response: We agree the structuring could have been better presented. In the revised article we introduce subparagraphs which help readers to logically follow the model evaluations' section. We also reiterate our reasonings for keeping all the information in this paragraph, especially because we see all of these as results and not methods.

Comment: Lines 401-404: Model comparison should not be included in the methods but in the results. Additionally, it would be best practice to include (in the methods) a minimum threshold for model selection (under which two compared models are considered to perform not significantly better than the other: for an example, see Bergmann & Morinaga, 2019).

Response: We moved the model comparisons to results (L110-112). We also clarify in methods, L446-449, the thresholds for model selection: "We compared evolutionary models using the widely established difference in AIC values, following the thresholds $\Delta\text{AIC} \leq 2$ (models equally supported), $2 < \Delta\text{AIC} < 10$ (considerable support for model with lower AIC), $\Delta\text{AIC} \geq 10$ (significant support for lower AIC model).".

Comment: Lines 404-406: If you found clustering of brightness where biomes select for different colours, why was this not explicitly tested statistically?

Response: Our aim here was to visually identify spatial variation of color brightness across the world. Afterwards we specifically analyze the association with the different habitat types. As Whittaker's biomes classification broadly organizes ecosystems based on temperature and precipitation patterns, leading to different habitat types, we formally analyzed habitat type because it is more biologically relevant at the species level.

Comment: Lines 415-419: I am admittedly a bit unfamiliar with this modelling approach, but why did you account only for interactions between two of the fixed effects, instead of all or more combinations of them, at least as a preliminary step in your model construction?

Response: We performed multiple model selections which included all possible interactions. We write in L460-463: “In all cases, we set brightness as response variable, and habitat openness, altitude, latitude, body mass, circadian rhythm and their interactions as fixed effects.” This indicates that all possible interactions were accounted for.

Comment: Lines 407-432: Sometimes it is unclear which functions are associated with which packages.

Response: We used >20 different packages for our analyses. We specified the crucial ones and we refer to our supplementary code for further information. Given that Reviewer #2 stated “the reader can easily find information on which exact methods were used in the analyses”, we kept the original version.

Comment: Lines 447-463: Here I would like to put the accent on the unnecessary redundancy between this part of the methods and the results section. As I wrote above, this section, and not the results, is where these methodological notes should belong.

Response: Unless the editor specifically requests it, we will maintain the current formatting. We hope it is clear that we did not overlook your suggestion; rather, we made a stylistic and narrative choice to help readers follow the narrative more easily.

Comment: Lines 295-298: I think that this part of the discussion should be explained more clearly. If the trends are all over the place depending on the clade, I do not see how you can draw a consistent unitary explanation from them. To me it reads like a circular argument.

Response: We think this is a very interesting finding. When performing family-level analyses we find that only smaller clades show associations with specific ecological variables, suggesting contrasting selection pressures within these small-bodied groups compared to larger organisms. As outlined above, we rephrased this sentence to clarify our point.

Comment: Lines 308-312: Maybe add some citations to this information. Also, the double parentheses in the last sentences make reading difficult. Perhaps break the sentence in two.

Response: Done – we have added the references to Mark O’Shea’s books. We also followed your suggestion and broke the sentence in two.

Comment: Reviewer recommendation: The article presents valuable findings that provide novel insights on a relatively unexplored topic. However, it is my opinion that significant changes and clarifications are required to grant its acceptance. If the relevant alterations are made, I would be happy to see this paper published in the journal.

Response: Thank you once again for your time and providing valuable feedback to improve our work!

Reviewer #2

General Comments

Comment: This paper analyses a large dataset of 1249 squamates, and is particularly interested in the link between brightness evolution and environmental covariables, such as habitat, altitude, and vegetation structure, which is argued to be the single most important driver of brightness evolution. The paper is well written and well structured, and of possible broad interest to the community.

I do not have any background in squamate evolution, so I will not discuss the biological conclusion, but rather focus on the statistical analyses carried in this work.

Response: Thank you very for assessing our study and providing valuable feedback on the statistical analyses.

Comment: The analyses that are carried on are well described, and the reader can easily find information on which exact methods were used in the analyses. I found the associated Rmd file particularly useful, and I could run part of it on my machine without any problem, which makes this work reproducible.

Response: Thank you – we have placed special emphasis on making the work reproducible.

Comment: The analyses use well documented statistical methods, and seem to be robust to a number of variations. However, as detailed below, I have a number of concern regarding the analyses, and their power to back up some of the claims of the paper. Potential concerns include : multiple testing, model selection, accounting for measurement error, lack of formal tests for some associations, and possible (not very consequential) code errors. As I'm not an expert in these organisms or kind of studies, some of the concerns I raise might be irrelevant, but I included them in case it might be useful.

Response: We carefully addressed your concerns and suggestions, and we incorporated them in our revised article.

Comment: Below are detailed some of my concerns. Attached is also a modified version of the "Squamata_Evo_Analyses.Rmd" file provided by the authors, with a few comments (marked as R chunk named "COMMENT").

Response: We think that there was a glitch in the system and we did not receive the modified Rmd file. Nevertheless, the comments made below were very clear and we were able to incorporate all your suggestions into the revised script. We also note that there was a typo in our dataset and one habitat entry was named "OpenClosed:" instead of "OpenClosed", which is now adjusted.

Comment: I thank the authors for their interesting and well-thought work, and the Associate Editor for trusting me with a review.

Response: Thank you again for the valuable feedback!

Detailed Comments

Comment: * I.85: "We found that brighter species are generally found in open habitats such as grasslands, deserts, and prairielands, and darker species in forested regions..."

This is just a visual finding, comparing the maps, isn't it ? Could a formal test, coming from the geostatistics literature, back up this claim ? I don't find the visual inspection very compelling. However formal tests using phylogenetic comparative methods are carried on afterward, so maybe this is enough.

Response: Our aim here was to visually identify spatial variation of color brightness across the world. Afterwards we specifically analyze the association with the different habitat types. Since Whittaker's biome classification broadly organizes ecosystems based on temperature and precipitation patterns, resulting in different habitat types, we chose to formally analyze habitat type as it is more biologically relevant at the species level.

Comment:* Fig.1: It is hard to tell in some regions if the squamates are very bright, or if white is just missing data ? (E.g. in Alaska and other Northern regions.) The contrast of the purple gradient is not very good in other regions, making the visual inspection difficult.

Response: Following a comment of Reviewer #1 we standardized the color brightness variation

palette throughout the article which provides starker colors, aiding for visual inspection. See below the revised figure.

Comment: * I.105: "estimated brightness of the squamate root was relatively high at 40%"
 How is this information exploited? The model used does not include any of the covariates that are later found to be important, so how reliable is this estimate? See also potential concern about the model of evolution being used (below).

Response: Following your comment below, we now used the 'reconstruct' function in "ape" to estimate the squamate root. As this is only an initial visual representation of how our variable of interest color brightness changed over time, here we did not consider covariates which are found later to be important. We adjusted the plotting and the analyses accordingly to your considerations (see below), including the model of evolution (to OU).

Comment: * I.127-137: all the supplementary figures seem to be issued from simple linear regression, where only one predictor is included on each analysis, and all other ignored (I.407-620 of the "Squamata_Evo_Analyses.Rmd" file). Could this be a problem in the interpretation of the result? This means that the effect that is shown is the one that is *not* controlled for all other covariates. But we could imagine a situation where, if only predictor "x" is included, then it has a significant effect, but if both "x" and "z" are included, then "x" has no effect anymore? Would it be possible, and maybe more informative, to use the predictions from the full model? Or maybe I misread the code?

Response: You are very correct – figures were plotted as a linear relationship between the significant predictors from the full model and the outcome variable (dorsal brightness). Our intention was to visually show the relationship between the two (or three) variables after we found evidence for their contribution. We understand your concern and all our figures have been updated to include the effects of all the variables. As many variables showed weak or no association with dorsal brightness, to avoid overfitting we built models including all significant predictors. Notably, main results did not change from the original article version.

Comment: * Fig.4: same remark.

Response: Adjusted! We also present the results in a clearer way by including the density of the posterior distributions besides the credible intervals and posterior means.

Comment: * Fig.S6: could CIs be included ?

Response: Yes, done.

Comment: * Fig.3: There are $10 \times 9 = 90$ test represented here (and more were performed, as non-supported interactions were omitted). Did you correct for multiple testing ?

Response: The model structures analyzed using MCMCglmm are independent, and the dredge function was applied separately within each model structure to evaluate different combinations of predictors. The dredge function facilitates model selection using DIC, which inherently penalizes model complexity. This approach does not involve multiple hypothesis testing per se but rather identifies the most parsimonious model within each structure. As such, our use of DIC-based selection ensures that we account for the number of predictors, thus mitigating concerns related to overfitting and model complexity. We believe this approach provides a robust framework for identifying the key predictors influencing the response variable.

Comment: * l.176: "We found that following a long-equilibrated, static period, in the interface between the Miocene and Pliocene, the tempo of brightness evolution dramatically increased (Fig. 5a, c), paralleling the cooling and aridification period observed from fluctuations in $\delta^{18}\text{O}$ over the past 95 Myr31."

Visually comparing temporal curves is known to be misleading in some cases. I wonder how compelling this analysis is to back up the claim.

Maybe a model that includes a formal relationship between $\delta^{18}\text{O}$ and BM rate could be better suited here.

See e.g.

Clavel, J. & Morlon, H., 2017. Accelerated body size evolution during cold climatic periods in the Cenozoic. *Proceedings of the National Academy of Science*, 114(16): 4183-4188.

With function "fit_t_env" from package RPANDA.

As the relationship between climate change and evolution seems to be particularly important to the paper (see e.g. introduction "A changing climate has a critical impact..."), visual comparisons of curves might not be sufficient evidence.

Response: We think there was a misunderstanding here due to our paragraph structure. We performed a formal relationship between σ^2 and $\delta^{18}\text{O}$ at nodes (Fig. S17). We clarified this in the revised text in L189-190: "We found significant positive correlations between $\delta^{18}\text{O}$ and rates of color brightness evolution (Fig. S17)", and moved the visual description afterwards.

Comment: * l.228: "Finally, to detect whether climate change through time has shaped the rate of evolution of color brightness and the retrieved result is not due to late burst patterns..."

I am not really convinced by this multiple regime analysis.

First, you test regimes defined by vegetation only, but maybe other covariables, defining other regimes, would lead to better fits ?

Second, in the code, it seems that you only test BM1, BMM and OUM, but not OU1, i.e. a single optimum OU. The preference for OUM could only be due to OU being better than BM (see comments on model selection below). Indeed, for the first family tested, Cordylidae, it looks like OU1 is better supported than OUM. (See COMMENT 3 in the attached script.)

I don't see how these analyses can exclude "late burst patterns".

Response: We removed "and the retrieved result is not due to late burst patterns" from the revised article to enhance clarity. Following the comparative analyses, we found that only habitat openness

(the revised term for vegetation structure) consistently shapes dorsal brightness. As different groups showed different associations with different predictors, we wanted to maintain consistent and comparable model structures. This allowed us to compare different groups in a consistent way, while focusing on our key predictor, habitat openness.

As for the code, you are correct once again. We naively missed to test OU1. Thank you for spotting this! We have now included OU1 in our tests and addressed this also in the main text. Specifically for cordylids, we obtained an AIC value for OU1=278.95, and OUM=279.35. As the difference (Δ) is far smaller than 2, we cannot conclude that OU1 is preferred over OUM, as both perform equally well.

Comment: * I.401-406:

- The fits actually use the package "geiger" ("fitContinuous") and not phytools.

- The fits are done without taking account any measurement errors, which are known to be important in PCM. See e.g.

SILVESTRO, Daniele, KOSTIKOVA, Anna, LITSIOS, Glenn, PEARMAN, Peter B. y SALAMIN, Nicolas.

Measurement errors should always be incorporated in phylogenetic comparative analysis.

Münkemüller, Tamara. 2015, 340–346. (6). ISSN: 2041-210X.

Measurement error can be estimated e.g. with package "phylolm".

Response: Thank you for addressing us to the relevant literature. We confirm that we have now fitted an OU model for continuous character evolution accounting for measurement error using "phylolm".

Comment: - The ancestral reconstruction is actually done with the BM, and not the best fitting OU model (function "fastAnc" ignores the "model" parameter, see COMMENT 2 in the attached script).

Response: What a glitch from us! We now used the 'reconstruct' function in "ape" setting the constraint towards the optimum (α) from the "phylolm" output and using the "GLS_OUS" model which assumes that the process starts from the optimum following Royer-Carenzi and Didier (2016)*. As the α value is low, it is not surprising that the estimates and the confidence intervals are similar to our previous estimations using a BM model.

*Royer-Carenzi, M. and Didier, G. (2016) A comparison of ancestral state reconstruction methods for quantitative characters. *Journal of Theoretical Biology*, 404, 126–142.

* I.408-432:

Comment: - Is there a reason to choose MCMCglm over a frequentist method, e.g. phylolm ?

Response: In our specific case study, we chose to use a Bayesian framework with MCMCglmm over a frequentist method like phylolm because it allowed us to interpret the results in terms of posterior distributions. This approach provides greater flexibility and adaptability, especially in accounting for uncertainty. While this choice can be seen as somewhat arbitrary, we believe the Bayesian framework offered advantages that were particularly beneficial for our analysis.

Comment: - MCMCglm includes the phylogeny, but can only account for a BM (when using the "gaussian" family, if I'm not mistaken). But the previous analysis seemed to show that the OU (with measurement error) was better supported. Could using a BM structure in the regression lead to biased results ?

Response: Even if certain models fit the data better in terms of statistical criteria, the presence of consistent evolutionary patterns across various models lends support to the robustness of our findings.

Nonetheless, we performed an additional sensitivity analysis by rescaling the tree under an OU model of evolution. To do so, we first compared the different models (BM, OU, EB) using 'phylolm' including all predictors. Then, after OU showed better support, we extracted the alpha value and rescaled the tree using 'ouTree'. The tree was then incorporated into the MCMCglmm. The result remained consistent where brighter species are more probable to occur in open rather than closed habitats (Table S2). Given the small alpha value (0.01) it is not surprising that the relationship did not

significantly change. Supported by these equal results between BM and OU models, and as we prioritized model structure for comparative purposes, we used the default BM model structure in our MCMCglmm analyses.

Comment: - I'm not familiar enough with MCMCglmm to know which model exactly was fitted. Do you include any measurement error (i.e. individual non phylogenetic variation) in addition to the phylogenetically structured errors ?

Response: As we did not have any a priori belief, we followed the default behavior of MCMCglmm, which has the variance structure in the form of $VG \otimes D$ where D is a diagonal matrix of measurement error variances and VG is a scalar fixed at 1.

Comment: - How did you choose for which interactions to include ? There might be a "multiple testing" issue here, as you fit many models in the script, with possible bias in the one you include (or exclude). In your script, l.218-385, 6 models with various interaction structures are fitted, before the dredge function is called. Would the dredge function on an other model give different results ? Should the "richest" model be used first, before being dredged ?

Response: We began with a set of biologically plausible predictors and considered all possible interactions that could be relevant based on our understanding of the system. This led to the formulation of 6 models, ranging from the most complex (4-way interactions) to the simplest (2-way interactions). To evaluate these models, we used the DIC, which allowed us to compare their performance. The model with the lowest DIC value was selected as the best-fitting model for our data. After selecting the best-fitting model based on DIC, we applied the dredge function to this model only. The purpose of the dredge function was to identify which predictors within this best-fitting model most strongly influence the response variable. By limiting the use of the dredge function to the best-fitting model, we minimized the risk of inflating type I error rates associated with multiple testing. This approach ensures that we are not arbitrarily fitting multiple models without a guiding criterion.

It is important to note that the initial step of comparing models using DIC is not the same as multiple hypothesis testing. Rather, it is a model selection process aimed at identifying the most parsimonious model that adequately describes the data. The dredge function's purpose is to refine our understanding of the selected model, not to perform multiple tests on multiple models. Therefore, our approach avoids the pitfalls of multiple testing by focusing on a single, well-supported model.

Comment: - Why is polymorphism treated as random effect, and not a fixed one ? Could other effect be moved from "fixed" to "random" ? How did you choose ?

Response: Differently from all other fixed effects, we don't have any specific biological prediction for polymorphism (why would a polymorphic species be brighter or darker?). As such we kept this variable as random effect to account for within species variation.

Comment: - Is there a reason for the specific threshold you chose ($DIC > 5$, $wAIC > 0.75$) ?

Response: The choice of $\Delta DIC < 5$ as a threshold for considering models to be substantially supported is based on established guidelines in model selection literature (similarly to ΔAIC and the cutoff at 2)*. As for $wAIC > 0.75$, this threshold is often used because it provides a good balance between being conservative enough to ensure that the predictors included are truly important and being inclusive enough to account for model uncertainty. A $wAIC$ of 0.75 implies that there is a 75% probability that the model (or predictor) is the best among the set of candidates, which is a robust level of confidence. The use of 0.75 is well-supported in the literature (Burnham & Anderson, 2002)* and aligns with common practices in model selection.

*Burnham, K. P., & Anderson, D. R. (Eds.). (2002). *Model selection and multimodel inference: a practical information-theoretic approach*. New York, NY: Springer New York.

Reviewer #3

Comment: This is an excellent paper that makes use of a very large data set on integument brightness to capture how brightness has evolved in different habitats and climates in squamates. The dataset is indeed impressive, and the question is highly relevant as past adaptive change relative to environmental change might inform how organisms might respond to the current wave of climate change.

Overall, I very much enjoyed reading the paper and its scope and impact are well worth publishing.

Response: Thank you for reviewing our study and for sharing your enthusiasm on the topic.

Comment: My main comment is about terminology and expression – throughout the authors are not very careful with respect to the fact that they are reporting relationships and not causality. They also assign selective agency to environmental factors that are unlikely to be direct agents, rather they covary with the actual selective mechanism. This I think needs to be tightened up before acceptance.

Response: We confirm that we have successfully incorporated and addressed all your suggestions/comments. See below.

Comment: Line 62: provide reference for this statement: ‘known to influence’

Response: Done! Provided relevant references.

Comment: Line 69 and onwards: I think there is an argument missing that transitions from correlation to causation. How do you know that vegetation brightness ‘selects’ for brighter integuments? I would argue that the selective agent is perhaps a predator or a physiological process (CT max) that reduces reproductive fitness and survival of darker individuals in these environments.

Response: We agree and rephrased the text, and used “...brighter integuments **are favored** in open habitats...” instead of “open habitats select for brighter integuments...”.

Comment: 73: similarly to the above argument, latitude is not an environmental pressure (and I assume the authors are implying a selective pressure) – latitude summarises a whole raft of environmental variation across the globe. Many traits vary with latitude, and this gives us cause to look at the exact environmental variables that change with latitude and then investigate the biological mechanisms.

Response: We changed as per suggestion to “selective pressures”. To clarify that latitude correlates with a suite of environmental variables, we clarified in L64-67 of the revised article: “...we compiled a new global dataset of squamates’ life-history traits and ecological information known to influence brightness^{18,26}, namely **habitat openness**, latitudinal and altitudinal distributions (**which correlate with environmental factors such as temperature, photoperiod, and UV exposure**)...”. We also included altitude here for the same reasons as latitude.

Comment: 86: you did not articulate that expectation in the introduction.

Response: We have added this expectation in L49-50 of the revised article.

Comment: 101: I wonder if ‘clustering’ is the right word.. is it not ‘correlate’?

Response: We considered it, but as this is a visual representation and not a formal test (which follows with the comparative methods), we kept the word clustering in the revised text.

Comment: 118: here the authors have the terminology correct – it is an association and not a causation

Response: Great.

Comment: Lien 199: here the terminology is not correct, it is a relationship, it is unclear what the actual cause is

Response: Here we meant, statistically speaking, a correlation analysis between two sets of variables (Fig.S17). That is why we specifically indicate “analyses” in the sentence. Relationship would not be appropriate in this case. In case you meant L119, we revised the sentence to (L132 of the revised text): “Next, we evaluated **the relationship between** ecological and environmental variables **and** the brightness of species...”.

Comment: 132: same argument, ‘open habitats’ are not the selective agent. The expression in 130 would also work here; brighter integument was selected for in open habitats’

Response: We revised the sentence accordingly to your suggestion but using “favored” instead of “selected”.

Comment: Fig 3 – this is a very helpful figure!

Response: Thank you!

Comment: 246: it would be helpful to mention the mechanisms behind these selective advantages

Response: We now explain that melanins’ properties are the underlying mechanisms behind these functions.

Discussion

Comment: The discussion is quite long and in parts repetitive (e.g. repeated mention that bright integuments reduce the absorption of irradiation). It could be shortened and focused on the 3-4 key results.

Response: We have to reconcile this consideration with the comments provided by Reviewer #1, who for example asked to expand on activity pattern and fossorial species. We did our best to merge these requests.

Comment: The Introduction nicely sets out how views of past adaptations can inform future responses to climate change, but the discussion does not strongly return to this concept.

Response: Good point. We have added in the last part of the discussion the following: “**By leveraging the database presented here, future research can gain deeper insights into how species may respond to changing environmental conditions, providing information for predicting species' responses to future climate scenarios, helping to inform conservation strategies and our understanding of biodiversity dynamics.**”.

Comment: 314: again, it is not vegetation structure but the impact it has on temperature that is the selective force, as articulated in line 248

Response: We clarified this last point by revising the text to: “Our results clearly show that brightness variation of squamates **is influenced by habitat openness and its underlying selective forces**”.

Response to reviewers to manuscript NCOMMS-23-56634A

Title: "Habitat openness shapes color evolution of squamates over deep time"

by Jonathan Goldenberg, Karen Bisschop, Joshua W. Lambert, Michaël P.J. Nicolai, Rampal S. Etienne, Liliana D'Alba, Matthew D. Shawkey

Dear Reviewers and Editor,

Thank you very much for your comments and appreciation. We have carefully addressed all remaining concerns and successfully incorporated them in our revised submission. As suggested by Reviewer #4, we have included a new analysis testing how evolutionary rates change over time in response to variations in $\delta^{18}\text{O}$ using RPANDA. These results align with our previous findings, providing additional evidence to support our conclusions. Below you can find our specific responses to all comments. All modifications in the revised article are indicated in red for easy reference.

Reviewer #1

Comment: I have reviewed the revised version of the manuscript, as well as the response to my comments and those of the other reviewers, and find the article is now satisfactory to grant ready acceptance for publication. My comments have been thoroughly addressed and responded to in a clear manner, and I have no further comments to add. I am keen to see this excellent article added to the literature on the evolution of colour variation in squamates.

Response: Thank you very much for your time and for offering valuable feedback to help us improve our work!

Reviewer #3

Comment: I have read the revised version and the authors have addressed most of my concerns by tightening up their arguments and language. The manuscript is now in very good shape.

I only have three small comments for the authors to consider:

Response: We sincerely appreciate your time and the great feedback you have provided to help us improve our study!

Comment: 61-69: this is a single sentence and too long and complex. Can you please edit it into 2 or even 3 sentences.

Response: we have revised the text and followed your suggestion. It now reads: "Here, to gain understanding of the mechanisms driving their color brightness evolution and to specifically evaluate whether selected drivers are consistent across taxa, despite the opposing predictions found in the literature¹⁴, we compiled a new global dataset of squamates' life-history traits and ecological information known to influence brightness^{18,26}. **This dataset includes** habitat openness, latitudinal and altitudinal distributions (which correlate with environmental factors such as temperature, photoperiod, and UV exposure), body mass, concealment, polymorphism, and circadian rhythm. **It covers** 1372 aquatic and terrestrial species from 25 families **representing** major squamate groups."

Comment: 74: I think the bright integument reflects incoming light rather than heat.

Response: we changed light for heat in the revised text: "... possibly to better reflect and thereby reduce the incoming **light**."

Comment: 76: my comment is again that latitude itself is not a selective pressure, the strength of selection might vary with latitude

Response: we adjusted the text and it now reads “Other family-specific **strength of selections** are latitude and interactions between...”.

Reviewer #4

Comment: This manuscript examines the evolution of brightness in squamates. The main conclusions are that habitat openness has been the primary driver of the evolution of color integuments and that this openness was influenced by aridification during the Miocene-Pliocene transition. I have not been involved in the earlier rounds of the review process and I'm not a squamates specialist. Overall, the manuscript is well-written, although I believe several details are missing from the Materials and Methods section (as noted by one of the previous reviewers), such as whether the data were transformed, what the package implementations were, etc. However, I do have some concerns regarding the conclusions, which may be overstated or driven by the statistical analyses. These issues need to be addressed before recommending publication.

Response: thank you for reviewing our work. We have carefully considered, addressed, and incorporated all your comments and suggestions which are outlined below.

Scaling of the Variables

Comment: It is not specified whether the predictors in the (Bayesian) regression were scaled, nor is it clear if the response variable (brightness) was log-transformed in both the regression and model fits. This is problematic because such transformations (e.g., using the logarithm of the studied trait) can affect model selection and the estimated parameters. For instance, in the regression analysis, if the predictors are not scaled, the interpretation of the estimated coefficients becomes difficult (note that the coefficients and their HPDs are not reported in the results or in the supplementary information – a summary table of these values, in addition to Figure 3, would help in interpreting the results and making comparisons in future studies). Furthermore, if the predictors were not scaled, the default priors used may bias the estimation of coefficients (and thus the relationships) depending on the scales of the predictors.

Response: we did not transform or scale our response variable, brightness, as it is already scaled from 0-100% and normally distributed. Regarding the continuous predictor, body mass, we used the log-transformed values as provided in Feldman et al. (2016). Scaling this variable further is unnecessary because it is already in an appropriate unit for analysis. Moreover, all other predictors in our model are categorical factors with multiple levels (such as altitude and distribution). Therefore, the concern about the scaling of continuous predictors is addressed in our approach. For clarity, we did not omit the reporting of coefficients and HPD in our supplementary materials. Due to the complexity of our models, which includes six predictors (five of which are factor variables with multiple levels) and numerous interaction effects, each model comprises over 80 variables. Given that our supplementary file already extends to 40 pages, we aimed to present the data concisely. Therefore, all figures in the supplementary material display the HPD and posterior means clearly (and account for the effects of all other predictors). For more detailed information, we refer readers to our associated rmd file and its environment.

Relationship Between the Rate of Evolution and Past Climatic (Habitat Opening) Changes

Comment: One of the main results of this paper is that changes in environments (habitat openness), as reflected in fluctuations in climate (foraminifera $\delta^{18}O$ curve), have driven brightness evolution. To

support this, the authors reconstruct branch-specific rates of evolution using a method that models the evolution of the rates as a geometric Brownian motion (GBM) (i.e., the log of the rate evolves as BM), and then perform correlation analyses with the curve. However, I believe that the observed pattern of increasing rates could be driven by two factors, and the conclusions may be overstated. First, the best-fit model was an Ornstein-Uhlenbeck (OU) process. This suggests either there is substantial measurement error in the brightness color proxy (see below) or that trait evolution was somehow constrained (the alpha value is not small, as suggested in the responses to the review, because it depends on the depth of the tree. Instead, for a 202 million-year-old tree, an alpha of 0.01 means that the signal halves after ~69 million years). If the traits are measured with error or follow an OU process, the rates in the more recent (shorter) branches will appear higher than they are, either to compensate for variance contributed by the errors or to account for relative changes in a shorter period in a stationary process such as OU. Second, the GBM model used has a variance that scales with time, and as a result, the average of log-normally distributed rates might be biased toward increasing over time. Together, these effects can lead to a spurious increase in rates near the present.

Response: we realize there may have been some misunderstandings due to our writing, which we address in this revision. First, the evolutionary timescale for our rate analyses is much shorter than 202 million years, as our study focused on specific clades with the Lacertidae crown age dating back approximately 80 million years. Second, we specifically tested multi-regime OU models by fitting generalized Ornstein-Uhlenbeck-based Hansen models of continuous characters evolving under discrete selective regimes. As stated in the manuscript (L228): “To detect whether climate change over time has shaped the rate of evolution of color brightness, and to ensure that the observed results are not due to late burst patterns, we compared a single-rate model, which is independent of vegetation structure, to multi-regime Ornstein-Uhlenbeck (OU) and Brownian motion (BM) models where the rate regimes depend on changes in vegetation structure. We found that the OU multi-regime model has the highest support (wAIC across families = 0.97-1.00), indicating that vegetation structure plays a significant role in shaping the evolution rate of color brightness.” This strong support suggests that the OU model, accounting for environmental factors, better explains the evolutionary dynamics of color brightness. This aligns with the point raised about an OU process reflecting constraint, while also providing a nuanced understanding that these constraints are tied to specific ecological factors rather than just being a result of measurement error.

Comment: In addition to the fact that spurious correlations can arise when time series are studied with improper tools, other periods of global cooling (e.g., the Eocene-Oligocene transition) do not show an increase in rates in Fig. 5. This suggests that either habitat openness was not a driver during these other periods, or that the observed pattern is due to potential biases, such as those discussed above.

Response: the observed lack of increased rates during periods such as the Eocene-Oligocene transition in Fig. 5 can be explained by the evolutionary timing and diversification of the examined squamates. During these periods, many clades had not yet diversified or were in their early stages of evolution. Therefore, the absence of an increase in rates is consistent with the evolutionary history of these groups and not due to statistical methods.

Comment: As one of the previous reviewers advised, there is a model in the RPANDA package (Clavel & Morlon, 2017, PNAS) that explicitly models the rates of evolution as a function of an external (e.g., climatic) curve. This model also allows for the joint estimation of measurement error and can thus account for possible bias toward inferring higher rates near the present.

Response: to provide another line of evidence for our claims, we now fit a model to study how evolutionary rates change over time in response to variations in $\delta^{18}\text{O}$. We have incorporated this analysis into our manuscript (Fig. S35) and included the output below. The similar patterns observed between RPANDA and Phytools improve our findings by demonstrating that different methods

converge on similar evolutionary insights. This consistency supports the robustness of our conclusions about the evolutionary rate dynamics.

Measurement Errors in Model Fit

Comment: It is not specified whether measurement errors (ME) were addressed in the analyses, or how they were handled. If no estimates of ME are available, it is generally possible to infer them jointly during model fitting (Silvestro et al., 2015, MEE). The "phylolm", "Geiger", or "RPANDA" packages allow for the estimation of measurement errors. This is important because ME can influence model selection and parameter inference (see the comment above on the increase in rates near the tips).

Response: in our analysis, we did not have any prior estimates for measurement errors. Therefore, we followed the default behavior of MCMCglmm, which incorporates measurement error by specifying the variance structure as $VG \otimes D$. In this structure, D is a diagonal matrix representing the measurement error variances, and VG is a scalar fixed at 1. This approach allows for the inclusion of ME in the model without requiring prior estimates, thereby accounting for potential biases and ensuring robust parameter inference. For clarity we added such considerations in our revised text (L 483-486).

Specific Comments

Comment: L. 79: Although it is not incorrect per se, I think "fractionation" would be a more precise term ("a measure of fractionation of ^{18}O to ^{16}O ") as it describes the process of changing the relative isotopic abundances

Response: thank you for this! We have incorporated your suggestion.

Comment: L. 226-227: It is not valid to compare differences in the DIC criterion (or likelihoods in general) that were computed on different datasets (e.g., with and without fossorial species) and use these differences as compelling evidence for higher support. If the predictors were standardized (see comment above), then the absolute values of the coefficients could be interpreted as effect sizes.

Response: this is a very good point. Instead of comparing the models via DIC, we now evaluate their credible intervals. By examining the credible intervals, we can assess the precision of the model's predictions. Larger credible intervals indicate less precise predictions, thereby providing a more reliable measure of model accuracy. This approach allows us to effectively compare model performance without relying on DIC differences computed from different datasets. We now write in L227-232: "Again, the model predicts the same patterns as the total one (Table S2) but with higher support. This is evident from the smaller credible intervals when comparing the models "without fossorial" vs. "with fossorial" species across different habitats: Open (41% [35,44] vs. 41% [36,46]), Semi-Open (39% [36,41] vs. 39% [36,43]), and Closed (34% [29,37] vs. 36% [31,39]). These smaller credible intervals confirm the higher uncertainties of brightness prediction in fossorial species (Fig. 4).".

Comment: L. 446-449: This is a misleading way of interpreting AIC differences, because for some nested models, the maximum possible AIC "delta" is much lower than these ranges. For instance, the support of the BM model over the EB or OU models cannot reach a delta AIC of >2 if BM is the true process and there is sufficient statistical power. I recommend that either the best model be used in downstream analyses, or that only the few best models be compared. While such guidelines are often used, they may be misleading for future studies that follow your protocol.

Response: to address this, we followed a systematic approach by comparing BM, OU, and EB models using the phylolm package, incorporating measurement error. Our analysis identified the OU model as the best-supported model with a AIC > 10 between OU and BM or EB. While we recognize that large AIC differences can be influenced by model nesting, our approach aimed to provide a comprehensive comparison. We ensured that our downstream analyses are based on the best-supported model, as indicated by the AIC values (EB = 8435.443, BM = 8756.224, OU = 8424.152).

Comment: L. 458-486: The citation and statement are incorrect. Most models of trait evolution (as opposed to diversification models, which the citation refers to) are based on the covariances between species that depend on their shared evolutionary history. Limited phylogenetic sampling—if sample size is sufficient—will not bias the results as it would for diversification models.

Response: we agree with this clarification. We revised the text and now reads (L492-494): "For statistical power and to avoid potential biases in estimating evolutionary rates, we applied more stringent selection criteria than previously, and we examined only seven families for which we had at least 50% of the focal species represented in the phylogeny"

Comment: L. 488-491: Please cite the package implementations used for fitting the various models.

Response: done! We have added that we fit these models using "phytools".

Comment: L. 505: "using the 'rescale' function".

Response: incorporated – thank you.

Comment: Fig. 2: Specify how ancestral states were reconstructed and which model was used.

Response: we clarified the model of evolution in the caption, and we explain that the visualization has been performed in "ggtree" following "phylolm" analyses (also in methods).

Comment: Fig. 4: Are these the marginal means? Were they computed after accounting for the covariates?

Response: Fig. 4 show posterior means density coupled with posterior means credible intervals accounting for the covariates. We wanted to understand the distribution and uncertainty of our model predictions.

Comment: Fig. 5: The figure shows confidence intervals (CIs) that are negative for the evolutionary rate, which cannot be correct, as rates cannot be negative. Proper CIs must be calculated (e.g., 95% percentile of bootstrapped datasets), but see the comment above regarding the methods used to assess the link between rates of evolution and the d18O curve

Response: you are correct. This is a statistical artifact due to higher uncertainties at the root of each examined clade. We have addressed this issue in the revised version – see below the revised output.

Response to reviewers to manuscript NCOMMS-23-56634B

Title: "Habitat openness shapes color evolution of squamates over deep time"

by Jonathan Goldenberg, Karen Bisschop, Joshua W. Lambert, Michaël P.J. Nicolai, Rampal S. Etienne, Liliana D'Alba, Matthew D. Shawkey

Dear Reviewer and Editor,

Thank you very much again for your comments. We acknowledge the importance of considering measurement errors in phylogenetic comparative methods. As such we followed the clear script provided by the reviewer and performed 77 additional models testing the impact of different noise and measurement error levels on brightness evolution rates. Overall, we found strong support for our conclusions which we detail below in our responses. All modifications in the revised article are indicated in red for easy reference. Fig. 5 and Fig. S35 have been revised and updated accordingly to the latest results.

Reviewer #4

Comment: I have read the revised version, and the authors have addressed some of my concerns, but not all of them. I am still not convinced by the claim that the rate of evolution follows climatic changes (or habitat opening). At the very least, I think the current analyses are not conclusive, as they fail to rule out potential biases.

Response: Thank you for these helpful suggestions. We agree and have incorporated them in the revised text. We merged the first two comments together for conciseness.

Comment: First, the authors' argument that support for a multi-optimum OU model is consistent with constraints rather than ME is insufficient. This conclusion would only be valid if ME was incorporated into the analysis. Furthermore, support for the OU (or OUM) model also suggests that increased rates near the present might be spurious. An OU process is stationary, meaning that closely related species could exhibit as much divergence as more distantly related ones (i.e., all species are constrained to evolve around an optimum). If rates are estimated from data generated under such a process, an apparent increase in rates might simply reflect the need for more changes to explain similar levels of divergence in more recently diverged species, not a true signal of rate increase, especially if the data is estimated with some uncertainty.

Second, in my previous reviews, I suggested that, in the absence of uncertainty estimates for brightness, the effect of ME on the observed increase in rates could be assessed by modeling it as a nuisance term (as in Silvestro et al. 2015). The authors explain that such a term was used for the regression analysis, but they do not provide evidence that this effect was considered in the analysis of evolutionary rates. I demonstrate with a simple code (see below) that I can replicate the pattern of increasing rates at the tips, similar to the one found by the authors—including the last figures shown in the response—by modeling a process with a constant rate and adding random noise at the tips (i.e., ME). This noise affects the estimation of rates in both RPANDA and Phytools. When estimating ME in RPANDA, this effect can be partly compensated for, although more simulations would be needed to assess the uncertainty.

Response: We followed your suggested script and examined the effect of noise and measurement error levels on brightness evolutionary rates by simulating low (square root $\sqrt{10}$, $\sqrt{50}$), moderate ($\sqrt{100}$, $\sqrt{150}$), and high ($\sqrt{200}$) variabilities. We found that (L196-201): "... all families followed the same late burst patterns when examining only simulated data (Fig. S35). We also found similar patterns when accounting for measurement errors, but interestingly, few

deviations emerged. For Varanidae, we observed an early burst pattern in the baseline model (i.e. no noise, only measurement error) as well as under a simulated low-noise condition. In Phrynosomatidae, the baseline and all simulated data, except one with moderate-high noise, followed early-burst patterns (Fig. S35)."

In methods we have added the reference to Silvestro et al. 2015 MEE (L519-523): "As another line of evidence, we evaluated how evolutionary rates change over time in response to $\delta^{18}\text{O}$ using *fit_t_env* and *plot_fit_t_env* in "RPANDA"⁹¹. This allowed us to examine the effects under different levels of simulated noise and measurement error, broadly defined as low (v10, v50), moderate (v100, v150), and high (v200), given their established importance in phylogenetic comparative methods⁹⁵."

Comment: Finally, the authors state that "the observed lack of increased rates during periods such as the Eocene-Oligocene transition in Fig. 5 can be explained by the evolutionary timing and diversification of the examined squamates." However, the increase in temperature during the late-Oligocene/early-Miocene began around ~25 Ma. Most of the crown ages of the clades considered date from >40 Ma (and >80 Ma for some). It seems unjustified, and not clearly discussed, to rule out the possibility that habitat opening could have left a significant imprint before this time (i.e. colder periods such as the Oligocene ~34-23Ma). Several studies have evidenced the possibility to detect early increases in rates.

In summary, without further evidence showing that the observed pattern cannot be explained by simple biases, I would remain cautious about the conclusions regarding the role of habitat opening in promoting the evolution of color brightness.

Response: Following the inclusion of measurement errors in our analyses and finding the interesting early burst patterns in Varanidae and Phrynosomatidae, we have added the following considerations in the discussion (L.319-332): "Our findings follow such a pattern in that long evolutionary stasis in brightness is followed by a rapid burst in variation associated with new environmental conditions (Fig. 5), a trend exhibited across all families under different, simulated noise levels (Fig. S35). Although most families consistently followed this pattern even when accounting for measurement errors, Varanidae and Phrynosomatidae partially deviated, showing potential early adaptations and static rates. These early burst and static patterns were observed in only seven out of the 84 examined evolutionary rate models, which are possibly driven by the rapid temperature decrease following the Azolla (49 Mya ca.) and the Eocene-Oligocene extinction (33.5 Mya ca.) events in Phrynosomatidae and Varanidae respectively. Whether these patterns represent a biological signature or reflect other factors remains uncertain. While this study focused on extant species, well-preserved squamate fossil records could provide an opportunity to empirically describe their colored integument, thereby calibrating trait reconstructions and offering further insights into the early evolution of these families."

Fig.S35. Evolutionary rate analyses. Here we fit a model of trait evolution for each examined family in which evolutionary rates depend on $\delta^{18}\text{O}$ fluctuations under different simulated noises (from $\sqrt{10}$ to $\sqrt{200}$; left column) and measurement error (right column).

Response to reviewers to manuscript NCOMMS-23-56634C

"Habitat openness and squamate color evolution over deep time"

by Jonathan Goldenberg, Karen Bisschop, Joshua W. Lambert, Michaël P.J. Nicolai, Rampal S. Etienne, Liliana D'Alba, Matthew D. Shawkey

Dear Reviewer and Editor,

Thank you once again for your valuable feedback. In response to Reviewer 4's comment and the editorial suggestion, we have revisited our conclusions regarding the evolutionary rate analyses of color brightness and its association with temperature changes. We have removed any definitive statements from the text and revised it accordingly. Below, in response to the reviewer's comment, we provide specific examples of these revisions.

Reviewer #4

Comment: I have reviewed the revised version and the authors' responses to my previous comments. I appreciate the authors' efforts to address the issues raised in the previous submission. However, I believe the additional simulations do not resolve the problem (although it improves a bit on the previous version); instead, they illustrate the difficulty of ruling out the possibility that the observed association with recent climate cooling is spurious.

The additional simulations (Fig. 5, S35) involved adding noise to the empirical data. This approach means that any bias in rate estimates due to noise or measurement error (ME) in the empirical data may be exaggerated in the simulated datasets. Indeed, as shown in Figure S35, the rates estimated toward the tips from data contaminated with extreme noise levels are higher than those estimated from the raw data. This observation supports the comments from the previous review that noise could generate the observed pattern of rate increases near the tips. However, this does not provide evidence that the rates estimated from the empirical data are not influenced by such noise. The simulations in Figure S35 demonstrate that with higher noise levels, the rate estimates near the present are considerably higher than those for data simulated without or with low levels of noise, even after attempting to control for noise by estimating an ME term. While estimating the ME does tend to reduce the effect of noise, it also reveals that there remains a bias that could spuriously support a climate effect. In other words, without such a bias, we would expect the curves in the right panel of the figure to overlap.

Given that the only climate association proposed is the recent global cooling starting from the late Miocene, and considering that noisy data can produce a similar pattern, along with the difficulty of controlling for such noise, I maintain that the current results are likely artefactual – or at least a spurious association cannot be ruled out. A more robust set of simulations and analyses aimed at reducing or controlling for the detected biases would be necessary to convincingly demonstrate an effect of recent climatic changes on the rate of evolution.

Response: Thank you for reviewing our work once again. To address the latest concerns, we have made the following changes to the text to remove any implications of causality.

Title

- "Habitat openness shapes color evolution of squamates over deep time" changed to "Habitat openness and squamate color evolution over deep time".

Abstract

- L21: "...brightness evolution rates track $\delta^{18}O$..." changed to "...brightness evolution rates **likely** track $\delta^{18}O$..."
- L22-23: "...such as those in the Miocene and Pliocene, due to the establishment of an arid climate..." changed to "...such as those in the Miocene and Pliocene. **This trend may** be due to the establishment of an arid climate..."
- L24: "Our findings reveal that..." changed to "Our findings **suggest** that... **is associated with**..."

Introduction

- L73: "...habitat openness persistently drives brightness evolution..." changed to "...habitat openness **is associated with** brightness evolution..."
- L79: "Moreover, brightness evolution rates track foraminiferal..." changed to "Moreover, brightness evolution rates **likely** track foraminiferal..."

Results

- L188: "Brightness evolution rates track climatic changes" changed to "Brightness evolution rates **and** climatic changes"
- L195-196: "...which track the aridification phases..." changed to "...which **associate with** the aridification phases..."
- L212-213: "...and discovered that most rates systematically increased starting in the mid-Miocene..." we removed "systematically" and now reads "...and discovered that most rates increased starting in the mid-Miocene..."
- L215-216: "...early-burst patterns in Phrynosomatidae and Varanidae were linked to the rapid cooling..." changed to "...early-burst patterns in Phrynosomatidae and Varanidae were **associated with** the rapid cooling..."

Discussion

- L298: "Given such opposing predictions, it is remarkable that we have here identified a single consistent factor driving global patterns of its evolution" we removed "consistent" and changed to "Given such opposing predictions, it is remarkable that we have here identified a single factor **associated with** global patterns of its evolution".
- L306: "Indeed, our analyses revealed that the rapid evolution of brighter and darker integuments..." changed to "Indeed, our analyses **suggest** that the rapid evolution of brighter and darker integuments..."
- L329-330: "Although most families consistently followed this pattern..." we removed "consistently" and now reads "Although most families followed this pattern..."
- L391: "Our results show that brightness variation of squamates is influenced by habitat openness and its underlying selective forces..." changed to "Our results **suggest** that brightness variation of squamates is **associated with** habitat openness and its underlying selective forces..."

Additionally, we have further clarified the limitations of our evolutionary rate associations:

- L336-340: "Whether these patterns represent a biological signature or reflect other factors remains uncertain. **Potential biases, such as measurement error and noise in rate estimates, may contribute to the observed trend, making it difficult to fully disentangle true evolutionary signals from artifacts. While our analyses attempt to account for these biases, some residual effects remain, highlighting the need for more robust simulations and sensitivity tests.** While this study focused on extant species, well-preserved squamate fossil records could provide an opportunity to empirically describe their colored integument,

thereby calibrating trait reconstructions and offering further insights into the early evolution of these families”.

Reviewer report for NCOMMS-23-56634

Key results – The manuscript presents a compelling and novel results on the evolution of colour brightness variation in global squamates. The main result, that brightness correlates closely with environmental variation, both spatially, temporally, and across clades, provides cogent answers to a relevant and hitherto mostly unexplored biological question. First, the evolution of these brightness features is visualised across different biomes, showing a clear prevalence of brighter colourations in open habitats, but not clearly across wide latitudinal gradients. Second, it is mapped across a phylogenetic tree to show the phylogenetic clustering and historical trends across the evolution of squamates. Further analyses evaluate the relations between brightness and a series of habitat variables, finding that vegetation structure is the strongest predictor of colour brightness evolution, as also shown previously in the geographical and phylogenetic visualisations. Finally, the manuscript reconstructs brightness evolution through time in a sample of squamate clades, showing how it tracks together with variations in climatic conditions as predicted by the variations in oxygen isotopes, which the authors take to represent an evolutionary response to variations in vegetation structure.

Validity – I believe that the study takes on an interesting biological question and does a good job at addressing it from a variety of different angles. I particularly appreciated how phylogenetic, ecological and spatial patterns are integrated in a compelling interpretation of the history of colour brightness in squamates. However, I do have some concerns with the interpretation of some of the results, due to the insufficient justification for the use of your proxy for vegetation structure. I do not take issue with the results themselves, here, but with the choice to use (and here I quote your own text) ‘variations in foraminiferal $\delta^{18}\text{O}$ as a proxy of vegetation structure change over time’. I went and checked the article you used as a reference, Herbert et al. (2016). While the article does say that there were changes in the vegetational compositions in different periods due to shifting climates, there is little that explicitly addresses vegetation structure itself beyond mentioning that there was a transition between wet and dry climate plants. In my opinion, here there is a leap in logic that should at least be better expanded upon. These estimates are based on variations of sea temperatures and levels through time, which in turn are estimated based on variations in oxygen isotope concentrations. It just does not seem to be strong enough of an association to be used as a proxy of vegetation structure, not without a clearer and stronger supporting explanation. To be clear, I am not saying that I think these interpretations are invalid, just that they should be better explained to the reader across the text. Additionally, I do have some concerns with the sampling strategy used in the article (see Data & Methodology section).

Significance – This work represents a valuable scientific contribution to a relatively unexplored field. Specifically, it is significant because a) it addresses a series of biological questions that have only been taken into scarce consideration by the existing literature, and b) it does so in Squamates, a study system which is well-suited for this type of study, having an impressive taxonomic diversity, global distribution, and a long and complex evolutionary history. The way these questions are addressed uses well-established analytical approaches which could allow their extension to broader taxonomic categories. Thus, I believe that this article will be a valid addition to the existing literature on the topic.

Data & methodology – The data used in this article appears to be generally well justified in both its acquisition and its uses, with some exceptions that need addressing. In particular, I have some issues with the naming of certain variables, and their representation within the article. Additionally, I am adding some notes and observations related to data collection and methodology. Below, I will list

the things that, in my opinion, would benefit from your consideration for the improvement of this work:

-vegetation structure: the name of this variable is not entirely appropriate to indicate the categories that you use in your work, specifically in the light of one of those categories, 'Subterrane'. What does living underground have specifically to do with vegetation? A lizard can be fossorial in a forest as well as in an open plains. I believe that these categories could be better framed as 'habitat openness', with subterrane being the least open of all categories. Additionally, the OpenClosed category would probably more understandable if it were named simply 'semi-open'.

-Related to the previous point, on a more methodological note, while you did run analyses excluding fossorial (subterrane) species, I wonder whether you considered examining the effect that fossoriality specifically has on brightness variation (against all other categories). Fossorial species are very interesting in that they probably do not conform to the same rules as other species (i.e., thermoregulation is less likely to affect their colouration), and their numerical representation across squamate clades is very large. Writing a bit more on that in your discussion would be beneficial to contextualising your findings.

-I noticed that you included a coding of the activity patterns of lizards in your analyses, and that in the results, you mentioned that diurnal species are brighter than the others. It honestly surprised me that not much was included about this in the discussion, as it seems like a very relevant result. For example, the relationship between lifestyle and activity patterns has been discussed by recent papers (i.e., Slavenko et al., 2022), and it would be great if this type of analysis were to be contextualised in the light of colour evolution.

-Taxon sampling: While you do provide a good justification for the numerical sampling of taxa in your analyses (that is, what proportion of each clade was considered sufficient), the taxonomic (and, relatedly, geographic) diversity within each of the clades included in your work is not very well represented. While the representation within genera is quite good, there are very few genera per clade included. Additionally, there is a disproportionate numerical representation of some groups against others. This also means that many clades were essentially completely excluded from your analyses, biasing the results towards very few select genera. I am aware that taxon sampling is a difficult thing to do (and also limited by the information available), and sampling philosophy can be subjective, but I would have at least expected to see a better representation of genera across the phylogeny of each group. Using an example from your own data (on skinks because I am more familiar with this group), when more than one-fourth (33/116) of your sample of skinks belongs to the genus *Plestiodon*, and about one-fifth (21/116) by the genus *Chalcides*, how much are your results really saying about skinks in general? How much about *Plestiodon* and *Chalcides*? There are 168 skink genera, only 25 of which are included in your dataset, with most of the dataset being represented by species belonging to five or six of them. In my opinion, this is a methodological weakness that should at least be acknowledged and addressed within the article's methods and discussion.

Analytical approach – The study takes a valid and adequate analytical approach to its analyses, both statistically and in terms of data interpretation. While I would in some cases use different approaches to the same statistical/phylogenetic problems, I do not find any particular issue with any aspect of the analyses. I particularly appreciated the model validation included in the text. It does show how much careful consideration went into this article, to ensure the validity of analyses and evaluate their potential pitfalls. The phylogenetic analyses employed here appear to be sound, but I

would like to see some more information about the alternative models of character evolution that you tested, and why you decided to use an OU early-burst model specifically.

Clarity and context – This article is relatively well contextualised within the existing literature, but it could use some major improvements in its clarity and stylistic choices. I will proceed to comment on each section in order:

Introduction: mostly well contextualised and clear, it sets the theoretical stage for the analyses that follow. I would have liked to see the questions/hypotheses or at least an aim of the paper to be more clearly spelled-out. And I am unsure whether you should summarise the results at the end of the introduction (that should be the abstract's role, but feel free to ignore this if that is in the journal's style). Instead, you should write what you did to address your hypotheses.

Results: a bit unclear, as it reads more like a mix of methods and results, which could confuse a reader trying to parse the text. This also means that there is quite a lot of redundancy between materials/methods and the results. This redundancy impacts clarity. The results section should be also better sectioned in smaller paragraphs to improve clarity.

Discussion: The discussion is generally well written, presents coherent arguments, and is clear in most of its parts. However, some of the results (i.e., the global relationship with activity pattern/circadian rhythm) are incompletely or not at all discussed, as the article chooses to selectively focus on some of the results instead of others. Additionally, the part of the discussion (lines 295-306) that focuses on the observed interaction between body size and altitude is very opaque, and its logic is not very clear. I read the paragraph several times, and I am not really sure of what are you trying to say with it. If your patterns are not at all consistent across taxonomic groups, perhaps it does not make much sense to try to explain them as a whole.

Materials/methods: No particular comment on this part, except what already stated in other sections of this report.

Figures: See specific comments below.

Suggested improvements – For general suggestions about the article, see each of the sections above. Below, I have listed some specific suggestions for parts of the text which require improvement.

Line 75: As this is the first time you mention the ratio of stable isotopes of oxygen, you should define it briefly.

Lines 86-88: Beyond the expected relationship between brightness and vegetation structure, it's really interesting that you found brightness not to correlate with latitudinal variation. Any reason why you think that may be the case?

Fig. 2: It is a bit hard to visualise the brightness scale on the ancestral reconstruction image. Perhaps a different (starker) scheme would be appropriate. The labelling of the different clades on the tree could be graphically accomplished better. Also, the separation between the clades is a bit hard to see. One possible suggestion is using alternating, thicker bands of two different colours above each of the clades.

Line 123: At least 50% of the focal species? What do you exactly mean by that? What are these 'focal species'?

Line 120: 'using phylogenetic comparative methods' is a bit vague. Please say which specific operation you performed. Additionally, I am unsure whether this part of this section should go into the methods instead. Mentioning the operations that you performed outside of the methods is fine, but I think that the choices of which groups were analysed and why definitely belongs to the methods and not the results.

125-126: So, you lost more than 70 species due to reclassifications? That sounds like a lot of species lost only due to reclassification. I would have expected a loss due to species being absent from the phylogeny, not due to reclassification. If possible, add example citations on the greater reclassifications efforts that have caused this taxonomic reduction.

Fig. 3: It's not a bad figure per se, but perhaps I would advocate removing the icons at the bottom of the image, or at least placing them at the top next to the relevant variables. They are a bit confusing in the context of the figure, otherwise. If space is an issue, you could try switching the X and the Y axis of the figure, making this operation easier.

150: Why 'surprisingly'?

Line 165 (caption): Empty cells denote a lack of *significant* correlation.

Fig. 4: This figure has some issues and would benefit from being made smaller and more informative. For example, the confidence intervals of the different categories could be illustrated a bit better. I also imagine that there will be some outliers, which are not shown here but would be really easy to display (depriving a reader of that information would diminish the informative value of the figure). A basic ggplot R script with `geom_boxplot` would fix these issues. In that case, I recommend keeping the colour exclusively in the confidence boxes. Moreover, there is an issue with the similarity between the white-to-black dorsal brightness bar on the Y axis, and an identical bar on the X axis that is used to indicate vegetation structure. As the categories are not explicitly continuous, it is questionable to use the variation in a continuous colour bar to indicate them. I also am unsure whether OpenClosed (an unfortunate category name – again, wouldn't 'semi-open' be better?) and especially Subterranean really represent vegetation, at least in the way they are displayed in the images. Is being in a subterranean cave 'vegetation' structure? Looks to me like this category is not about vegetation at all, but about habitat structure or type. See previous comments about this point.

Line 168: 95% *confidence intervals*.

Line 173: Again, it is unclear to me what do you exactly mean by 'focal species'.

Line 179: Reiterating the point from the general evaluation of the article, I am unconvinced that the variation in O₂ isotopes means what you take it to mean, or at least, that the evidence you present is very solid. Yes, overall vegetation structure saw increases or decreases in certain periods, but it seems to me like an oversimplification of the real trends. Could any control factor be introduced or at least considered for latitude or (reconstructed) biomes?

Lines 204-210: This is, again, something that would be more suitable as a part of the methods section.

Line 213: It is a good thing that you checked for phylogenetic bias in these two groups, but at that point you might as well extend these analyses to all the groups. The fact that the models for those two groups are not affected by phylogeny does not imply that all the other groups will be.

Line 203: The Model Evaluation section lists a series of supplementary analyses that you performed to evaluate how robust your models are, which is a laudable effort, but I believe that its structure

necessitates improvement. First, the choice to keep this section in a single paragraph makes it hard to keep track of the text. It needs to be rewritten in a more focused way that separates in at least a couple of distinct paragraphs. Additionally, I am unsure whether parts of this section wouldn't be better included in the methods. In my view, and I will leave it to the editors to weigh in on this matter, the best way to make this part work would be to include the evaluation in the methods sequentially after each original version of the model and keeping the same sequence in the results. For example, in the methods I would write something like: 'to assess the evolution of dorsal brightness in squamates, we did **X**. Then, to validate it, we did **Y** for these groups and these other groups etc...', and in the results 'The result of **X** was **x**, which was validated by doing **Y**.'

Lines 401-404: Model comparison should not be included in the methods but in the results. Additionally, it would be best practice to include (in the methods) a minimum threshold for model selection (under which two compared models are considered to perform not significantly better than the other: for an example, see Bergmann & Morinaga, 2019).

Lines 404-406: If you found clustering of brightness where biomes select for different colours, why was this not explicitly tested statistically?

Lines 415-419: I am admittedly a bit unfamiliar with this modelling approach, but why did you account only for interactions between two of the fixed effects, instead of all or more combinations of them, at least as a preliminary step in your model construction?

Lines 407-432: Sometimes it is unclear which functions are associated with which packages.

Lines 447-463: Here I would like to put the accent on the unnecessary redundancy between this part of the methods and the results section. As I wrote above, this section, and not the results, is where these methodological notes should belong.

Lines 295-298: I think that this part of the discussion should be explained more clearly. If the trends are all over the place depending on the clade, I do not see how you can draw a consistent unitary explanation from them. To me it reads like a circular argument.

Lines 308-312: Maybe add some citations to this information. Also, the double parentheses in the last sentences make reading difficult. Perhaps break the sentence in two.

Reviewer recommendation: The article presents valuable findings that provide novel insights on a relatively unexplored topic. However, it is my opinion that significant changes and clarifications are required to grant its acceptance. If the relevant alterations are made, I would be happy to see this paper published in the journal.

References

Bergmann, P. J., & Morinaga, G. (2019). The convergent evolution of snake-like forms by divergent evolutionary pathways in squamate reptiles. *Evolution*, 73(3), 481-496.

Herbert, T. D., Lawrence, K. T., Tzanova, A., Peterson, L. C., Caballero-Gill, R., & Kelly, C. S. (2016). Late Miocene global cooling and the rise of modern ecosystems. *Nature Geoscience*, 9(11), 843-847.

Slavenko, A., Dror, L., Camaiti, M., Farquhar, J. E., Shea, G. M., Chapple, D. G., & Meiri, S. (2022). Evolution of diel activity patterns in skinks (Squamata: Scincidae), the world's second-largest family of terrestrial vertebrates. *Evolution*, 76(6), 1195-1208.

```
---
title: "Squamate macroevolutionary analyses"
output: html_notebook
editor_options: Jonathan Goldenberg
  chunk_output_type: console
---
```

```
```${Reset command}
```

```
closeAllConnections()
rm(list=ls())
```
```

The below code provides the Rscript to replicate the results for the manuscript *'Vegetation structure shapes color evolution of squamates over deep time'*

The script is divided in *17 chunks* - To quickly navigate through them use the *Outline* option:

1. Load the library and dataset * L36**
2. Ancestral state reconstructions * L89**
3. Cluster evaluation for altitude categories * L167**
4. Bayesian modeling Squamata level (full) * L183** Here you can also load the full environment (*L188**)
5. Bayesian modeling Squamata level (substet ≥ 5 pictures) * L614**
6. Bayesian modeling Squamata level (substet w/o fossorial species) * L659**
7. Bayesian modeling Squamata level (bootstrap) * 703**
8. Bayesian modeling Varanidae * L901**
9. Bayesian modeling Viperidae * L947**
10. Bayesian modeling Elapidae * L1069**
11. Bayesian modeling Scincidae * L1114**
12. Bayesian modeling Lacertidae * L1261**
13. Bayesian modeling Cordylidae * L1407**
14. Bayesian modeling Diplodactylidae * L1628**
15. Bayesian modeling Phrynosomatidae * L1673**
16. Bayesian modeling Tropiduridae * L1818**
17. Evolutionary rate analyses * L1929**

```
#1. Data Preparation
```

```
```${r Load libraries and dataset, and prepare the data for the phylogenetic analyses}
```

```
require(ape);require(phytools);require(geiger);require(phyloilm);require(phangorn);require(plotrix)
)
require(MCMCglmm);require(MuMIn);require(ggplot2);require(readxl);require(interactions);require(OUwie)
)
require(strap);require(viridis);require(ggtree);require(dplyr);require(deeptime);require(ggpubr)
)
```

```
#setwd("2022-04-12")
```

```
squamata_tree <- read.nexus("AppendixS3.txt") #load tree
squamata_data <- read_excel("Squamata_Data_Analysis.xlsx") #load dataset
```

```
squamata_data$animal <- squamata_data$Species.Zheng.Wiens #specify animal column
squamata_data <- squamata_data %>% select(animal, everything()) #move animal column to first position
```

```
Convert phylo object to ultrametric
```

```
ult.nnls <- force.ultrametric(squamata_tree, method=c("nnls","extend")) #we used this not as a statistical method to infer the ultrametric tree but as a numerical precision correction as indicated in the R documentation: https://www.rdocumentation.org/packages/phytools/versions/1.2-0/topics/force.ultrametric
is.ultrametric(ult.nnls)
```

```
Match data to tree & Map trait on tree
```

```
squamata_data_ZengWiens <- squamata_data #duplicate dataset to keep the original
squamata_data_ZengWiens <-
squamata_data_ZengWiens[which(squamata_data_ZengWiens$animal != "NA"),]
#remove NAs from species list
squamata<- data.frame(as.list(squamata_data_ZengWiens))
rownames(squamata) <-squamata[,1]
row.names(squamata)
names(squamata)
```

```
squamata_noDorsNa <- squamata[which(squamata$Dorsal != "NA"),] #remove NAs
```

```

from Dorsal Brightness

squamata_tree_red <-
treedata(ult.nnls, squamata_noDorsNa, sort=T, warnings=T) $phy
data_squamata <-
as.data.frame(treedata(squamata_tree_red, squamata_noDorsNa, sort=T, warnings=T) $data
)
name.check(squamata_tree_red, data_squamata) #  OK

data_squamata$Dorsal <- as.numeric(data_squamata$Dorsal) # Make numeric
Dorsal
data_squamata$Log.Body.Mass <- as.numeric(data_squamata$Log.Body.Mass) #
Make numeric Log body mass
data_squamata$Altitude.Low <- as.numeric(data_squamata$Altitude.Low) # Make
numeric Altitude.Low
data_squamata$Altitude.High <- as.numeric(data_squamata$Altitude.High) #
Make numeric Altitude.High
data_squamata$Count <- as.numeric(data_squamata$Count) # Make numeric Count
data_squamata$Altitude <- as.factor(data_squamata$Altitude) # Make factor
Altitude
data_squamata$Habitat <- as.factor(data_squamata$Habitat) # Make factor
Habitat
data_squamata$Camouflage <- as.factor(data_squamata$Camouflage) # Make
factor Camouflage
data_squamata$Distribution <- as.factor(data_squamata$Distribution) # Make
factor Distribution
data_squamata$Polymorphic <- as.factor(data_squamata$Polymorphic) # Make
factor Polymorphic
data_squamata$Activity.Pattern <- as.factor(data_squamata$Activity.Pattern)
Make factor Activity.Pattern

dorsaltrait <- setNames(data_squamata$Dorsal, data_squamata$animal) #GIVES
TRAIT TO MAP (Dorsal Brightness)
sizetrait <- setNames(data_squamata$Log.Body.Mass, data_squamata$animal)
#GIVES TRAIT TO MAP (Log Body Mass)
habitatrait <- as.factor(setNames(data_squamata$Habitat,
data_squamata$animal)) #GIVES TRAIT TO MAP (Habitat)

...

```{COMMENT 1}
## In the "data_squamata" object, the Habitat has a factor called
"OpenClosed", and one "OpenClosed:".
## Is that normal ? Or maybe a compatibility problem ? I just ran your
script above, without touching the data of the repo.
levels(data_squamata$Habitat)
# [1] "Closed"      "Fossorial"    "Open"         "OpenClosed"  "OpenClosed:"
```

#2. Ancestral State Reconstruction
```{Ancestral state reconstructions and data visualization}

phylogig(squamata_tree_red, dorsaltrait, method="lambda") # -> 0.751224

fitBM<-fitContinuous(squamata_tree_red,dorsaltrait)
fitOU<-fitContinuous(squamata_tree_red,dorsaltrait,model="OU")
fitEB<-fitContinuous(squamata_tree_red,dorsaltrait,model="EB")

aic.vals<-setNames(c(fitBM$opt$aicc,fitOU$opt$aicc,fitEB$opt$aicc),
c("BM","OU","EB"))
aic.vals
aic.w(aic.vals) #OU best supported

fit <- phytools::fastAnc(squamata_tree_red,dorsaltrait,model="OU")
td <- data.frame(node = nodeid(squamata_tree_red, names(dorsaltrait)),
trait = dorsaltrait)
nd <- data.frame(node = names(fit$ace), trait = fit$ace)

d <- rbind(td, nd)
d$node <- as.numeric(d$node)
tree <- full_join(squamata_tree_red, d, by = 'node')

p1 <- ggtree(tree, aes(color=trait), layout = 'circular',
ladderize = FALSE, continuous = TRUE, size=0.8, open.angle =
10) +
scale_color_gradientn(colours=c("black","grey","ivory"),name="Dorsal
\nBrightness (%)")

```

```

heatmapData <- as.data.frame(sapply(data_squamata$Habitat, as.character))
rownames(heatmapData) <- rownames(data_squamata)

p2 <- gheatmap(p1, heatmapData, offset = 2, color=NULL,
              colnames_position="top",
              colnames_angle=90, colnames_offset_y = 0.0001,
              hjust=0.00001, font.size=0.00001, width = 0.05)+

scale_fill_manual(values=viridis::viridis(4)[c(4,3,1,2)],breaks=c("Open","OpenClosed","Closed","Fossorial"),nam
e = "Vegetation Cover")

p3 <- p2 +
  geom_cladelabel("Serpentes",node=findMRCA(squamata_tree_red,
c("Bitis_arietans","Ophiophagus_hannah")),
  offset=40, offset.text=10) +
  geom_cladelabel("Varanidae",node=findMRCA(squamata_tree_red,
c("Varanus_storri","Varanus_exanthematicus")),
  offset=25, offset.text=10) +
  geom_cladelabel("Anguimorpha",node=findMRCA(squamata_tree_red,
c("Varanus_storri","Diploglossus_bilobatus")),
  offset=40, offset.text=10) +
  geom_cladelabel("Lacertidae",node=findMRCA(squamata_tree_red,
c("Darevskia_bendimahiensis","Gallotia_stehlini")),
  offset=25, offset.text=10) +
  geom_cladelabel("Teoidea",node=findMRCA(squamata_tree_red,
c("Gymnophthalmus_cryptus","Salvator_duseni")),
  offset=40, offset.text=10) +
  geom_cladelabel("Scincidae",node=findMRCA(squamata_tree_red,
c("Acontias_litoralis","Androngo_trivittatus")),
  offset=25, offset.text=10) +
  geom_cladelabel("Gekkota",node=findMRCA(squamata_tree_red,
c("Diplodactylus_conspicillatus","Phyllurus_kabikabi")),
  offset=40, offset.text=10) +
  geom_cladelabel("Diplodactylidae",node=findMRCA(squamata_tree_red,
c("Diplodactylus_pulcher","Pseudothecadactylus_lindneri")),
  offset=25, offset.text=10) +
  geom_cladelabel("Cordyloidea",node=findMRCA(squamata_tree_red,
c("Gerrhosaurus_nigrolineatus","Cordylus_beraduccii")),
  offset=40, offset.text=10) +
  geom_cladelabel("Cordylidae",node=findMRCA(squamata_tree_red,
c("Platysaurus_mitchelli","Cordylus_cordylus")),
  offset=25, offset.text=10) +
  geom_cladelabel("Iguania",node=findMRCA(squamata_tree_red,
c("Leiocephalus_barahonensis","Tropidurus_callathelys")),
  offset=40, offset.text=10) +
  geom_cladelabel("Viperidae",node=findMRCA(squamata_tree_red,
c("Bitis_arietans","Azemiops_feae")),
  offset=25, offset.text=10) +
  geom_cladelabel("Elapidae",node=findMRCA(squamata_tree_red,
c("Hemiaspis_signata","Calliophis_bivirgata")),
  offset=25, offset.text=10) +
  geom_cladelabel("Phrynosomatidae",node=findMRCA(squamata_tree_red,
c("Sceloporus_cyanogenys","Uma_exsul")),
  offset=25, offset.text=10) +
  geom_cladelabel("Tropiduridae",node=findMRCA(squamata_tree_red,
c("Tropidurus_hispidus","Stenocercus_ochoi")),
  offset=25, offset.text=10)

plot(p3)
...

```{COMMENT 2}
library(phylolm)
Errors should be included in a fit.
fit_BM <- phylolm(dorsaltrait ~ 1, phy = squamata_tree_red, model = "BM")

```

```

fit_BM_err <- phylolm(dorsaltrait ~ 1, phy = squamata_tree_red, model =
"BM", measurement_error = TRUE)
fit_OU <- phylolm(dorsaltrait ~ 1, phy = squamata_tree_red, model =
"OUfixedRoot")
fit_OU_err <- phylolm(dorsaltrait ~ 1, phy = squamata_tree_red, model =
"OUfixedRoot", measurement_error = TRUE)

aic.vals_lm<-setNames(c(fit_BMaic,fit_BM_erraic,
fit_OUaic,fit_OU_erraic),
c("BM","BM err","OU", "OU err"))
aic.vals_lm
aic.w(aic.vals_lm)
OU with errors best supported

phytools::fastAnc fro mphytools 2.1-1 does not support the OU model.
It always uses the BM, and the "model" argument is ignored.
Did you use an other version ?
fit <- phytools::fastAnc(squamata_tree_red,dorsaltrait,model="OU")
fit2 <- phytools::fastAnc(squamata_tree_red,dorsaltrait,model="BM")
all.equal(fit, fit2)
\`

#3. Evaluation Altitude Categories
\`{Evaluation altitude categories by data visualization}

#Check the clustering
squamata_data_elev <- squamata_data[which(squamata_data$Altitude !=
"Unknown"),]
squamata_data_elev <- squamata_data_elev[which(squamata_data_elev$Dorsal !=
"NA"),] #remove NAs from Dorsal Brightness

squamata_data_elev$Altitude.Low <-
as.numeric(squamata_data_elev$Altitude.Low)
squamata_data_elev$Altitude.High <-
as.numeric(squamata_data_elev$Altitude.High)

ggplot(data=squamata_data_elev,
aes(x=Altitude.Low,y=Altitude.High,color=Altitude))+geom_point()+
theme_bw()+ xlab("Minimum Altitude (m)") + ylab("Maximum Altitude (m)")
\`

#4. Bayesian modeling at squamate level
\`{MCMCglmm, model selections, and interaction plots for the entire
phylogeny}

Each of these models require several hours to run, please be patient
Alternatively, you can load the saved environment
load(file='TotalEnv.RData')

Remove "Temperate-Polar" distribution: only two entries in the dataset
data_squamata_NoPolar <- data_squamata[which(data_squamata$Distribution !=
"Temperate-Polar"),]
data_squamata_NoPolar <- droplevels(data_squamata_NoPolar)

Make MCMCglmm call updateable so that we can use the dredge function to
change model compositions
MCMCglmm.updateable<- updateable(MCMCglmm)

#Let's first evaluate which Altitude parameter performs better

With categories (Altitude)

data_squamata_NoPolarNa <-
data_squamata_NoPolar[which(data_squamata_NoPolar$Altitude.Low != "NA"),]
#remove NAs from Altitude.Low
data_squamata_NoPolarNa <-
data_squamata_NoPolarNa[which(data_squamata_NoPolarNa$Altitude.High !=
"NA"),] #remove NAs from Altitude.High
data_squamata_NoPolarNa <-
data_squamata_NoPolarNa[which(data_squamata_NoPolarNa$Altitude.Diff !=
"NA"),] #remove NAs from Altitude.Diff
data_squamata_NoPolarNa <- droplevels(data_squamata_NoPolarNa)

start.time3 <- Sys.time()
global.model_dorsal_noPolar_alt<- MCMCglmm.updateable(Dorsal ~ Habitat +
Altitude + Distribution +
Activity.Pattern +
Log.Body.Mass +

```

```

Habitat:Camouflage,
random = ~

Polymorphic,

squamata_tree_red,

burnin = 40000,
nitt = 100000,
thin = 20,
singular.ok=TRUE,
pr=TRUE,
verbose = F,
family = "gaussian",
data =

data_squamata_NoPolarNa)
end.time3 <- Sys.time()
time.taken3 <- end.time3 - start.time3
time.taken3 # 25.15649 secs

With continuous data (Altitude.Low + Altitude.High + Altitude.Diff)

start.time2 <- Sys.time()
global.model_dorsal_noPolar_alt.cont<- MCMCglmm.updateable(Dorsal ~ Habitat
+ Altitude.Low +

Distribution +

Altitude.High +

Log.Body.Mass +

Altitude.Diff +

+ Habitat:Camouflage,

Activity.Pattern

Polymorphic,

random = ~

squamata_tree_red,

pedigree =

burnin = 40000,
nitt = 100000,
thin = 20,

singular.ok=TRUE,

pr=TRUE,
verbose = F,
family =

"gaussian",

data =

data_squamata_NoPolarNa)
end.time2 <- Sys.time()
time.taken2 <- end.time2 - start.time2
time.taken2 # 51.26037 secs

global.model_dorsal_noPolar_alt$DIC #Better with categories, DIC:8052.049
global.model_dorsal_noPolar_alt.cont$DIC #Worse with numerical values,
DIC:8512.521

As the model with altitudinal categorical data performs better, we are
going to use this classification in our models ###

But first, we are going to evaluate which model composition performs
better ###

start.time <- Sys.time()
global.model_dorsal_noPolar<- MCMCglmm.updateable(Dorsal ~
Habitat*Altitude*Distribution*Log.Body.Mass+

Habitat*Activity.Pattern+

Habitat:Camouflage,
random = ~ Polymorphic,
pedigree =

squamata_tree_red,

burnin = 40000,
nitt = 100000,
thin = 20,
singular.ok=TRUE,
pr=TRUE,
verbose = F,
family = "gaussian",
data =

data_squamata_NoPolar)
end.time <- Sys.time()
time.taken <- end.time - start.time
time.taken # 1.443197 mins

start.time.red1 <- Sys.time()
global.model_dorsal_noPolar_1<- MCMCglmm.updateable(Dorsal ~

```

```

Habitat*Altitude*Distribution*Log.Body.Mass+

Habitat*Activity.Pattern+
 Habitat:Camouflage,
 random = ~ Polymorphic,
 pedigree =

squamata_tree_red,
 burnin = 40000,
 nitt = 100000,
 thin = 20,
 singular.ok=TRUE,
 pr=TRUE,
 verbose = F,
 family = "gaussian",
 data =

data_squamata_NoPolar)
end.time.red1 <- Sys.time()
time.taken.red1 <- end.time.red1 - start.time.red1
time.taken.red1 # 1.444705 mins

start.time.red2 <- Sys.time()
global.model_dorsal_noPolar_2<- MCMCglmm.updateable(Dorsal ~
Habitat*Altitude+
 Habitat*Distribution+

Altitude*Distribution*Log.Body.Mass+

Habitat*Activity.Pattern+
 Habitat:Camouflage,
 random = ~ Polymorphic,
 pedigree =

squamata_tree_red,
 burnin = 40000,
 nitt = 100000,
 thin = 20,
 singular.ok=TRUE,
 pr=TRUE,
 verbose = F,
 family = "gaussian",
 data =

data_squamata_NoPolar)
end.time.red2 <- Sys.time()
time.taken.red2 <- end.time.red2 - start.time.red2
time.taken.red2 # 56.20218 secs

start.time.red3 <- Sys.time()
global.model_dorsal_noPolar_3<- MCMCglmm.updateable(Dorsal ~
Habitat*Altitude+
 Habitat*Distribution+

Altitude*Log.Body.Mass+

Distribution*Log.Body.Mass+

Habitat*Activity.Pattern+
 Habitat:Camouflage,
 random = ~ Polymorphic,
 pedigree =

squamata_tree_red,
 burnin = 40000,
 nitt = 100000,
 thin = 20,
 singular.ok=TRUE,
 pr=TRUE,
 verbose = F,
 family = "gaussian",
 data =

data_squamata_NoPolar)
end.time.red3 <- Sys.time()
time.taken.red3 <- end.time.red3 - start.time.red3
time.taken.red3 # 44.99293 secs

start.time.red4 <- Sys.time()
global.model_dorsal_noPolar_4<- MCMCglmm.updateable(Dorsal ~
Habitat*Altitude+
 Habitat*Distribution+

Altitude*Log.Body.Mass+

Distribution*Log.Body.Mass+

Altitude*Distribution+

```

```

Habitat*Activity.Pattern+
 Habitat:Camouflage,
 random = ~ Polymorphic,
 pedigree =

squamata_tree_red,

 burnin = 40000,
 nitt = 100000,
 thin = 20,
 singular.ok=TRUE,
 pr=TRUE,
 verbose = F,
 family = "gaussian",
 data =

data_squamata_NoPolar)
end.time.red4<- Sys.time()
time.taken.red4 <- end.time.red4 - start.time.red4
time.taken.red4 # 50.40567 secs

start.time.red5 <- Sys.time()
global.model_dorsal_noPolar_5<- MCMCglmm.updateable(Dorsal ~
Habitat*Altitude+
 Habitat*Distribution+

Altitude*Log.Body.Mass+

Distribution*Log.Body.Mass+

Habitat*Log.Body.Mass+

Habitat*Activity.Pattern+
 Habitat:Camouflage,
 random = ~ Polymorphic,
 pedigree =

squamata_tree_red,

 burnin = 40000,
 nitt = 100000,
 thin = 20,
 singular.ok=TRUE,
 pr=TRUE,
 verbose = F,
 family = "gaussian",
 data =

data_squamata_NoPolar)
end.time.red5 <- Sys.time()
time.taken.red5 <- end.time.red5 - start.time.red5
time.taken.red5 # 47.70871 secs

DIC(global.model_dorsal_noPolar, global.model_dorsal_noPolar_1,
global.model_dorsal_noPolar_2, global.model_dorsal_noPolar_3,
global.model_dorsal_noPolar_4, global.model_dorsal_noPolar_5) #model 3
(DIC: 8533.273) and 5 (DIC: 8536.888) perform equally well

We selected model 5 over model 3, because model 5 presents an
additional interaction effect that can provide further biological insights
###

Now we can dredge the model and evaluate which parameters are
significantly influencing dorsal brightness ###

start.time01 <- Sys.time()
dredge.MCMCglmm_dorsal_noPolar<- dredge(global.model_dorsal_noPolar_5,
rank="DIC")
end.time01 <- Sys.time()
time.taken01 <- end.time01 - start.time01
time.taken01 # 2.973191 hours

dorsal_noPolar<- subset(dredge.MCMCglmm_dorsal_noPolar,DIC<8529.8) # DIC <5
Several models are showing a variation of DIC <5. Variables are strongly
supported
if they are present in all mostly supported models (DIC <5) and have a
cumulative Akaike weight of > 0.75.
Less strongly supported if they are present in any of the mostly
supported models (DIC <5) and have a
cumulative Akaike of > 0.75.
dorsal_noPolar
sw(dredge.MCMCglmm_dorsal_noPolar) # Akaike weights

Habitat, Activity Pattern, Altitude, Distribution, Log Body Mass,
Camouflage:Habitat, and Altitude:Habitat strongly supported.
Activity Pattern:Habitat less strongly supported.

```

```

#Habitat
model_noPolar_habitat <- MCMCglmm(Dorsal ~ Habitat,
 pedigree = squamata_tree_red,
 burnin = 40000,
 nitt = 100000,
 singular.ok=TRUE,
 thin = 20,
 pr=TRUE,
 verbose = F,
 family = "gaussian",
 data = data_squamata_NoPolar)

#Activity Pattern
model_noPolar_activity <- MCMCglmm(Dorsal ~ Activity.Pattern,
 pedigree = squamata_tree_red,
 burnin = 40000,
 nitt = 100000,
 singular.ok=TRUE,
 thin = 20,
 pr=TRUE,
 verbose = F,
 family = "gaussian",
 data = data_squamata_NoPolar)

#Altitude
model_noPolar_altitude <- MCMCglmm(Dorsal ~ Altitude,
 pedigree = squamata_tree_red,
 burnin = 40000,
 nitt = 100000,
 singular.ok=TRUE,
 thin = 20,
 pr=TRUE,
 verbose = F,
 family = "gaussian",
 data = data_squamata_NoPolar)

#Distribution
model_noPolar_distribution <- MCMCglmm(Dorsal ~ Distribution,
 pedigree = squamata_tree_red,
 burnin = 40000,
 nitt = 100000,
 singular.ok=TRUE,
 thin = 20,
 pr=TRUE,
 verbose = F,
 family = "gaussian",
 data = data_squamata_NoPolar)

#Log Body Mass
model_noPolar_mass <- MCMCglmm(Dorsal ~ Log.Body.Mass,
 pedigree = squamata_tree_red,
 burnin = 40000,
 nitt = 100000,
 singular.ok=TRUE,
 thin = 20,
 pr=TRUE,
 verbose = F,
 family = "gaussian",
 data = data_squamata_NoPolar)

#Camouflage*Habitat
model_noPolar_cam.hab <- MCMCglmm(Dorsal ~ Camouflage*Habitat,
 pedigree = squamata_tree_red,
 burnin = 40000,
 nitt = 100000,
 singular.ok=TRUE,
 thin = 20,
 pr=TRUE,
 verbose = F,
 family = "gaussian",
 data = data_squamata_NoPolar)

#Altitude*Habitat
model_noPolar_alt.hab <- MCMCglmm(Dorsal ~ Altitude*Habitat,
 pedigree = squamata_tree_red,
 burnin = 40000,
 nitt = 100000,
 singular.ok=TRUE,
 thin = 20,
 pr=TRUE,
 verbose = F,
 family = "gaussian",

```

```

data = data_squamata_NoPolar)

#Create dataset (categorical variables) to plug in the predict function
newish_habitat <- expand.grid(Dorsal=rep(0, times=500),

Habitat=factor(c("Open", "OpenClosed", "Closed", "Fossorial")))
#newish_camouflage <- expand.grid(Dorsal=rep(0, times=500),
#
Camouflage=factor(c("Conspicuous", "Cryptic", "Uncertain")))
newish_activity <- expand.grid(Dorsal=rep(0, times=500),

Activity.Pattern=factor(c("Diurnal", "Nocturnal", "Cathemeral", "Unknown")))
newish_mass <- expand.grid(Dorsal=rep(0, times=200),
 Log.Body.Mass=runif(200,
min=min(data_squamata_NoPolar$Log.Body.Mass),

max=max(data_squamata_NoPolar$Log.Body.Mass)))
newish_altitude <- expand.grid(Dorsal=rep(0, times=500),
 Altitude=factor(c("Low", "High", "Low-Medium", "Medium-
High", "All", "Unknown")))
newish_distribution <- expand.grid(Dorsal=rep(0, times=500),
 Distribution=factor(c("Subtropical", "Subtropical-
Temperate", "Temperate",
"Subtropical",
"Temperate", "Tropical", "Tropical-
Subtropical",
"Temperate")))
newish_alt.hab <- expand.grid(Dorsal=rep(0, times=500),

Habitat=factor(c("Open", "OpenClosed", "Closed", "Fossorial")),
 Altitude=factor(c("Low", "High", "Low-Medium", "Medium-
High", "All", "Unknown")))
newish_cam.hab <- expand.grid(Dorsal=rep(0, times=500),

Habitat=factor(c("Open", "OpenClosed", "Closed", "Fossorial")),

Camouflage=factor(c("Conspicuous", "Cryptic", "Uncertain")))

#predict values of dependent variable for each combination based on model
output
m.pred_habitat<-predict(model_noPolar_habitat,type="response", se.fit=TRUE,
interval="confidence",allow.new.levels=TRUE,
 newdata=newish_habitat)
m.pred_activity<-predict(model_noPolar_activity,type="response",
se.fit=TRUE, interval="confidence",allow.new.levels=TRUE,
 newdata=newish_activity)
m.pred_mass<-predict(model_noPolar_mass,type="response", se.fit=TRUE,
interval="confidence",allow.new.levels=TRUE,
 newdata=newish_mass)
m.pred_altitude<-predict(model_noPolar_altitude,type="response",
se.fit=TRUE, interval="confidence",allow.new.levels=TRUE,
 newdata=newish_altitude)
m.pred_distribution<-predict(model_noPolar_distribution,type="response",
se.fit=TRUE, interval="confidence",allow.new.levels=TRUE,
 newdata=newish_distribution)
m.pred_alt.hab<-predict(model_noPolar_alt.hab,type="response", se.fit=TRUE,
interval="confidence",allow.new.levels=TRUE,
 newdata=newish_alt.hab)
m.pred_cam.hab<-predict(model_noPolar_cam.hab,type="response", se.fit=TRUE,
interval="confidence",allow.new.levels=TRUE,
 newdata=newish_cam.hab)

fitted_habitat <- cbind(newish_habitat,m.pred_habitat) # combine the
created dataset with the predicted values
fitted_activity <- cbind(newish_activity,m.pred_activity) # combine the
created dataset with the predicted values
fitted_mass <- cbind(newish_mass,m.pred_mass) # combine the created dataset
with the predicted values
fitted_altitude <- cbind(newish_altitude,m.pred_altitude) # combine the
created dataset with the predicted values
fitted_distribution <- cbind(newish_distribution,m.pred_distribution) #
combine the created dataset with the predicted values
fitted_cam.hab <- cbind(newish_cam.hab,m.pred_cam.hab) # combine the
created dataset with the predicted values
fitted_alt.hab <- cbind(newish_alt.hab,m.pred_alt.hab) # combine the
created dataset with the predicted values

model_fit1<-lm(fit~Habitat,fitted_habitat)
model_fit2<-lm(fit~Activity.Pattern,fitted_activity)
model_fit3<-lm(fit~Log.Body.Mass,fitted mass)

```

```

model_fit4<-lm(fit~Distribution,fitted_distribution)
model_fit5<-lm(fit~Altitude,fitted_altitude)
model_fit7<-lm(fit~Camouflage*Habitat,fitted_cam.hab)
model_fit8<-lm(fit~Altitude*Habitat,fitted_alt.hab)

fit~Habitat
fitted_habitat$Habitat <- factor(fitted_habitat$Habitat,levels = c("Open",
"OpenClosed", "Closed", "Fossorial"))
ggplot(fitted_habitat,aes(x=Habitat,y=fit,ymin=lwr,ymax=upr)) +
 geom_point() +
 geom_errorbar()+theme_bw() +
 theme(panel.grid.minor.x=element_blank(),
 panel.grid.major.x=element_blank()) +
 guides(fill=FALSE) +
 scale_y_continuous(name="Dorsal brightness: fitted values (%)")+
 scale_x_discrete(name="Vegetation")

fit~Activity.Pattern
fitted_activity$Activity.Pattern <-
factor(fitted_activity$Activity.Pattern,levels =
c("Diurnal", "Cathemeral", "Nocturnal", "Unknown"))
ggplot(fitted_activity,aes(x=Activity.Pattern,y=fit,ymin=lwr,ymax=upr)) +
 geom_point() +
 geom_errorbar()+theme_bw() +
 theme(panel.grid.minor.x=element_blank(),
 panel.grid.major.x=element_blank()) +
 guides(fill=FALSE) +
 scale_y_continuous(name="Dorsal brightness: fitted values (%)")+
 scale_x_discrete(name="Circadian rhythm")

fit~Distribution
fitted_distribution$Distribution <-
factor(fitted_distribution$Distribution,levels = c("Tropical", "Tropical-
Subtropical", "Subtropical", "Subtropical-Temperate", "Temperate", "Tropical-
Subtropical-Temperate"))
ggplot(fitted_distribution,aes(x=Distribution,y=fit,ymin=lwr,ymax=upr)) +
 geom_point() +
 geom_errorbar()+theme_bw() +
 theme(panel.grid.minor.x=element_blank(),
 panel.grid.major.x=element_blank()) +
 guides(fill=FALSE) +
 scale_y_continuous(name="Dorsal brightness: fitted values (%)")+
 scale_x_discrete(name="Latitude")

fit~Altitude
fitted_altitude$Altitude <- factor(fitted_altitude$Altitude,levels =
c("Low", "Low-Medium", "Medium-High", "High", "All", "Unknown"))
ggplot(fitted_altitude,aes(x=Altitude,y=fit,ymin=lwr,ymax=upr)) +
 geom_point() +
 geom_errorbar()+theme_bw() +
 theme(panel.grid.minor.x=element_blank(),
 panel.grid.major.x=element_blank()) +
 guides(fill=FALSE) +
 scale_y_continuous(name="Dorsal brightness: fitted values (%)")+
 scale_x_discrete(name="Altitude")

fit~Log.Body.Mass
ggplot(fitted_mass,aes(x=Log.Body.Mass,y=fit,ymin=lwr,ymax=upr)) +
 geom_point() +
 geom_errorbar()+theme_bw() +
 theme(panel.grid.minor.x=element_blank(),
 panel.grid.major.x=element_blank()) +
 guides(fill=FALSE) +
 scale_y_continuous(name="Dorsal brightness: fitted values (%)")+
 scale_x_continuous(name="Log body mass (g)")

fit~Camouflage*Habitat
model_fit7$model$Habitat <- factor(model_fit7$model$Habitat,levels =
c("Open", "OpenClosed", "Closed", "Fossorial"))
r <- cat_plot(model_fit7, pred=Camouflage,
modx=Habitat,plot.points=F,interval=T,
 int.type=c("confidence"),point.size = 0.8, colors = "Dark2",
 y.label="Dorsal Brightness (%)",x.label="Concealment type",
 legend.main="Habitat",geom = "line")

fit~Altitude*Habitat
model_fit8$model$Altitude <- factor(model_fit8$model$Altitude,levels =
c("Low", "Low-Medium", "Medium-High", "High", "All", "Unknown"))
model_fit8$model$Habitat <- factor(model_fit8$model$Habitat,levels =
c("Open", "OpenClosed", "Closed", "Fossorial"))

```

```

s <- cat_plot(model_fit8, pred=Altitude,
modx=Habitat,plot.points=F,interval=T,
 int.type=c("confidence"),point.size = 0.8, colors = "Dark2",
 y.label="Dorsal Brightness (%)",x.label="Altitude",
 legend.main="Habitat",geom = "line")

#Plot predicted significant interactions in one figure
ggarrange(r, s, nrow=1, labels = c("A", "B"),common.legend = T)
...

#5. Squamata level with picture count >= 5
```{picture count >= 5}

data_squamata_NoPolar_subset <- subset(data_squamata_NoPolar, Count>4)
data_squamata_NoPolar_subset <- droplevels(data_squamata_NoPolar_subset)

# Make MCMCglmm call updateable so that we can use the dredge function to
change model compositions
MCMCglmm.updateable<- updateable(MCMCglmm)

start.time.subset <- Sys.time()
global.model_dorsal_noPolar_subset<- MCMCglmm.updateable(Dorsal ~
Habitat*Altitude+
Habitat*Distribution+
Altitude*Log.Body.Mass+
Distribution*Log.Body.Mass+
Habitat*Log.Body.Mass+
Habitat*Activity.Pattern+
Habitat:Camouflage,
Polymorphic,
squamata_tree_red,
random = ~
pedigree =
burnin = 40000,
nitt = 100000,
thin = 20,
singular.ok=TRUE,
pr=TRUE,
verbose = F,
family =
"gaussian",
data =
data_squamata_NoPolar_subset)
end.time.subset <- Sys.time()
time.taken.subset <- end.time.subset - start.time.subset
time.taken.subset # 43.39659 secs

start.time.subset1 <- Sys.time()
dredge.MCMCglmm_dorsal_noPolar_subset<-
dredge(global.model_dorsal_noPolar_subset, rank="DIC")
end.time.subset1 <- Sys.time()
time.taken.subset1 <- end.time.subset1 - start.time.subset1
time.taken.subset1 # 2.494542 hours

dredge.MCMCglmm_dorsal_noPolar_subset #evaluate top models
dorsal_noPolar_subset <-
subset(dredge.MCMCglmm_dorsal_noPolar_subset,DIC<6844.1) # DIC <5
dorsal_noPolar_subset
sw(dredge.MCMCglmm_dorsal_noPolar_subset)
...

#6. Squamata level without fossorial species
```{w/o fossorial species}

data_squamata_NoPolar_nofoss <-
data_squamata_NoPolar[which(data_squamata_NoPolar$Habitat != "Fossorial"),]
data_squamata_NoPolar_nofoss <- droplevels(data_squamata_NoPolar_nofoss)

Make MCMCglmm call updateable so that we can use the dredge function to
change model compositions
MCMCglmm.updateable<- updateable(MCMCglmm)

```

```

start.time.nofoss <- Sys.time()
global.model_dorsal_noPolar_nofoss<- MCMCglmm.updateable(Dorsal ~
Habitat*Altitude+

Habitat*Distribution+

Altitude*Log.Body.Mass+

Distribution*Log.Body.Mass+

Habitat*Log.Body.Mass+

Habitat*Activity.Pattern+

Habitat:Camouflage,

Polymorphic,

squamata_tree_red,

"gaussian",

data_squamata_NoPolar_nofoss)
end.time.nofoss <- Sys.time()
time.taken.nofoss <- end.time.nofoss - start.time.nofoss
time.taken.nofoss # 44.05605 secs

start.time.nofoss1 <- Sys.time()
dredge.MCMCglmm_dorsal_noPolar_nofoss<-
dredge(global.model_dorsal_noPolar_nofoss, rank="DIC")
end.time.nofoss1 <- Sys.time()
time.taken.nofoss1 <- end.time.nofoss1 - start.time.nofoss1
time.taken.nofoss1 # 2.722836 hours

dredge.MCMCglmm_dorsal_noPolar_nofoss #evaluate top models
dorsal_noPolar_nofoss <-
subset(dredge.MCMCglmm_dorsal_noPolar_nofoss,DIC<7866.8) # DIC <5
dorsal_noPolar_nofoss
sw(dredge.MCMCglmm_dorsal_noPolar_nofoss)
...

#7. Squamata level with bootstrapped values
```{bootstrapped values}

#Here we bootstrapped the values of dorsal brightness. We took only species
for which we had 5 or more pictures, we randomly selected 3 pictures of
these, and we extracted the brightness values from these. We repeated the
process 3 times and processed the analyses for each of them.

bootstrap_data <- read_excel("bootstrap_data.xlsx") #load dataset

bootstrap_data$animal <- bootstrap_data$Species.Zheng.Wiens #specify animal
column
bootstrap_data <- bootstrap_data %>% select(animal, everything()) #move
animal column to first position

# Match data to tree & Map trait on tree
bootstrap_data_ZengWiens <- bootstrap_data #duplicate dataset to keep the
original
bootstrap_data_ZengWiens <-
bootstrap_data_ZengWiens[which(bootstrap_data_ZengWiens$animal != "NA"),]
#remove NAs from species list
squamata.bootstrap<- data.frame(as.list(bootstrap_data_ZengWiens))
rownames(squamata.bootstrap) <-squamata.bootstrap[,1]
row.names(squamata.bootstrap)
names(squamata.bootstrap)

squamata.bootstrap_noDors1Na <-
squamata.bootstrap[which(squamata.bootstrap$Dorsal.bootstrap1 != "NA"),]
#remove NAs from Dorsal Brightness (bootstrap 1)
squamata.bootstrap_noDors2Na <-
squamata.bootstrap[which(squamata.bootstrap$Dorsal.bootstrap2 != "NA"),]
#remove NAs from Dorsal Brightness (bootstrap 2)
squamata.bootstrap_noDors3Na <-
squamata.bootstrap[which(squamata.bootstrap$Dorsal.bootstrap3 != "NA"),]

```

```

#remove NAs from Dorsal Brightness (bootstrap 3)

#1
squamata_tree_red_boots1 <-
treedata(ult.nnls,squamata.bootstrap_noDors1Na,sort=T,warnings=T)$phy
data_squamata_boots1 <-
as.data.frame(treedata(squamata_tree_red_boots1,squamata.bootstrap_noDors1Na,sort=T,warnings=T)$data
)
name.check(squamata_tree_red_boots1, data_squamata_boots1) #  OK

#2
squamata_tree_red_boots2 <-
treedata(ult.nnls,squamata.bootstrap_noDors2Na,sort=T,warnings=T)$phy
data_squamata_boots2 <-
as.data.frame(treedata(squamata_tree_red_boots2,squamata.bootstrap_noDors2Na,sort=T,warnings=T)$data
)
name.check(squamata_tree_red_boots2, data_squamata_boots2) #  OK

#3
squamata_tree_red_boots3 <-
treedata(ult.nnls,squamata.bootstrap_noDors3Na,sort=T,warnings=T)$phy
data_squamata_boots3 <-
as.data.frame(treedata(squamata_tree_red_boots3,squamata.bootstrap_noDors3Na,sort=T,warnings=T)$data
)
name.check(squamata_tree_red_boots3, data_squamata_boots3) #  OK

#1
data_squamata_boots1$Dorsal.bootstrap1 <-
as.numeric(data_squamata_boots1$Dorsal.bootstrap1) # Make numeric Dorsal
bootstrap1
data_squamata_boots1$Log.Body.Mass <-
as.numeric(data_squamata_boots1$Log.Body.Mass) # Make numeric Log body mass
data_squamata_boots1$Altitude.Low <-
as.numeric(data_squamata_boots1$Altitude.Low) # Make numeric Altitude.Low
data_squamata_boots1$Altitude.High <-
as.numeric(data_squamata_boots1$Altitude.High) # Make numeric Altitude.High
data_squamata_boots1$Count <- as.numeric(data_squamata_boots1$Count) # Make
numeric Count
data_squamata_boots1$Altitude <- as.factor(data_squamata_boots1$Altitude) #
Make factor Altitude
data_squamata_boots1$Habitat <- as.factor(data_squamata_boots1$Habitat) #
Make factor Habitat
data_squamata_boots1$Camouflage <-
as.factor(data_squamata_boots1$Camouflage) # Make factor Camouflage
data_squamata_boots1$Distribution <-
as.factor(data_squamata_boots1$Distribution) # Make factor Distribution
data_squamata_boots1$Polymorphic <-
as.factor(data_squamata_boots1$Polymorphic) # Make factor Polymorphic
data_squamata_boots1$Activity.Pattern <-
as.factor(data_squamata_boots1$Activity.Pattern) # Make factor
Activity.Pattern

#2
data_squamata_boots2$Dorsal.bootstrap2 <-
as.numeric(data_squamata_boots2$Dorsal.bootstrap2) # Make numeric Dorsal
bootstrap2
data_squamata_boots2$Log.Body.Mass <-
as.numeric(data_squamata_boots2$Log.Body.Mass) # Make numeric Log body mass
data_squamata_boots2$Altitude.Low <-
as.numeric(data_squamata_boots2$Altitude.Low) # Make numeric Altitude.Low
data_squamata_boots2$Altitude.High <-
as.numeric(data_squamata_boots2$Altitude.High) # Make numeric Altitude.High
data_squamata_boots2$Count <- as.numeric(data_squamata_boots2$Count) # Make
numeric Count
data_squamata_boots2$Altitude <- as.factor(data_squamata_boots2$Altitude) #
Make factor Altitude
data_squamata_boots2$Habitat <- as.factor(data_squamata_boots2$Habitat) #
Make factor Habitat
data_squamata_boots2$Camouflage <-
as.factor(data_squamata_boots2$Camouflage) # Make factor Camouflage
data_squamata_boots2$Distribution <-
as.factor(data_squamata_boots2$Distribution) # Make factor Distribution
data_squamata_boots2$Polymorphic <-
as.factor(data_squamata_boots2$Polymorphic) # Make factor Polymorphic
data_squamata_boots2$Activity.Pattern <-
as.factor(data_squamata_boots2$Activity.Pattern) # Make factor
Activity.Pattern

#3
data_squamata_boots3$Dorsal.bootstrap3 <-
as.numeric(data_squamata_boots3$Dorsal.bootstrap3) # Make numeric Dorsal
bootstrap1

```

```

data_squamata_boots3$Log.Body.Mass <-
as.numeric(data_squamata_boots3$Log.Body.Mass) # Make numeric Log body mass
data_squamata_boots3$Altitude.Low <-
as.numeric(data_squamata_boots3$Altitude.Low) # Make numeric Altitude.Low
data_squamata_boots3$Altitude.High <-
as.numeric(data_squamata_boots3$Altitude.High) # Make numeric Altitude.High
data_squamata_boots3$Count <- as.numeric(data_squamata_boots3$Count) # Make
numeric Count
data_squamata_boots3$Altitude <- as.factor(data_squamata_boots3$Altitude) #
Make factor Altitude
data_squamata_boots3$Habitat <- as.factor(data_squamata_boots3$Habitat) #
Make factor Habitat
data_squamata_boots3$Camouflage <-
as.factor(data_squamata_boots3$Camouflage) # Make factor Camouflage
data_squamata_boots3$Distribution <-
as.factor(data_squamata_boots3$Distribution) # Make factor Distribution
data_squamata_boots3$Polymorphic <-
as.factor(data_squamata_boots3$Polymorphic) # Make factor Polymorphic
data_squamata_boots3$Activity.Pattern <-
as.factor(data_squamata_boots3$Activity.Pattern) # Make factor
Activity.Pattern

#1
data_squamata_boots1_NoPolar <-
data_squamata_boots1[which(data_squamata_boots1$Distribution != "Temperate-
Polar"),]
data_squamata_boots1_NoPolar <- droplevels(data_squamata_boots1_NoPolar)

#2
data_squamata_boots2_NoPolar <-
data_squamata_boots2[which(data_squamata_boots2$Distribution != "Temperate-
Polar"),]
data_squamata_boots2_NoPolar <- droplevels(data_squamata_boots2_NoPolar)

#3
data_squamata_boots3_NoPolar <-
data_squamata_boots3[which(data_squamata_boots3$Distribution != "Temperate-
Polar"),]
data_squamata_boots3_NoPolar <- droplevels(data_squamata_boots3_NoPolar)

# Make MCMCglmm call updateable so that we can use the dredge function to
change model compositions
MCMCglmm.updateable<- updateable(MCMCglmm)

#1
start.time.boots1 <- Sys.time()
global.model_boots1<- MCMCglmm.updateable(Dorsal.bootstrap1 ~
Habitat*Altitude+

Habitat*Distribution+

Altitude*Log.Body.Mass+

Distribution*Log.Body.Mass+

Habitat*Log.Body.Mass+

Habitat*Activity.Pattern+

Habitat:Camouflage,

Polymorphic,

squamata_tree_red,

"gaussian",

random = ~

pedigree =

burnin = 40000,
nitt = 100000,
thin = 20,
singular.ok=TRUE,
pr=TRUE,
verbose = F,
family =

data =

end.time.boots1 <- Sys.time()
time.taken.boots1 <- end.time.boots1 - start.time.boots1
time.taken.boots1 # 1.060965 mins

start.time.boots1.1 <- Sys.time()
dredge.MCMCglmm_boots1<- dredge(global.model_boots1, rank="DIC")
end.time.boots1.1 <- Sys.time()
time.taken.boots1.1 <- end.time.boots1.1 - start.time.boots1.1

```

```

data =
data_squamata_boots3_NoPolar)
end.time.boots3 <- Sys.time()
time.taken.boots3 <- end.time.boots3 - start.time.boots3
time.taken.boots3 # 55.28211 secs

start.time.boots3.3 <- Sys.time()
dredge.MCMCglmm_boots3<- dredge(global.model_boots3, rank="DIC")
end.time.boots3.3 <- Sys.time()
time.taken.boots3.3 <- end.time.boots3.3 - start.time.boots3.3
time.taken.boots3.3 # 3.345041 hours

dredge.MCMCglmm_boots3 #evaluate top models
boots3_top <- subset(dredge.MCMCglmm_boots3,DIC<7191.0) # DIC <5
boots3_top
sw(dredge.MCMCglmm_boots3)
...

#8. Varanidae
```{Varanidae}

data_squamata_Varidae <- data_squamata[which(data_squamata$Family ==
"Varidae"),]
#data_squamata_Varidae <-
data_squamata_Varidae[which(data_squamata_Varidae$Distribution !=
"Subtropical-Temperate"),] # only one entry for Subtropical-Temperate
data_squamata_Varidae <- droplevels(data_squamata_Varidae)

Make MCMCglmm call updateable so that we can use the dredge function to
change model compositions
MCMCglmm.updateable<- updateable(MCMCglmm)

start.time.Var <- Sys.time()
global.model_dorsal_Varidae<- MCMCglmm.updateable(Dorsal ~
Habitat*Altitude+
Habitat*Distribution+
Altitude*Log.Body.Mass+
Distribution*Log.Body.Mass+
Habitat*Log.Body.Mass+
Habitat*Activity.Pattern+
Habitat:Camouflage,
random = ~ Polymorphic,
pedigree =
squamata_tree_red,
burnin = 40000,
nitt = 100000,
thin = 20,
singular.ok=TRUE,
pr=TRUE,
verbose = F,
family = "gaussian",
data =

data_squamata_Varidae)
end.time.Var <- Sys.time()
time.taken.Var <- end.time.Var - start.time.Var
time.taken.Var # 4.704553 secs

start.time.Var1 <- Sys.time()
dredge.MCMCglmm_dorsal_Varidae<- dredge(global.model_dorsal_Varidae,
rank="DIC")
end.time.Var1 <- Sys.time()
time.taken.Var1 <- end.time.Var1 - start.time.Var1
time.taken.Var1 # 19.25191 mins

dredge.MCMCglmm_dorsal_Varidae # evaluate top models
dorsal_Varidae <- subset(dredge.MCMCglmm_dorsal_Varidae,DIC<353.9) #
DIC <5
dorsal_Varidae
sw(dredge.MCMCglmm_dorsal_Varidae)
...

#9. Viperidae
```{Viperidae}

```

```

data_squamata_Viperidae <- data_squamata[which(data_squamata$Family ==
"Viperidae"),]
data_squamata_Viperidae <- droplevels(data_squamata_Viperidae)

# Make MCMCglmm call updateable so that we can use the dredge function to
change model compositions
MCMCglmm.updateable<- updateable(MCMCglmm)

start.time.Vip <- Sys.time()
global.model_dorsal_Viperidae<- MCMCglmm.updateable(Dorsal ~
Habitat*Altitude+
                                Habitat*Distribution+
                                Altitude*Log.Body.Mass+
                                Distribution*Log.Body.Mass+
                                Habitat*Log.Body.Mass+
                                Habitat*Activity.Pattern+
                                Habitat:Camouflage,
                                random = ~ Polymorphic,
                                pedigree =
                                burnin = 40000,
                                nitt = 100000,
                                thin = 20,
                                singular.ok=TRUE,
                                pr=TRUE,
                                verbose = F,
                                family = "gaussian",
                                data =
                                data_squamata_tree_red,

data_squamata_Viperidae)
end.time.Vip <- Sys.time()
time.taken.Vip <- end.time.Vip - start.time.Vip
time.taken.Vip # 12.26776 secs

start.time.Vip1 <- Sys.time()
dredge.MCMCglmm_dorsal_Viperidae<- dredge(global.model_dorsal_Viperidae,
rank="DIC")
end.time.Vip1 <- Sys.time()
time.taken.Vip1 <- end.time.Vip1 - start.time.Vip1
time.taken.Vip1 # 47.72456 mins

dredge.MCMCglmm_dorsal_Viperidae # evaluate top models
dorsal_Viperidae <- subset(dredge.MCMCglmm_dorsal_Viperidae,DIC<1336.4) #
DIC <5
dorsal_Viperidae
sw(dredge.MCMCglmm_dorsal_Viperidae)

### Viperidae with Alencar phylo ###

data_squamata_viperidae_Alencar <- squamata_data[
which(squamata_data$Family=='Viperidae'), ]
data_squamata_viperidae_Alencar <-
data_squamata_viperidae_Alencar[which(data_squamata_viperidae_Alencar$Distributio
n != "Temperate-Polar"), ]
data_squamata_viperidae_Alencar$animal <-
data_squamata_viperidae_Alencar$Species.Alencar #specify animal column
data_squamata_viperidae_Alencar <- data_squamata_viperidae_Alencar %>%
select(animal, everything()) #move animal column to first position

viperidae_tree <- read.tree("mmc9.nwk") #load tree

# Convert phylo object to ultrametric
ult.nnls_viperidae <- force.ultrametric(viperidae_tree,
method=c("nnls","extend"))
is.ultrametric(ult.nnls_viperidae)

# Match data to tree
data_squamata_viperidae_Alencar <-
data_squamata_viperidae_Alencar[which(data_squamata_viperidae_Alencar$anima
l != "NA"),] #remove NAs from species list
squamata_viperidae<- data.frame(as.list(data_squamata_viperidae_Alencar))
rownames(squamata_viperidae) <-squamata_viperidae[,1]
row.names(squamata_viperidae)
names(squamata_viperidae)

squamata_viperidae_noDorsNa <-
squamata_viperidae[which(squamata_viperidae$Dorsal != "NA"),] #remove NAs
from Dorsal Brightness
squamata_viperidae_noDorsNa <- droplevels(squamata_viperidae_noDorsNa)

```

```

squamata_viperidae_tree_red <-
treedata(ult.nnls_viperidae,squamata_viperidae_noDorsNa,
          sort=T,warnings=T)$phy
data_squamata_viperidae <-
as.data.frame(treedata(squamata_viperidae_tree_red,

squamata_viperidae_noDorsNa,sort=T,warnings=T)$data)
name.check(squamata_viperidae_tree_red, data_squamata_viperidae)

data_squamata_viperidae$Dorsal <-
as.numeric(data_squamata_viperidae$Dorsal) # Make numeric Dorsal
data_squamata_viperidae$Log.Body.Mass <-
as.numeric(data_squamata_viperidae$Log.Body.Mass) # Make numeric Log body
mass
data_squamata_viperidae$Altitude.Low <-
as.numeric(data_squamata_viperidae$Altitude.Low) # Make numeric
Altitude.Low
data_squamata_viperidae$Altitude.High <-
as.numeric(data_squamata_viperidae$Altitude.High) # Make numeric
Altitude.High
data_squamata_viperidae$Count <- as.numeric(data_squamata_viperidae$Count)
# Make numeric Count
data_squamata_viperidae$Altitude <-
as.factor(data_squamata_viperidae$Altitude) # Make factor Altitude
data_squamata_viperidae$Habitat <-
as.factor(data_squamata_viperidae$Habitat) # Make factor Habitat
data_squamata_viperidae$Camouflage <-
as.factor(data_squamata_viperidae$Camouflage) # Make factor Camouflage
data_squamata_viperidae$Distribution <-
as.factor(data_squamata_viperidae$Distribution) # Make factor Distribution
data_squamata_viperidae$Polymorphic <-
as.factor(data_squamata_viperidae$Polymorphic) # Make factor Polymorphic
data_squamata_viperidae$Activity.Pattern <-
as.factor(data_squamata_viperidae$Activity.Pattern) # Make factor
Activity.Pattern

# Make MCMCglmm call updateable so that we can use the dredge function to
change model compositions
MCMCglmm.updateable<- updateable(MCMCglmm)

start.time.VipAlencar <- Sys.time()
global.model_dorsal_viperidae_Alencar<- MCMCglmm.updateable(Dorsal ~
Habitat*Altitude+

Habitat*Distribution+

Altitude*Log.Body.Mass+

Distribution*Log.Body.Mass+

Habitat*Log.Body.Mass+

Habitat*Activity.Pattern+

Habitat:Camouflage,

Polymorphic,

squamata_viperidae_tree_red,

singular.ok=TRUE,

"gaussian",

data_squamata_viperidae)
end.time.VipAlencar <- Sys.time()
time.taken.VipAlencar <- end.time.VipAlencar - start.time.VipAlencar
time.taken.VipAlencar # 16.23449 secs

start.time.VipAlencar1 <- Sys.time()
dredge.MCMCglmm_dorsal_viperidae_Alencar<-
dredge(global.model_dorsal_viperidae_Alencar, rank="DIC")
end.time.VipAlencar1 <- Sys.time()
time.taken.VipAlencar1 <- end.time.VipAlencar1 - start.time.VipAlencar1
time.taken.VipAlencar1 # 55.28007 mins

```

```

dredge.MCMCglmm_dorsal_viperidae_Alencar # evaluate top models
dorsal_viperidae_Alencar <-
subset(dredge.MCMCglmm_dorsal_viperidae_Alencar,DIC<1698.1) # DIC <5
dorsal_viperidae_Alencar
sw(dredge.MCMCglmm_dorsal_viperidae_Alencar)

...

#10. Elapidae
```{Elapidae}

data_squamata_Elapidae <- data_squamata[which(data_squamata$Family ==
"Elapidae"),]
data_squamata_Elapidae <- droplevels(data_squamata_Elapidae)

Make MCMCglmm call updateable so that we can use the dredge function to
change model compositions
MCMCglmm.updateable<- updateable(MCMCglmm)

start.time.Ela <- Sys.time()
global.model_dorsal_Elapidae<- MCMCglmm.updateable(Dorsal ~
Habitat*Altitude+
 Habitat*Distribution+
Altitude*Log.Body.Mass+
Distribution*Log.Body.Mass+
 Habitat*Log.Body.Mass+
Habitat*Activity.Pattern+
 Habitat:Camouflage,
 random = ~ Polymorphic,
 pedigree =
squamata_tree_red,
 burnin = 40000,
 nitt = 100000,
 thin = 20,
 singular.ok=TRUE,
 pr=TRUE,
 verbose = F,
 family = "gaussian",
 data =

data_squamata_Elapidae)
end.time.Ela <- Sys.time()
time.taken.Ela <- end.time.Ela - start.time.Ela
time.taken.Ela # 9.463595 secs

start.time.Elal <- Sys.time()
dredge.MCMCglmm_dorsal_Elapidae<- dredge(global.model_dorsal_Elapidae,
rank="DIC")
end.time.Elal <- Sys.time()
time.taken.Elal <- end.time.Elal - start.time.Elal
time.taken.Elal # 29.62978 mins

dredge.MCMCglmm_dorsal_Elapidae # evaluate top models
dorsal_Elapidae <- subset(dredge.MCMCglmm_dorsal_Elapidae,DIC<727.1) # DIC
<5
dorsal_Elapidae
sw(dredge.MCMCglmm_dorsal_Elapidae)

...

#11. Scincidae
```{Scincidae}

data_squamata_Scincidae <- data_squamata[which(data_squamata$Family ==
"Scincidae"),]
data_squamata_Scincidae <- droplevels(data_squamata_Scincidae)

# Make MCMCglmm call updateable so that we can use the dredge function to
change model compositions
MCMCglmm.updateable<- updateable(MCMCglmm)

start.time.Sci <- Sys.time()
global.model_dorsal_Scincidae<- MCMCglmm.updateable(Dorsal ~
Habitat*Altitude+
                                Habitat*Distribution+
Altitude*Log.Body.Mass+

```

```

Distribution*Log.Body.Mass+
Habitat*Log.Body.Mass+
Habitat*Activity.Pattern+
                                Habitat:Camouflage,
                                random = ~ Polymorphic,
                                pedigree =
squamata_tree_red,
                                burnin = 40000,
                                nitt = 100000,
                                thin = 20,
                                singular.ok=TRUE,
                                pr=TRUE,
                                verbose = F,
                                family = "gaussian",
                                data =

data_squamata_Scincidae)
end.time.Sci <- Sys.time()
time.taken.Sci <- end.time.Sci - start.time.Sci
time.taken.Sci # 10.77959 secs

start.time.Sci1 <- Sys.time()
dredge.MCMCglmm_dorsal_Scincidae<- dredge(global.model_dorsal_Scincidae,
rank="DIC")
end.time.Sci1 <- Sys.time()
time.taken.Sci1 <- end.time.Sci1 - start.time.Sci1
time.taken.Sci1 # 34.59481 mins

dredge.MCMCglmm_dorsal_Scincidae #evaluate top models
dorsal_Scincidae <- subset(dredge.MCMCglmm_dorsal_Scincidae,DIC<840.2) #
DIC <5
dorsal_Scincidae
sw(dredge.MCMCglmm_dorsal_Scincidae)

scincidae_altitude <- MCMCglmm(Dorsal ~ Altitude,
                                pedigree = squamata_tree_red,
                                burnin = 40000,
                                nitt = 100000,
                                singular.ok=TRUE,
                                thin = 20,
                                pr=TRUE,
                                verbose = F,
                                family = "gaussian",
                                data = data_squamata_Scincidae)

newish_scincidae_altitude <- expand.grid(Dorsal=rep(0, times=500),
                                Altitude=factor(c("Low", "High", "Low-Medium", "Medium-
High", "All", "Unknown"))))

m.pred_scincidae_altitude<-predict(scincidae_altitude,type="response",
se.fit=TRUE, interval="confidence",allow.new_levels=TRUE,
                                newdata=newish_scincidae_altitude)

fitted_scincidae_altitude <-
cbind(newish_scincidae_altitude,m.pred_scincidae_altitude) # combine the
created dataset with the predicted values

model_fit10<-lm(fit~Altitude,fitted_scincidae_altitude)

# fit~Altitude
fitted_scincidae_altitude$Altitude <-
factor(fitted_scincidae_altitude$Altitude,levels = c("Low","Low-
Medium","Medium-High","High","All","Unknown"))
ggplot(fitted_scincidae_altitude,aes(x=Altitude,y=fit,ymin=lwr,ymax=upr)) +
geom_point() +
geom_errorbar()+theme_bw() +
theme(panel.grid.minor.x=element_blank(),
panel.grid.major.x=element_blank()) +
guides(fill=FALSE) +
scale_y_continuous(name="Dorsal brightness: fitted values (%)")+
scale_x_discrete(name="Altitude")

scincidae_mass <- MCMCglmm(Dorsal ~ Log.Body.Mass,
                                pedigree = squamata_tree_red,
                                burnin = 40000,
                                nitt = 100000,
                                singular.ok=TRUE,
                                thin = 20,

```

```

        pr=TRUE,
        verbose = F,
        family = "gaussian",
        data = data_squamata_Scincidae)

newish_scincidae_mass <- expand.grid(Dorsal=rep(0, times=200),
                                   Log.Body.Mass=runif(200,
min=min(data_squamata_Scincidae$Log.Body.Mass),
max=max(data_squamata_Scincidae$Log.Body.Mass)))

m.pred_scincidae_mass<-predict(scincidae_mass,type="response", se.fit=TRUE,
interval="confidence",allow.new.levels=TRUE,
                               newdata=newish_scincidae_mass)

fitted_scincidae_mass <- cbind(newish_scincidae_mass,m.pred_scincidae_mass)
# combine the created dataset with the predicted values

model_fit12<-lm(fit~Log.Body.Mass,fitted_scincidae_mass)

# fit~mass
ggplot(fitted_scincidae_mass,aes(x=Log.Body.Mass,y=fit,ymin=lwr,ymax=upr))
+
  geom_point() +
  geom_errorbar()+theme_bw() +
  theme(panel.grid.minor.x=element_blank(),
        panel.grid.major.x=element_blank()) +
  guides(fill=FALSE) +
  scale_y_continuous(name="Dorsal brightness: fitted values (%)")+
  scale_x_continuous(name="Log body mass (g)")

scincidae_althab <- MCMCglmm(Dorsal ~ Log.Body.Mass*Altitude,
                           pedigree = squamata_tree_red,
                           burnin = 40000,
                           nitt = 100000,
                           singular.ok=TRUE,
                           thin = 20,
                           pr=TRUE,
                           verbose = F,
                           family = "gaussian",
                           data = data_squamata_Scincidae)

newish_scincidae_althab <- expand.grid(Dorsal=rep(0, times=100),
                                   Altitude=factor(c("Low", "High", "Low-Medium", "Medium-
High", "All", "Unknown"))),
                                   Log.Body.Mass=runif(100,
min=min(data_squamata_Scincidae$Log.Body.Mass),
max=max(data_squamata_Scincidae$Log.Body.Mass)))

m.pred_scincidae_althab<-predict(scincidae_althab,type="response",
se.fit=TRUE, interval="confidence",allow.new.levels=TRUE,
                               newdata=newish_scincidae_althab)

fitted_scincidae_althab <-
cbind(newish_scincidae_althab,m.pred_scincidae_althab) # combine the
created dataset with the predicted values

model_fit19<-lm(fit~Log.Body.Mass*Altitude,fitted_scincidae_althab)

# fit~Body mass:Altitude
model_fit19$model$Altitude <- factor(model_fit19$model$Altitude,levels =
c("Low", "Low-Medium", "Medium-High", "High", "All", "Unknown"))
scinci_mass_inter <- interact_plot(model_fit19, pred=Log.Body.Mass,
                                  modx=Altitude,plot.points=F,interval=T,
                                  int.type=c("confidence"),point.size = 0.8, colors =
"Greys",
                                  y.label="Dorsal Brightness: fitted values
(%)",x.label="Log body mass (g)",
                                  legend.main="Altitude")

scinci_mass_inter + geom_point(data=data_squamata_Scincidae,
aes(x=Log.Body.Mass,y=Dorsal,color=Altitude),
inherit.aes = FALSE)

...

#12. Lacertidae
```{Lacertidae}

```

```

data_squamata_Lacertidae <- data_squamata[which(data_squamata$Family ==
"Lacertidae"),]
data_squamata_Lacertidae <-
data_squamata_Lacertidae[which(data_squamata_Lacertidae$Distribution !=
"Temperate-Polar"),]
data_squamata_Lacertidae <- droplevels(data_squamata_Lacertidae)

Make MCMCglmm call updateable so that we can use the dredge function to
change model compositions
MCMCglmm.updateable<- updateable(MCMCglmm)

#Only Diurnal species so we remove Activity Pattern from the model
start.time.Lac <- Sys.time()
global.model_dorsal_Lacertidae<- MCMCglmm.updateable(Dorsal ~
Habitat*Altitude+

Habitat*Distribution+

Altitude*Log.Body.Mass+

Distribution*Log.Body.Mass+

Habitat*Log.Body.Mass+

 Habitat:Camouflage,
 random = ~

Polymorphic, pedigree =

squamata_tree_red, burnin = 40000,
 nitt = 100000,
 thin = 20,
 singular.ok=TRUE,
 pr=TRUE,
 verbose = F,
 family = "gaussian",
 data =

data_squamata_Lacertidae)
end.time.Lac <- Sys.time()
time.taken.Lac <- end.time.Lac - start.time.Lac
time.taken.Lac # 10.06897 secs

start.time.Lac1 <- Sys.time()
dredge.MCMCglmm_dorsal_Lacertidae<- dredge(global.model_dorsal_Lacertidae,
rank="DIC")
end.time.Lac1 <- Sys.time()
time.taken.Lac1 <- end.time.Lac1 - start.time.Lac1
time.taken.Lac1 # 13.19641 mins

dredge.MCMCglmm_dorsal_Lacertidae #evaluate top models
dorsal_Lacertidae <- subset(dredge.MCMCglmm_dorsal_Lacertidae,DIC<1209.6) #
DIC <5
dorsal_Lacertidae
sw(dredge.MCMCglmm_dorsal_Lacertidae)

lacertidae_latitude <- MCMCglmm(Dorsal ~ Distribution,
 pedigree = squamata_tree_red,
 burnin = 40000,
 nitt = 100000,
 singular.ok=TRUE,
 thin = 20,
 pr=TRUE,
 verbose = F,
 family = "gaussian",
 data = data_squamata_Lacertidae)

newish_lacertidae_latitude <- expand.grid(Dorsal=rep(0, times=500),
 Distribution=factor(c("Subtropical","Subtropical-
Temperate","Temperate",
 "Tropical", "Tropical-
Subtropical")))

m.pred_lacertidae_latitude<-predict(lacertidae_latitude,type="response",
se.fit=TRUE, interval="confidence",allow.new.levels=TRUE,
 newdata=newish_lacertidae_latitude)

fitted_lacertidae_latitude <-
cbind(newish_lacertidae_latitude,m.pred_lacertidae_latitude) # combine the
created dataset with the predicted values

model_fit15<-lm(fit~Distribution,fitted_lacertidae_latitude)

fit~Latitude

```

```
fitted_lacertidae_latitude$Distribution <-
factor(fitted_lacertidae_latitude$Distribution, levels = c("Tropical",
"Tropical-Subtropical", "Subtropical", "Subtropical-
Temperate", "Temperate", "Tropical-Subtropical-Temperate"))
ggplot(fitted_lacertidae_latitude, aes(x=Distribution, y=fit, ymin=lwr, ymax=upr)
) +
 geom_point() +
 geom_errorbar()+theme_bw() +
 theme(panel.grid.minor.x=element_blank(),
 panel.grid.major.x=element_blank()) +
 guides(fill=FALSE) +
 scale_y_continuous(name="Dorsal brightness: fitted values (%)")+
 scale_x_discrete(name="Latitude")
```

```
lacertidae_altitude <- MCMCglmm(Dorsal ~ Altitude,
 pedigree = squamata_tree_red,
 burnin = 40000,
 nitt = 100000,
 singular.ok=TRUE,
 thin = 20,
 pr=TRUE,
 verbose = F,
 family = "gaussian",
 data = data_squamata_Lacertidae)
```

```
newish_lacertidae_altitude <- expand.grid(Dorsal=rep(0, times=500),
 Altitude=factor(c("Low", "Low-Medium", "Medium-
High", "High", "All")))
```

```
m.pred_lacertidae_altitude<-predict(lacertidae_altitude,type="response",
se.fit=TRUE, interval="confidence", allow.new.levels=TRUE,
newdata=newish_lacertidae_altitude)
```

```
fitted_lacertidae_altitude <-
cbind(newish_lacertidae_altitude,m.pred_lacertidae_altitude) # combine the
created dataset with the predicted values
```

```
model_fit_lacalt<-lm(fit~Altitude,fitted_lacertidae_altitude)
```

```
fit~altitude
fitted_lacertidae_altitude$Altitude <-
factor(fitted_lacertidae_altitude$Altitude, levels = c("Low", "Low-
Medium", "Medium-High", "High", "All"))
ggplot(fitted_lacertidae_altitude, aes(x=Altitude, y=fit, ymin=lwr, ymax=upr))
+
 geom_point() +
 geom_errorbar()+theme_bw() +
 theme(panel.grid.minor.x=element_blank(),
 panel.grid.major.x=element_blank()) +
 guides(fill=FALSE) +
 scale_y_continuous(name="Dorsal brightness: fitted values (%)")+
 scale_x_discrete(name="Altitude")
```

```
lacertidae_habitat <- MCMCglmm(Dorsal ~ Habitat,
 pedigree = squamata_tree_red,
 burnin = 40000,
 nitt = 100000,
 singular.ok=TRUE,
 thin = 20,
 pr=TRUE,
 verbose = F,
 family = "gaussian",
 data = data_squamata_Lacertidae)
```

```
newish_lacertidae_habitat <- expand.grid(Dorsal=rep(0, times=500),
 Habitat=factor(c("Open", "OpenClosed", "Closed")))
```

```
m.pred_lacertidae_habitat<-predict(lacertidae_habitat,type="response",
se.fit=TRUE, interval="confidence", allow.new.levels=TRUE,
newdata=newish_lacertidae_habitat)
```

```
fitted_lacertidae_habitat <-
cbind(newish_lacertidae_habitat,m.pred_lacertidae_habitat) # combine the
created dataset with the predicted values
```

```
model_fit15<-lm(fit~Habitat,fitted_lacertidae_habitat)
```

```
fit~Habitat
fitted_lacertidae_habitat$Habitat <-
```

```

factor(fitted_lacertidae_habitat$Habitat, levels =
c("Open", "OpenClosed", "Closed"))
ggplot(fitted_lacertidae_habitat, aes(x=Habitat, y=fit, ymin=lwr, ymax=upr)) +
 geom_point() +
 geom_errorbar()+theme_bw() +
 theme(panel.grid.minor.x=element_blank(),
 panel.grid.major.x=element_blank()) +
 guides(fill=FALSE) +
 scale_y_continuous(name="Dorsal brightness: fitted values (%)")+
 scale_x_discrete(name="Vegetation")
...

#13. Cordylidae
```{Cordylidae}
data_squamata_Cordylidae <- data_squamata[which(data_squamata$Family ==
"Cordylidae"),]
data_squamata_Cordylidae <- droplevels(data_squamata_Cordylidae)

# Make MCMCglmm call updateable so that we can use the dredge function to
change model compositions
MCMCglmm.updateable<- updateable(MCMCglmm)

#Only Diurnal species so we remove Activity Pattern from the model
start.time.Cor <- Sys.time()
global.model_dorsal_Cordylidae<- MCMCglmm.updateable(Dorsal ~
Habitat*Altitude+
Habitat*Distribution+
Altitude*Log.Body.Mass+
Distribution*Log.Body.Mass+
Habitat*Log.Body.Mass+
Polymorphic,
squamata_tree_red,
                                Habitat:Camouflage,
                                random = ~
                                pedigree =
                                burnin = 40000,
                                nitt = 100000,
                                thin = 20,
                                singular.ok=TRUE,
                                pr=TRUE,
                                verbose = F,
                                family = "gaussian",
                                data =
data_squamata_Cordylidae)
end.time.Cor <- Sys.time()
time.taken.Cor <- end.time.Cor - start.time.Cor
time.taken.Cor # 3.479548 secs

start.time.Cor1 <- Sys.time()
dredge.MCMCglmm_dorsal_Cordylidae<- dredge(global.model_dorsal_Cordylidae,
rank="DIC")
end.time.Cor1 <- Sys.time()
time.taken.Cor1 <- end.time.Cor1 - start.time.Cor1
time.taken.Cor1 # 5.890287 mins

dredge.MCMCglmm_dorsal_Cordylidae #evaluate top models
dorsal_Cordylidae <- subset(dredge.MCMCglmm_dorsal_Cordylidae, DIC<272.6) #
DIC <5
dorsal_Cordylidae
sw(dredge.MCMCglmm_dorsal_Cordylidae)

cordylidae_altitude <- MCMCglmm(Dorsal ~ Altitude,
                                pedigree = squamata_tree_red,
                                burnin = 40000,
                                nitt = 100000,
                                singular.ok=TRUE,
                                thin = 20,
                                pr=TRUE,
                                verbose = F,
                                family = "gaussian",
                                data = data_squamata_Cordylidae)

newish_cordylidae_altitude <- expand.grid(Dorsal=rep(0, times=500),
                                Altitude=factor(c("Low", "High", "Low-Medium", "Medium-
High", "All", "Unknown")))

```

```

m.pred_cordylidae_altitude<-predict(cordylidae_altitude,type="response",
se.fit=TRUE, interval="confidence",allow.new.levels=TRUE,
newdata=newish_cordylidae_altitude)

fitted_cordylidae_altitude <-
cbind(newish_cordylidae_altitude,m.pred_cordylidae_altitude) # combine the
created dataset with the predicted values

model_fit11<-lm(fit~Altitude,fitted_cordylidae_altitude)

# fit~Altitude
fitted_cordylidae_altitude$Altitude <-
factor(fitted_cordylidae_altitude$Altitude,levels = c("Low","Low-
Medium","Medium-High","High","All","Unknown"))
ggplot(fitted_cordylidae_altitude,aes(x=Altitude,y=fit,ymin=lwr,ymax=upr))
+
geom_point() +
geom_errorbar()+theme_bw() +
theme(panel.grid.minor.x=element_blank(),
panel.grid.major.x=element_blank()) +
guides(fill=FALSE) +
scale_y_continuous(name="Dorsal brightness: fitted values (%)")+
scale_x_discrete(name="Altitude")

cordylidae_mass <- MCMCglmm(Dorsal ~ Log.Body.Mass,
pedigree = squamata_tree_red,
burnin = 40000,
nitt = 100000,
singular.ok=TRUE,
thin = 20,
pr=TRUE,
verbose = F,
family = "gaussian",
data = data_squamata_Cordylidae)

newish_cordylidae_mass <- expand.grid(Dorsal=rep(0, times=200),
Log.Body.Mass=runif(200,
min=min(data_squamata_Cordylidae$Log.Body.Mass),
max=max(data_squamata_Cordylidae$Log.Body.Mass)))

m.pred_cordylidae_mass<-predict(cordylidae_mass,type="response",
se.fit=TRUE, interval="confidence",allow.new.levels=TRUE,
newdata=newish_cordylidae_mass)

fitted_cordylidae_mass <-
cbind(newish_cordylidae_mass,m.pred_cordylidae_mass) # combine the created
dataset with the predicted values

model_fit13<-lm(fit~Log.Body.Mass,fitted_cordylidae_mass)

# fit~mass
ggplot(fitted_cordylidae_mass,aes(x=Log.Body.Mass,y=fit,ymin=lwr,ymax=upr))
+
geom_point() +
geom_errorbar()+theme_bw() +
theme(panel.grid.minor.x=element_blank(),
panel.grid.major.x=element_blank()) +
guides(fill=FALSE) +
scale_y_continuous(name="Dorsal brightness: fitted values (%)")+
scale_x_continuous(name="Log body mass (g)")

cordylidae_althab <- MCMCglmm(Dorsal ~ Log.Body.Mass*Altitude,
pedigree = squamata_tree_red,
burnin = 40000,
nitt = 100000,
singular.ok=TRUE,
thin = 20,
pr=TRUE,
verbose = F,
family = "gaussian",
data = data_squamata_Cordylidae)

newish_cordylidae_althab <- expand.grid(Dorsal=rep(0, times=100),
Altitude=factor(c("Low","High","Low-Medium","Medium-
High","All","Unknown")),
Log.Body.Mass=runif(100,
min=min(data_squamata_Cordylidae$Log.Body.Mass),
max=max(data_squamata_Cordylidae$Log.Body.Mass)))

```

```

m.pred_cordylidae_althab<-predict(cordylidae_althab,type="response",
se.fit=TRUE, interval="confidence",allow.new.levels=TRUE,
newdata=newish_cordylidae_althab)

fitted_cordylidae_althab <-
cbind(newish_cordylidae_althab,m.pred_cordylidae_althab) # combine the
created dataset with the predicted values

model_fit20<-lm(fit~Log.Body.Mass*Altitude,fitted_cordylidae_althab)

# fit~Body mass:Altitude
model_fit20$model$Altitude <- factor(model_fit20$model$Altitude,levels =
c("Low","Low-Medium","Medium-High","High","All","Unknown"))
cord_mass_inter <- interact_plot(model_fit19, pred=Log.Body.Mass,
modx=Altitude,plot.points=F,interval=T,
int.type=c("confidence"),point.size = 0.8, colors =
"Greys",
y.label="Dorsal Brightness: fitted values
(%)",x.label="Log body mass (g)",
legend.main="Altitude")

cord_mass_inter + geom_point(data=data_squamata_Cordylidae,
aes(x=Log.Body.Mass,y=Dorsal,color=Altitude),
inherit.aes = FALSE)

#Cordylidae with Stanley phylo

data_squamata_cordylidae_Stanley <- squamata_data[
which(squamata_data$Family=='Cordylidae'), ]
data_squamata_cordylidae_Stanley$animal <-
data_squamata_cordylidae_Stanley$Species.Stanley #specify animal column
data_squamata_cordylidae_Stanley <- data_squamata_cordylidae_Stanley %>%
select(animal, everything()) #move animal column to first position

cordylidae_tree <- read.nexus("Cordylidae.txt") #load tree

# Convert phylo object to ultrametric
#ult.nnls_cordylidae <- force.ultrametric(cordylidae_tree,
method=c("nnls","extend"))
#is.ultrametric(ult.nnls_cordylidae)

# Match data to tree
data_squamata_cordylidae_Stanley <-
data_squamata_cordylidae_Stanley[which(data_squamata_cordylidae_Stanley$anima
l != "NA"),] #remove NAs from species list
squamata_cordylidae<- data.frame(as.list(data_squamata_cordylidae_Stanley))
rownames(squamata_cordylidae) <-squamata_cordylidae[,1]
row.names(squamata_cordylidae)
names(squamata_cordylidae)

squamata_cordylidae_noDorsNa <-
squamata_cordylidae[which(squamata_cordylidae$Dorsal != "NA"),] #remove NAs
from Dorsal Brightness
#squamata_cordylidae_noDorsNa <-
squamata_cordylidae_noDorsNa[which(squamata_cordylidae_noDorsNa$Polymorphic
!= "NA"),] #remove NAs from Polymorphism
squamata_cordylidae_noDorsNa <- droplevels(squamata_cordylidae_noDorsNa)

squamata_cordylidae_tree_red <-
treedata(cordylidae_tree,squamata_cordylidae_noDorsNa,
sort=T,warnings=T)$phy

data_squamata_cordylidae <-
as.data.frame(treedata(squamata_cordylidae_tree_red,

squamata_cordylidae_noDorsNa,sort=T,warnings=T)$data)
name.check(squamata_cordylidae_tree_red, data_squamata_cordylidae)

data_squamata_cordylidae$Dorsal <-
as.numeric(data_squamata_cordylidae$Dorsal) # Make numeric Dorsal
data_squamata_cordylidae$Log.Body.Mass <-
as.numeric(data_squamata_cordylidae$Log.Body.Mass) # Make numeric Log body
mass
data_squamata_cordylidae$Altitude.Low <-
as.numeric(data_squamata_cordylidae$Altitude.Low) # Make numeric
Altitude.Low
data_squamata_cordylidae$Altitude.High <-
as.numeric(data_squamata_cordylidae$Altitude.High) # Make numeric
Altitude.High
data_squamata_cordylidae$Count <-
as.numeric(data_squamata_cordylidae$Count) # Make numeric Count

```

```

data_squamata_cordylidae$Altitude <-
as.factor(data_squamata_cordylidae$Altitude) # Make factor Altitude
data_squamata_cordylidae$Habitat <-
as.factor(data_squamata_cordylidae$Habitat) # Make factor Habitat
data_squamata_cordylidae$Camouflage <-
as.factor(data_squamata_cordylidae$Camouflage) # Make factor Camouflage
data_squamata_cordylidae$Distribution <-
as.factor(data_squamata_cordylidae$Distribution) # Make factor Distribution
data_squamata_cordylidae$Polymorphic <-
as.factor(data_squamata_cordylidae$Polymorphic) # Make factor Polymorphic

# Make MCMCglmm call updateable so that we can use the dredge function to
change model compositions
MCMCglmm.updateable<- updateable(MCMCglmm)

start.time.CorStanley <- Sys.time()
global.model_dorsal_cordylidae_Stanley<- MCMCglmm.updateable(Dorsal ~
Habitat*Altitude+

Habitat*Distribution+

Altitude*Log.Body.Mass+

Distribution*Log.Body.Mass+

Habitat*Log.Body.Mass+

Habitat:Camouflage,
random = ~
Polymorphic,
pedigree =
squamata_cordylidae_tree_red,
burnin =
40000,
nitt = 100000,
thin = 20,

singular.ok=TRUE,
pr=TRUE,
verbose = F,
family =

"gaussian",
data =

data_squamata_cordylidae)
end.time.CorStanley <- Sys.time()
time.taken.CorStanley <- end.time.CorStanley - start.time.CorStanley
time.taken.CorStanley # 3.820939 secs

start.time.CorStanley1 <- Sys.time()
dredge.MCMCglmm_dorsal_cordylidae_Stanley<-
dredge(global.model_dorsal_cordylidae_Stanley, rank="DIC")
end.time.CorStanley1 <- Sys.time()
time.taken.CorStanley1 <- end.time.CorStanley1 - start.time.CorStanley1
time.taken.CorStanley1 # 5.406325 mins

dredge.MCMCglmm_dorsal_cordylidae_Stanley # evaluate top models
dorsal_cordylidae_Stanley <-
subset(dredge.MCMCglmm_dorsal_cordylidae_Stanley,DIC<327.7) # DIC <5
dorsal_cordylidae_Stanley
sw(dredge.MCMCglmm_dorsal_cordylidae_Stanley)

...

#14. Diplodactylidae
```{Diplodactylidae}

data_squamata_Diplodactylidae <- data_squamata[which(data_squamata$Family
== "Diplodactylidae"),]
data_squamata_Diplodactylidae <- droplevels(data_squamata_Diplodactylidae)

Make MCMCglmm call updateable so that we can use the dredge function to
change model compositions
MCMCglmm.updateable<- updateable(MCMCglmm)

start.time.Dip <- Sys.time()
global.model_dorsal_Diplodactylidae<- MCMCglmm.updateable(Dorsal ~
Habitat*Altitude+

Habitat*Distribution+

```

```

Altitude*Log.Body.Mass+
Distribution*Log.Body.Mass+
Habitat*Log.Body.Mass+
Habitat*Activity.Pattern+
Habitat:Camouflage,
Polymorphic,
squamata_tree_red,

"gaussian",

data_squamata_Diplodactylidae)
end.time.Dip <- Sys.time()
time.taken.Dip <- end.time.Dip - start.time.Dip
time.taken.Dip # 7.034083 secs

start.time.Dipl <- Sys.time()
dredge.MCMCglmm_dorsal_Diplodactylidae<-
dredge(global.model_dorsal_Diplodactylidae, rank="DIC")
end.time.Dipl <- Sys.time()
time.taken.Dipl <- end.time.Dipl - start.time.Dipl
time.taken.Dipl # 25.62877 mins

dredge.MCMCglmm_dorsal_Diplodactylidae #evaluate top models
dorsal_Diplodactylidae <-
subset(dredge.MCMCglmm_dorsal_Diplodactylidae,DIC<618.3) # DIC <5
dorsal_Diplodactylidae
sw(dredge.MCMCglmm_dorsal_Diplodactylidae)
...

#15. Phrynosomatidae
```{Phrynosomatidae}

data_squamata_Phrynosomatidae <- data_squamata[which(data_squamata$Family
== "Phrynosomatidae"),]
data_squamata_Phrynosomatidae <- droplevels(data_squamata_Phrynosomatidae)

# Make MCMCglmm call updateable so that we can use the dredge function to
change model compositions
MCMCglmm.updateable<- updateable(MCMCglmm)

start.time.Phry <- Sys.time()
global.model_dorsal_Phrynosomatidae<- MCMCglmm.updateable(Dorsal ~
Habitat*Altitude+

Habitat*Distribution+

Altitude*Log.Body.Mass+
Distribution*Log.Body.Mass+
Habitat*Log.Body.Mass+
Habitat*Activity.Pattern+

Habitat:Camouflage,
Polymorphic,
squamata_tree_red,

"gaussian",

random = ~
pedigree =
burnin = 40000,
nitt = 100000,
thin = 20,
singular.ok=TRUE,
pr=TRUE,
verbose = F,
family =
data =

```

```

data_squamata_Phrynosomatidae)
end.time.Phry <- Sys.time()
time.taken.Phry <- end.time.Phry - start.time.Phry
time.taken.Phry # 7.705505 secs

start.time.Phry1 <- Sys.time()
dredge.MCMCglmm_dorsal_Phrynosomatidae<-
dredge(global.model_dorsal_Phrynosomatidae, rank="DIC")
end.time.Phry1 <- Sys.time()
time.taken.Phry1 <- end.time.Phry1 - start.time.Phry1
time.taken.Phry1 # 28.79781 mins

dredge.MCMCglmm_dorsal_Phrynosomatidae #evaluate top models
dorsal_Phrynosomatidae <-
subset(dredge.MCMCglmm_dorsal_Phrynosomatidae,DIC<681.4) # DIC <5
dorsal_Phrynosomatidae
sw(dredge.MCMCglmm_dorsal_Phrynosomatidae)

phrynosomatidae_latitude <- MCMCglmm(Dorsal ~ Distribution,
                                     pedigree = squamata_tree_red,
                                     burnin = 40000,
                                     nitt = 100000,
                                     singular.ok=TRUE,
                                     thin = 20,
                                     pr=TRUE,
                                     verbose = F,
                                     family = "gaussian",
                                     data = data_squamata_Phrynosomatidae)

newish_phrynosomatidae_latitude <- expand.grid(Dorsal=rep(0, times=500),
                                               Distribution=factor(c("Subtropical", "Subtropical-
Temperate", "Temperate",
                                                                    "Tropical", "Tropical-
Subtropical",
                                                                    "Tropical-Subtropical-
Temperate")))

m.pred_phrynosomatidae_latitude<-
predict(phrynosomatidae_latitude,type="response", se.fit=TRUE,
interval="confidence",allow.new.levels=TRUE,
        newdata=newish_phrynosomatidae_latitude)

fitted_phrynosomatidae_latitude <-
cbind(newish_phrynosomatidae_latitude,m.pred_phrynosomatidae_latitude) #
combine the created dataset with the predicted values

model_fit16<-lm(fit~Distribution,fitted_phrynosomatidae_latitude)

# fit~Latitude
fitted_phrynosomatidae_latitude$Distribution <-
factor(fitted_phrynosomatidae_latitude$Distribution,levels = c("Tropical",
"Tropical-Subtropical", "Subtropical", "Subtropical-Temperate", "Tropical-
Subtropical-Temperate", "Temperate"))
ggplot(fitted_phrynosomatidae_latitude,aes(x=Distribution,y=fit,ymin=lwr,ymax=upr)
) +
  geom_point() +
  geom_errorbar()+theme_bw() +
  theme(panel.grid.minor.x=element_blank(),
        panel.grid.major.x=element_blank()) +
  guides(fill=FALSE) +
  scale_y_continuous(name="Dorsal brightness: fitted values (%)")+
  scale_x_discrete(name="Latitude")

phrynosomatidae_altitude <- MCMCglmm(Dorsal ~ Altitude,
                                     pedigree = squamata_tree_red,
                                     burnin = 40000,
                                     nitt = 100000,
                                     singular.ok=TRUE,
                                     thin = 20,
                                     pr=TRUE,
                                     verbose = F,
                                     family = "gaussian",
                                     data = data_squamata_Phrynosomatidae)

newish_phrynosomatidae_altitude <- expand.grid(Dorsal=rep(0, times=500),
                                               Altitude=factor(c("Low", "Low-Medium", "Medium-High",
"High", "All", "Unknown")))

m.pred_phrynosomatidae_altitude<-
predict(phrynosomatidae_altitude,type="response", se.fit=TRUE,
interval="confidence",allow.new.levels=TRUE,
        newdata=newish_phrynosomatidae_altitude)

```

```

fitted_phrynosomatidae_altitude <-
cbind(newwish_phrynosomatidae_altitude,m.pred_phrynosomatidae_altitude) #
combine the created dataset with the predicted values

model_fit_phryalt<-lm(fit~Altitude,fitted_phrynosomatidae_altitude)

# fit~Altitude
fitted_phrynosomatidae_altitude$Altitude <-
factor(fitted_phrynosomatidae_altitude$Altitude,levels = c("Low","Low-
Medium","Medium-High","High", "All","Unknown"))
ggplot(fitted_phrynosomatidae_altitude,aes(x=Altitude,y=fit,ymin=lwr,ymax=upr)
) +
  geom_point() +
  geom_errorbar()+theme_bw() +
  theme(panel.grid.minor.x=element_blank(),
        panel.grid.major.x=element_blank()) +
  guides(fill=FALSE) +
  scale_y_continuous(name="Dorsal brightness: fitted values (%)")+
  scale_x_discrete(name="Altitude")

phrynosomatidae_lathab <- MCMCglmm(Dorsal ~ Habitat*Distribution,
                                pedigree = squamata_tree_red,
                                burnin = 40000,
                                nitt = 100000,
                                singular.ok=TRUE,
                                thin = 20,
                                pr=TRUE,
                                verbose = F,
                                family = "gaussian",
                                data = data_squamata_Phrynosomatidae)

newwish_phrynosomatidae_lathab <- expand.grid(Dorsal=rep(0, times=100),
                                             Distribution=factor(c("Subtropical","Subtropical-
Temperate","Temperate",
                                                                "Tropical", "Tropical-
Subtropical",
                                                                "Tropical-Subtropical-
Temperate")),
                                             Habitat=factor(c("Open","OpenClosed","Closed")))

m.pred_phrynosomatidae_lathab<-
predict(phrynosomatidae_lathab,type="response", se.fit=TRUE,
interval="confidence",allow.new.levels=TRUE,
        newdata=newwish_phrynosomatidae_lathab)

fitted_phrynosomatidae_lathab <-
cbind(newwish_phrynosomatidae_lathab,m.pred_phrynosomatidae_lathab) #
combine the created dataset with the predicted values

model_fit_phrylathab<-
lm(fit~Distribution*Habitat,fitted_phrynosomatidae_lathab)

# fit~Habitat:Latitude
model_fit_phrylathab$model$Distribution <-
factor(model_fit_phrylathab$model$Distribution,levels = c("Tropical",
"Tropical-Subtropical","Subtropical","Subtropical-Temperate","Tropical-
Subtropical-Temperate","Temperate"))
model_fit_phrylathab$model$Habitat <-
factor(model_fit_phrylathab$model$Habitat,levels =
c("Open","OpenClosed","Closed"))
cat_plot(model_fit_phrylathab, pred=Distribution,
modx=Habitat,plot.points=F,interval=T,
        int.type=c("confidence"),point.size = 0.8, colors = "Dark2",
        y.label="Dorsal Brightness (%)",x.label="Latitude",
        legend.main="Habitat",geom = "line")

...

#16. Tropiduridae
```{Tropiduridae}
data_squamata_Tropiduridae <- data_squamata[which(data_squamata$Family ==
"Tropiduridae"),]
data_squamata_Tropiduridae <- droplevels(data_squamata_Tropiduridae)

Make MCMCglmm call updateable so that we can use the dredge function to
change model compositions
MCMCglmm.updateable<- updateable(MCMCglmm)

start.time.Tro <- Sys.time()

```

```

global.model_dorsal_Tropiduridae<- MCMCglmm.updateable(Dorsal ~
Habitat*Altitude+

Habitat*Distribution+

Altitude*Log.Body.Mass+

Distribution*Log.Body.Mass+

Habitat*Log.Body.Mass+

Habitat*Activity.Pattern+

Habitat:Camouflage,

Polymorphic,

squamata_tree_red,

 random = ~
 pedigree =

 burnin = 40000,
 nitt = 100000,
 thin = 20,
 singular.ok=TRUE,
 pr=TRUE,
 verbose = F,
 family = "gaussian",
 data =

data_squamata_Tropiduridae)
end.time.Tro <- Sys.time()
time.taken.Tro <- end.time.Tro - start.time.Tro
time.taken.Tro # 4.988067 secs

start.time.Tro1 <- Sys.time()
dredge.MCMCglmm_dorsal_Tropiduridae<-
dredge(global.model_dorsal_Tropiduridae, rank="DIC")
end.time.Tro1 <- Sys.time()
time.taken.Tro1 <- end.time.Tro1 - start.time.Tro1
time.taken.Tro1 # 22.29357 mins

dredge.MCMCglmm_dorsal_Tropiduridae # evaluate top models
dorsal_Tropiduridae <-
subset(dredge.MCMCglmm_dorsal_Tropiduridae,DIC<492.5) # DIC <5
dorsal_Tropiduridae
sw(dredge.MCMCglmm_dorsal_Tropiduridae)

tropiduridae_mass <- MCMCglmm(Dorsal ~ Log.Body.Mass,
 pedigree = squamata_tree_red,
 burnin = 40000,
 nitt = 100000,
 singular.ok=TRUE,
 thin = 20,
 pr=TRUE,
 verbose = F,
 family = "gaussian",
 data = data_squamata_Tropiduridae)

newish_tropiduridae_mass <- expand.grid(Dorsal=rep(0, times=200),
 Log.Body.Mass=runif(200,
min=min(data_squamata_Tropiduridae$Log.Body.Mass),
max=max(data_squamata_Tropiduridae$Log.Body.Mass)))

m.pred_tropiduridae_mass<-predict(tropiduridae_mass,type="response",
se.fit=TRUE, interval="confidence",allow.new.levels=TRUE,
 newdata=newish_tropiduridae_mass)

fitted_tropiduridae_mass <-
cbind(newish_tropiduridae_mass,m.pred_tropiduridae_mass) # combine the
created dataset with the predicted values

model_fit14<-lm(fit~Log.Body.Mass,fitted_tropiduridae_mass)

fit~mass
ggplot(fitted_tropiduridae_mass,aes(x=Log.Body.Mass,y=fit,ymin=lwr,ymax=upr)
) +
geom_point() +
geom_errorbar()+theme_bw() +
theme(panel.grid.minor.x=element_blank(),
panel.grid.major.x=element_blank()) +
guides(fill=FALSE) +
scale_y_continuous(name="Dorsal brightness: fitted values (%)")+
scale_x_continuous(name="Log body mass (g)")

```

```

tropicaluridae_masshab <- MCMCglmm(Dorsal ~ Log.Body.Mass*Habitat,
 pedigree = squamata_tree_red,
 burnin = 40000,
 nitt = 100000,
 singular.ok=TRUE,
 thin = 20,
 pr=TRUE,
 verbose = F,
 family = "gaussian",
 data = data_squamata_Tropicaluridae)

newish_tropicaluridae_masshab <- expand.grid(Dorsal=rep(0, times=200),
 Habitat=factor(c("Open", "OpenClosed", "Closed")),
 Log.Body.Mass=runif(200,
 min=min(data_squamata_Tropicaluridae$Log.Body.Mass),
 max=max(data_squamata_Tropicaluridae$Log.Body.Mass)))

m.pred_tropicaluridae_masshab<-predict(tropicaluridae_masshab,type="response",
 se.fit=TRUE, interval="confidence", allow.new.levels=TRUE,
 newdata=newish_tropicaluridae_masshab)

fitted_tropicaluridae_masshab <-
cbind(newish_tropicaluridae_masshab,m.pred_tropicaluridae_masshab) # combine
the created dataset with the predicted values

model_fit18<-lm(fit~Log.Body.Mass*Habitat,fitted_tropicaluridae_masshab)

fit~Body mass:Vegetation
model_fit18$model$Habitat <- factor(model_fit18$model$Habitat, levels =
c("Open", "OpenClosed", "Closed"))
tropi_mass_inter <- interact_plot(model_fit18, pred=Log.Body.Mass,
modx=Habitat, plot.points=F, interval=T,
 int.type=c("confidence"), point.size = 0.8, colors =
"Greys",
 y.label="Dorsal Brightness: fitted values
(%)", x.label="Log body mass (g)",
 legend.main="Vegetation")

tropi_mass_inter + geom_point(data=data_squamata_Tropicaluridae,
aes(x=Log.Body.Mass,y=Dorsal,color=Habitat),
inherit.aes = FALSE)

...

#17. Evolutionary rates
```{Evolutionary rates and comparisons across clades}

### First, let's map dorsal brightness and match data to tree for all focal
families ###

#Varanidae
varanidae_tree <-
treedata(squamata_tree_red, data_squamata_Varidae, sort=T, warnings=T)$phy
varanidae_data <-
as.data.frame(treedata(varanidae_tree, data_squamata_Varidae, sort=T, warnings=T)$data
)
name.check(varanidae_tree, varanidae_data) #  OK
dorsaltrait.varanidae <- setNames(data_squamata_Varidae$Dorsal,
data_squamata_Varidae$animal) #GIVES TRAIT TO MAP (Dorsal Brightness)

#Viperidae
viper_tree <-
treedata(squamata_tree_red, data_squamata_Viperidae, sort=T, warnings=T)$phy
viper_data <-
as.data.frame(treedata(viper_tree, data_squamata_Viperidae, sort=T, warnings=T)$data
)
name.check(viper_tree, viper_data) #  OK
dorsaltrait.viper <- setNames(data_squamata_Viperidae$Dorsal,
data_squamata_Viperidae$animal) #GIVES TRAIT TO MAP (Dorsal Brightness)

#Elapidae
elapid_tree <-
treedata(squamata_tree_red, data_squamata_Elapidae, sort=T, warnings=T)$phy
elapid_data <-
as.data.frame(treedata(elapid_tree, data_squamata_Elapidae, sort=T, warnings=T)$data
)
name.check(elapid_tree, elapid_data) #  OK
dorsaltrait.elapidae <- setNames(data_squamata_Elapidae$Dorsal,

```

```

data_squamata_Elapidae$animal) #GIVES TRAIT TO MAP (Dorsal Brightness)

#Scincidae
skink_tree <-
treedata(squamata_tree_red,data_squamata_Scincidae,sort=T,warnings=T)$phy
skink_data <-
as.data.frame(treedata(skink_tree,data_squamata_Scincidae,sort=T,warnings=T)$data
)
name.check(skink_tree, skink_data) #  OK
dorsaltrait.skink <- setNames(data_squamata_Scincidae$Dorsal,
data_squamata_Scincidae$animal) #GIVES TRAIT TO MAP (Dorsal Brightness)

#Lacertidae
lacertid_tree <-
treedata(squamata_tree_red,data_squamata_Lacertidae,sort=T,warnings=T)$phy
lacertid_data <-
as.data.frame(treedata(lacertid_tree,data_squamata_Lacertidae,sort=T,warnings=T)$data
)
name.check(lacertid_tree, lacertid_data) #  OK
dorsaltrait.lacertid <- setNames(data_squamata_Lacertidae$Dorsal,
data_squamata_Lacertidae$animal) #GIVES TRAIT TO MAP (Dorsal Brightness)

#Cordylidae
cordylid_tree <-
treedata(squamata_tree_red,data_squamata_Cordylidae,sort=T,warnings=T)$phy
cordylid_data <-
as.data.frame(treedata(cordylid_tree,data_squamata_Cordylidae,sort=T,warnings=T)$data
)
name.check(cordylid_tree, cordylid_data) #  OK
dorsaltrait.cordylid <- setNames(data_squamata_Cordylidae$Dorsal,
data_squamata_Cordylidae$animal) #GIVES TRAIT TO MAP (Dorsal Brightness)

#Diplodactylidae
diplo_tree <-
treedata(squamata_tree_red,data_squamata_Diplodactylidae,sort=T,warnings=T)$phy
diplo_data <-
as.data.frame(treedata(diplo_tree,data_squamata_Diplodactylidae,sort=T,warnings=T)$data
)
name.check(diplo_tree, diplo_data) #  OK
dorsaltrait.diplo <- setNames(data_squamata_Diplodactylidae$Dorsal,
data_squamata_Diplodactylidae$animal) #GIVES TRAIT TO MAP (Dorsal
Brightness)

#Phrynosomatidae
phryno_tree <-
treedata(squamata_tree_red,data_squamata_Phrynosomatidae,sort=T,warnings=T)$phy
phryno_data <-
as.data.frame(treedata(phryno_tree,data_squamata_Phrynosomatidae,sort=T,warnings=T)$data
)
name.check(phryno_tree, phryno_data) #  OK
dorsaltrait.phryno <- setNames(data_squamata_Phrynosomatidae$Dorsal,
data_squamata_Phrynosomatidae$animal) #GIVES TRAIT TO MAP (Dorsal
Brightness)

#Tropiduridae
tropi_tree <-
treedata(squamata_tree_red,data_squamata_Tropiduridae,sort=T,warnings=T)$phy
tropi_data <-
as.data.frame(treedata(tropi_tree,data_squamata_Tropiduridae,sort=T,warnings=T)$data
)
name.check(tropi_tree, tropi_data) #  OK
dorsaltrait.tropi <- setNames(data_squamata_Tropiduridae$Dorsal,
data_squamata_Tropiduridae$animal) #GIVES TRAIT TO MAP (Dorsal Brightness)

### Now we fit a multi-rate Brownian evolution model and extract mean
evolution rate per edge for the selected focal clades ###

start.time.fit1 <- Sys.time()
fit.ml.dorsum.cordylidae<-
multirateBM(cordylid_tree,dorsaltrait.cordylid,n.iter=3,parallel=TRUE)
end.time.fit1 <- Sys.time()
time.taken.fit1 <- end.time.fit1 - start.time.fit1
time.taken.fit1 # 15.92268 secs
SIG2<-apply(cordylid_tree$edge,1,function(e,x)
  mean(x[e]),x=fit.ml.dorsum.cordylidae$SIG2)
time <- as.data.frame((nodeHeights(cordylid_tree)[,1]))
cordylidae_rates <- cbind(time,SIG2)
cordylidae_rates$time <- cordylidae_rates$`(nodeHeights(cordylid_tree)[,
1])`

```

```

cordylidae_rates$time_adj <- cordylidae_rates$time-
max(cordylidae_rates$time)

start.time.fit2 <- Sys.time()
fit.ml.dorsum.varanidae<-
multirateBM(varanidae_tree,dorsaltrait.varanidae,n.iter=3,parallel=TRUE)
end.time.fit2 <- Sys.time()
time.taken.fit2 <- end.time.fit2 - start.time.fit2
time.taken.fit2 # 56.32161 secs
SIG2<-apply(varanidae_tree$edge,1,function(e,x)
  mean(x[e]),x=fit.ml.dorsum.varanidae$SIG2)
time <- as.data.frame((nodeHeights(varanidae_tree)[,1]))
varanidae_rates <- cbind(time,SIG2)
varanidae_rates$time <- varanidae_rates$(nodeHeights(varanidae_tree)[,
1])`
varanidae_rates$time_adj <- varanidae_rates$time-max(varanidae_rates$time)

start.time.fit3 <- Sys.time()
fit.ml.dorsum.tropiduridae<-
multirateBM(tropi_tree,dorsaltrait.tropi,n.iter=3,parallel=TRUE)
end.time.fit3 <- Sys.time()
time.taken.fit3 <- end.time.fit3 - start.time.fit3
time.taken.fit3 # 1.040777 mins
SIG2<-apply(tropi_tree$edge,1,function(e,x)
  mean(x[e]),x=fit.ml.dorsum.tropiduridae$SIG2)
time <- as.data.frame((nodeHeights(tropi_tree)[,1]))
tropiduridae_rates <- cbind(time,SIG2)
tropiduridae_rates$time <- tropiduridae_rates$(nodeHeights(tropi_tree)[,
1])`
tropiduridae_rates$time_adj <- tropiduridae_rates$time-
max(tropiduridae_rates$time)

start.time.fit4 <- Sys.time()
fit.ml.dorsum.diplodactylidae<-
multirateBM(diplo_tree,dorsaltrait.diplo,n.iter=3,parallel=TRUE)
end.time.fit4 <- Sys.time()
time.taken.fit4 <- end.time.fit4 - start.time.fit4
time.taken.fit4 # 1.675107 mins
SIG2<-apply(diplo_tree$edge,1,function(e,x)
  mean(x[e]),x=fit.ml.dorsum.diplodactylidae$SIG2)
time <- as.data.frame((nodeHeights(diplo_tree)[,1]))
diplodactylidae_rates <- cbind(time,SIG2)
diplodactylidae_rates$time <-
diplodactylidae_rates$(nodeHeights(diplo_tree)[, 1])`
diplodactylidae_rates$time_adj <- diplodactylidae_rates$time-
max(diplodactylidae_rates$time)

start.time.fit5 <- Sys.time()
fit.ml.dorsum.elapidae<-
multirateBM(elapid_tree,dorsaltrait.elapidae,n.iter=3,parallel=TRUE)
end.time.fit5 <- Sys.time()
time.taken.fit5 <- end.time.fit5 - start.time.fit5
time.taken.fit5 # 3.846337 mins
SIG2<-apply(elapid_tree$edge,1,function(e,x)
  mean(x[e]),x=fit.ml.dorsum.elapidae$SIG2)
time <- as.data.frame((nodeHeights(elapid_tree)[,1]))
elapidae_rates <- cbind(time,SIG2)
elapidae_rates$time <- elapidae_rates$(nodeHeights(elapid_tree)[, 1])`
elapidae_rates$time_adj <- elapidae_rates$time-max(elapidae_rates$time)

start.time.fit6 <- Sys.time()
fit.ml.dorsum.phrynosomatidae<-
multirateBM(phryno_tree,dorsaltrait.phryno,n.iter=3,parallel=TRUE)
end.time.fit6 <- Sys.time()
time.taken.fit6 <- end.time.fit6 - start.time.fit6
time.taken.fit6 # 5.551292 mins
SIG2<-apply(phryno_tree$edge,1,function(e,x)
  mean(x[e]),x=fit.ml.dorsum.phrynosomatidae$SIG2)
time <- as.data.frame((nodeHeights(phryno_tree)[,1]))
phrynosomatidae_rates <- cbind(time,SIG2)
phrynosomatidae_rates$time <-
phrynosomatidae_rates$(nodeHeights(phryno_tree)[, 1])`
phrynosomatidae_rates$time_adj <- phrynosomatidae_rates$time-
max(phrynosomatidae_rates$time)

start.time.fit7 <- Sys.time()
fit.ml.dorsum.scincidae<-
multirateBM(skink_tree,dorsaltrait.skink,n.iter=3,parallel=TRUE)
end.time.fit7 <- Sys.time()
time.taken.fit7 <- end.time.fit7 - start.time.fit7
time.taken.fit7 # 3.877081 mins
SIG2<-apply(skink_tree$edge,1,function(e,x)

```

```

      mean(x[e]),x=fit.ml.dorsum.scincidae$SIG2)
time <- as.data.frame((nodeHeights(skink_tree)[,1]))
scincidae_rates <- cbind(time,SIG2)
scincidae_rates$time <- scincidae_rates`(nodeHeights(skink_tree)[, 1])`
scincidae_rates$time_adj <- scincidae_rates$time-max(scincidae_rates$time)

start.time.fit8 <- Sys.time()
fit.ml.dorsum.lacertidae<-
multirateBM(lacertid_tree,dorsaltrait.lacertid,n.iter=3,parallel=TRUE)
end.time.fit8 <- Sys.time()
time.taken.fit8 <- end.time.fit8 - start.time.fit8
time.taken.fit8 # 28.49093 mins
SIG2<-apply(lacertid_tree$edge,1,function(e,x)
  mean(x[e]),x=fit.ml.dorsum.lacertidae$SIG2)
time <- as.data.frame((nodeHeights(lacertid_tree)[,1]))
lacertidae_rates <- cbind(time,SIG2)
lacertidae_rates$time <- lacertidae_rates`(nodeHeights(lacertid_tree)[,
1])`
lacertidae_rates$time_adj <- lacertidae_rates$time-
max(lacertidae_rates$time)

start.time.fit9 <- Sys.time()
fit.ml.dorsum.viperidae<-
multirateBM(viper_tree,dorsaltrait.viper,n.iter=3,parallel=TRUE)
end.time.fit9 <- Sys.time()
time.taken.fit9 <- end.time.fit9 - start.time.fit9
time.taken.fit9 # 24.58505 mins
SIG2<-apply(viper_tree$edge,1,function(e,x)
  mean(x[e]),x=fit.ml.dorsum.viperidae$SIG2)
time <- as.data.frame((nodeHeights(viper_tree)[,1]))
viperidae_rates <- cbind(time,SIG2)
viperidae_rates$time <- viperidae_rates`(nodeHeights(viper_tree)[, 1])`
viperidae_rates$time_adj <- viperidae_rates$time-max(viperidae_rates$time)

#combine all rates in one dataset
varanidae_rates$clade <- rep(c("Varanidae"))
varanidae_rates[1] <- NULL

viperidae_rates$clade <- rep(c("Viperidae"))
viperidae_rates[1] <- NULL

elapidae_rates$clade <- rep(c("Elapidae"))
elapidae_rates[1] <- NULL

phrynosomatidae_rates$clade <- rep(c("Phrynosomatidae"))
phrynosomatidae_rates[1] <- NULL

scincidae_rates$clade <- rep(c("Scincidae"))
scincidae_rates[1] <- NULL

lacertidae_rates$clade <- rep(c("Lacertidae"))
lacertidae_rates[1] <- NULL

cordylidae_rates$clade <- rep(c("Cordylidae"))
cordylidae_rates[1] <- NULL

diplodactylidae_rates$clade <- rep(c("Diplodactylidae"))
diplodactylidae_rates[1] <- NULL

tropiduridae_rates$clade <- rep(c("Tropiduridae"))
tropiduridae_rates[1] <- NULL

evo_rates <- rbind(varanidae_rates,
  viperidae_rates,
  #elapidae_rates,
  phrynosomatidae_rates,
  #scincidae_rates,
  lacertidae_rates,
  cordylidae_rates,
  diplodactylidae_rates,
  tropiduridae_rates)

#Plot brightness evo rates for all tips in the focal clades
#par(mfrow=c(3,3)) ## if selecting to plot all in one panel, set fsize=0.2

plot(fit.ml.dorsum.varanidae, fsize=0.8)
plot(fit.ml.dorsum.viperidae, fsize=0.3)
#plot(fit.ml.dorsum.elapidae, fsize=0.2)
#plot(fit.ml.dorsum.scincidae, fsize=0.2)
plot(fit.ml.dorsum.lacertidae, fsize=0.3)
plot(fit.ml.dorsum.cordylidae, fsize=0.9)
plot(fit.ml.dorsum.diplodactylidae, fsize=0.4)

```

```

plot(fit.ml.dorsum.phrynosomatidae, fsize=0.4)
plot(fit.ml.dorsum.tropiduridae, fsize=0.5)

dev.off()

#Plot separately mean brightness evo rates

ggplot(varanidae_rates, aes(time, SIG2)) + geom_smooth(color="black",
fill="darkgrey") + theme_bw() + theme(panel.grid = element_blank())

ggplot(viperidae_rates, aes(time, SIG2)) + geom_smooth(color="black",
fill="darkgrey") + theme_bw() + theme(panel.grid = element_blank())

ggplot(elapidae_rates, aes(time, SIG2)) + geom_smooth(color="black",
fill="darkgrey") + theme_bw() + theme(panel.grid = element_blank())

ggplot(scincidae_rates, aes(time, SIG2)) + geom_smooth(color="black",
fill="darkgrey") + theme_bw() + theme(panel.grid = element_blank())

ggplot(lacertidae_rates, aes(time, SIG2)) + geom_smooth(color="black",
fill="darkgrey") + theme_bw() + theme(panel.grid = element_blank())

ggplot(cordylidae_rates, aes(time, SIG2)) + geom_smooth(color="black",
fill="darkgrey") + theme_bw() + theme(panel.grid = element_blank())

ggplot(diplodactylidae_rates, aes(time, SIG2)) + geom_smooth(color="black",
fill="darkgrey") + theme_bw() + theme(panel.grid = element_blank())

ggplot(phrynosomatidae_rates, aes(time, SIG2)) + geom_smooth(color="black",
fill="darkgrey") + theme_bw() + theme(panel.grid = element_blank())

ggplot(tropiduridae_rates, aes(time, SIG2)) + geom_smooth(color="black",
fill="darkgrey") + theme_bw() + theme(panel.grid = element_blank())

getScaleData("epoch") #check epoch bins to place correct thresholds in the
geological time scale
max(nodeHeights(lacertid_tree)) #85.11085 Ma, oldest common ancestor within
clades and we will use this time to set the x axis upper limit

# Overlay in one single plot
colors <- c("Varanidae" = "#009DDC",
"Viperidae" = "#2A2D34",
"Lacertidae" = "#AD636C",
"Cordylidae" = "#8A628A",
"Diplodactylidae" = "#6761A8",
"Phrynosomatidae" = "#347E8D",
"Tropiduridae" = "#009B72")

#with individual points
a <- ggplot(evo_rates, aes(time_adj, SIG2, colour= clade, fill=clade)) +
geom_smooth() +
geom_jitter(aes(shape=clade, color=clade)) +
scale_shape_manual(values=c(3,2,4,5,8,1,9,10,13))+
scale_color_manual(values=colors) +
scale_fill_manual(values=colors) +
coord_geo(dat = list("epoch","periods"),xlim = c(-91.75,0),ylim =
c(min(SIG2)-22,max(SIG2)+3),
pos = list("b","b"), height =
list(unit(1,"lines"),unit(1,"lines")),
abbrv = list(T,F), neg = T, size = "auto",expand = F) +
scale_x_continuous(breaks = seq(-100, 0, 20), labels = -seq(-100, 0,
20)) +
labs(x = "Mya", y = expression(paste("Brightness evolution rate (
",sigma^2,")")) +
annotate("rect", xmin=-0.0117, xmax=-2.5800, ymin=min(SIG2)-22,
ymax=Inf, alpha=.4, fill="grey") +
annotate("rect", xmin=-5.3330, xmax=-23.0300, ymin=min(SIG2)-22,
ymax=Inf, alpha=.4, fill="grey") +
annotate("rect", xmin=-33.9000, xmax=-56.0000, ymin=min(SIG2)-22,
ymax=Inf, alpha=.4, fill="grey") +
annotate("rect", xmin=-66.0000, xmax=-91.75, ymin=min(SIG2)-22,
ymax=Inf, alpha=.4, fill="grey") +
geom_vline(xintercept = -66,linetype="dashed") + #K-PG mass extinction
geom_vline(xintercept = -55.8,linetype="dashed") + #PETM
geom_vline(xintercept = -53.7,linetype="dashed") + #ETM-2
geom_vline(xintercept = -49,linetype="dashed") + #Azolla event
geom_vline(xintercept = -5.3,linetype="dashed") + #Pliocene climate
geom_vline(xintercept = -2.5,linetype="dashed") + #Quaternary
glaciation
theme_bw() + theme(axis.line = element_line(),
panel.grid.major = element_blank(),
panel.grid.minor = element_blank(),
panel.border = element_blank(),

```

```

        panel.background = element_blank(),
        legend.position="top",
        legend.title=element_blank())

#without individual points
b <- ggplot(evo_rates, aes(time_adj, SIG2, colour= clade, fill=clade)) +
  geom_smooth() +
  scale_color_manual(values=colors) +
  scale_fill_manual(values=colors) +
  coord_geo(dat = list("epoch","periods"),xlim = c(-91.75,0),ylim =
c(min(SIG2)-22,max(SIG2)+3),
        pos = list("b","b"), height =
list(unit(1,"lines"),unit(1,"lines")),
        abbrev = list(T,F), neg = T, size = "auto",expand = F) +
  scale_x_continuous(breaks = seq(-100, 0, 20), labels = -seq(-100, 0, 20))
+
  labs(x = "Mya", y = expression(paste("Brightness evolution rate (
",sigma^2,")"))) +
  annotate("rect", xmin=-0.0117, xmax=-2.5800, ymin=min(SIG2)-22, ymax=Inf,
alpha=.4, fill="grey") +
  annotate("rect", xmin=-5.3330, xmax=-23.0300, ymin=min(SIG2)-22,
ymax=Inf, alpha=.4, fill="grey") +
  annotate("rect", xmin=-33.9000, xmax=-56.0000, ymin=min(SIG2)-22,
ymax=Inf, alpha=.4, fill="grey") +
  annotate("rect", xmin=-66.0000, xmax=-91.75, ymin=min(SIG2)-22, ymax=Inf,
alpha=.4, fill="grey") +
  geom_vline(xintercept = -66,linetype="dashed") + #K-PG mass extinction
  geom_vline(xintercept = -55.8,linetype="dashed") + #PETM
  geom_vline(xintercept = -53.7,linetype="dashed") + #ETM-2
  geom_vline(xintercept = -49,linetype="dashed") + #Azolla event
  geom_vline(xintercept = -5.3,linetype="dashed") + #Pliocene climate
  geom_vline(xintercept = -2.5,linetype="dashed") + #Quaternary glaciation
  theme_bw() + theme(axis.line = element_line(),
  panel.grid.major = element_blank(),
  panel.grid.minor = element_blank(),
  panel.border = element_blank(),
  panel.background = element_blank(),
  legend.position="top",
  legend.title=element_blank())

#zoom in from Miocene with individual points
c <- ggplot(evo_rates, aes(time_adj, SIG2, colour= clade, fill=clade)) +
  geom_smooth() +
  geom_jitter(aes(shape=clade, color=clade)) +
  scale_shape_manual(values=c(3,2,4,5,8,1,9,10,13))+
  scale_color_manual(values=colors) +
  scale_fill_manual(values=colors) +
  coord_geo(dat = list("stages","epoch"),xlim = c(-23.0300,0),ylim =
c(min(SIG2)-3,max(SIG2)),
        pos = list("b","b"), height =
list(unit(1,"lines"),unit(1,"lines")),
        abbrev = list(T,F), neg = T, size = "auto",expand = F) +
  scale_x_continuous(breaks = seq(-25, 0, 5), labels = -seq(-25, 0, 5)) +
  labs(x = "Mya", y = expression(paste("Brightness evolution rate (
",sigma^2,")"))) +
  annotate("rect", xmin=-0.0117, xmax=-0.1290, ymin=min(SIG2)-3, ymax=Inf,
alpha=.4, fill="grey") +
  annotate("rect", xmin=-0.7740, xmax=-1.8000, ymin=min(SIG2)-3, ymax=Inf,
alpha=.4, fill="grey") +
  annotate("rect", xmin=-2.5800, xmax=-3.6000, ymin=min(SIG2)-3, ymax=Inf,
alpha=.4, fill="grey") +
  annotate("rect", xmin=-5.3330, xmax=-7.2460, ymin=min(SIG2)-3, ymax=Inf,
alpha=.4, fill="grey") +
  annotate("rect", xmin=-11.6300, xmax=-13.820, ymin=min(SIG2)-3, ymax=Inf,
alpha=.4, fill="grey") +
  annotate("rect", xmin=-20.4400, xmax=-15.9700, ymin=min(SIG2)-3,
ymax=Inf, alpha=.4, fill="grey") +
  geom_vline(xintercept = -5.3,linetype="dashed") + #Pliocene climate
  geom_vline(xintercept = -2.5,linetype="dashed") + #Quaternary glaciation
  theme_bw() + theme(axis.line = element_line(),
  panel.grid.major = element_blank(),
  panel.grid.minor = element_blank(),
  panel.border = element_blank(),
  panel.background = element_blank(),
  legend.position="top",
  legend.title=element_blank())

#zoom in from Miocene without individual points
d <- ggplot(evo_rates, aes(time_adj, SIG2, colour= clade, fill=clade)) +
  geom_smooth() +
  scale_color_manual(values=colors) +
  scale_fill_manual(values=colors) +

```

```

coord_geo(dat = list("stages","epoch"),xlim = c(-23.0300,0),ylim =
c(min(SIG2)-3,max(SIG2)),
  pos = list("b","b"), height =
list(unit(1,"lines"),unit(1,"lines")),
  abbrv = list(T,F), neg = T, size = "auto",expand = F) +
  scale_x_continuous(breaks = seq(-25, 0, 5), labels = -seq(-25, 0, 5)) +
  labs(x = "Mya", y = expression(paste("Brightness evolution rate (
",sigma^2,")")) +
  annotate("rect", xmin=-0.0117, xmax=-0.1290, ymin=min(SIG2)-3, ymax=Inf,
alpha=.4, fill="grey") +
  annotate("rect", xmin=-0.7740, xmax=-1.8000, ymin=min(SIG2)-3, ymax=Inf,
alpha=.4, fill="grey") +
  annotate("rect", xmin=-2.5800, xmax=-3.6000, ymin=min(SIG2)-3, ymax=Inf,
alpha=.4, fill="grey") +
  annotate("rect", xmin=-5.3330, xmax=-7.2460, ymin=min(SIG2)-3, ymax=Inf,
alpha=.4, fill="grey") +
  annotate("rect", xmin=-11.6300, xmax=-13.820, ymin=min(SIG2)-3, ymax=Inf,
alpha=.4, fill="grey") +
  annotate("rect", xmin=-20.4400, xmax=-15.9700, ymin=min(SIG2)-3,
ymax=Inf, alpha=.4, fill="grey") +
  geom_vline(xintercept = -5.3,linetype="dashed") + #Pliocene climate
  geom_vline(xintercept = -2.5,linetype="dashed") + #Quaternary glaciation
  theme_bw() + theme(axis.line = element_line(),
  panel.grid.major = element_blank(),
  panel.grid.minor = element_blank(),
  panel.border = element_blank(),
  panel.background = element_blank(),
  legend.position="top",
  legend.title=element_blank())

#Load delta-O18 data from Gaskell et al. 2022
paleo_climate <- read_excel("Paleo_d180.xlsx")

#Plot paleo data for the last 23 million years (zoom in from Miocene)
e <- ggplot(paleo_climate, aes(x=age, y=d180)) +
  geom_smooth(method = "loess", color="black", fill="grey") +
  geom_line( color="black") +
  labs(x = "Mya", y = expression(paste(" ",delta^18,"O"))) +
  scale_x_reverse() +
  coord_geo(dat = list("stage","epoch"),xlim = c(23.0300,0),
  ylim = c(0.5,max(paleo_climate$d180)+0.5),
  pos = list("b","b"), height =
list(unit(1,"lines"),unit(1,"lines")),
  abbrv = list(T,F), neg = F, size = "auto",expand = F) +
  annotate("rect", xmin=0.0117, xmax=0.1290, ymin=0.5,
  ymax=Inf, alpha=.4, fill="grey") +
  annotate("rect", xmin=0.7740, xmax=1.8000, ymin=0.5,
  ymax=Inf, alpha=.4, fill="grey") +
  annotate("rect", xmin=2.5800, xmax=3.6000, ymin=0.5,
  ymax=Inf, alpha=.4, fill="grey") +
  annotate("rect", xmin=5.3330, xmax=7.2460, ymin=0.5,
  ymax=Inf, alpha=.4, fill="grey") +
  annotate("rect", xmin=11.6300, xmax=13.820, ymin=0.5,
  ymax=Inf, alpha=.4, fill="grey") +
  annotate("rect", xmin=20.4400, xmax=15.9700, ymin=0.5,
  ymax=Inf, alpha=.4, fill="grey") +
  geom_vline(xintercept = 5.3,linetype="dashed") + #Pliocene climate
  geom_vline(xintercept = 2.5,linetype="dashed") + #Quaternary glaciation
  theme_bw() + theme(axis.line = element_line(),
  panel.grid.major = element_blank(),
  panel.grid.minor = element_blank(),
  panel.border = element_blank(),
  panel.background = element_blank())

#Plot paleo data for the last 92 million years
f <- ggplot(paleo_climate, aes(x=age, y=d180)) +
  geom_smooth(method = "loess", color="black", fill="grey") +
  geom_line( color="black") +
  labs(x = "Mya", y = expression(paste(" ",delta^18,"O"))) +
  scale_x_reverse() +
  coord_geo(dat = list("epoch","periods"),xlim = c(91.75,0),
  ylim = c(min(paleo_climate$d180)-
0.5,max(paleo_climate$d180)+0.5),
  pos = list("b","b"), height =
list(unit(1,"lines"),unit(1,"lines")),
  abbrv = list(T,F), neg = F, size = "auto",expand = F) +
  annotate("rect", xmin=0.0117, xmax=2.5800, ymin=min(paleo_climate$d180)-
0.5,
  ymax=Inf, alpha=.4, fill="grey") +
  annotate("rect", xmin=5.3330, xmax=23.0300, ymin=min(paleo_climate$d180)-
0.5,
  ymax=Inf, alpha=.4, fill="grey") +

```

```

  annotate("rect", xmin=33.9000, xmax=56.0000,
ymin=min(paleo_climate$d180)-0.5,
  ymax=Inf, alpha=.4, fill="grey") +
  annotate("rect", xmin=66.0000, xmax=91.75, ymin=min(paleo_climate$d180)-
0.5,
  ymax=Inf, alpha=.4, fill="grey") +
  geom_vline(xintercept = 66,linetype="dashed") + #K-PG mass extinction
  geom_vline(xintercept = 55.8,linetype="dashed") + #PETM
  geom_vline(xintercept = 53.7,linetype="dashed") + #ETM-2
  geom_vline(xintercept = 49,linetype="dashed") + #Azolla event
  geom_vline(xintercept = 5.3,linetype="dashed") + #Pliocene climate
  geom_vline(xintercept= 2.5,linetype="dashed") + #Quaternary glaciation
  theme_bw() + theme(axis.line = element_line(),
  panel.grid.major = element_blank(),
  panel.grid.minor = element_blank(),
  panel.border = element_blank(),
  panel.background = element_blank())

```

```

#with individual points
ggarrange(a, c, f, e,
  labels = c("a", "b", "c", "d"),
  #heights = c(1,1.5,1),
  ncol = 2, nrow = 2,
  common.legend = T,
  align = "v")

```

```

#without individual points
ggarrange(b, d, f, e,
  labels = c("a", "b", "c", "d"),
  #heights = c(1,1.5,1),
  ncol = 2, nrow = 2,
  common.legend = T,
  align = "v")

```

```

ggarrange(a, f,
  labels = c("A", "B"),
  ncol = 1, nrow = 2)

```

```

#-----
#-----
#-----

```

```

### multitree to test for evolutionary rate variation of dorsal brightness
between clades ###

```

```

trees<-c(varanidae_tree,viper_tree,
  lacertid_tree, cordylid_tree,
  diplo_tree, phryno_tree, tropi_tree)
x<-list(dorsaltrait.varanidae, dorsaltrait.viper,
  dorsaltrait.lacertid, dorsaltrait.cordylid,
  dorsaltrait.diplo, dorsaltrait.phryno, dorsaltrait.tropi)

```

```

fit.all<-ratebytree(trees,x)
fit.all

```

```

fit.all.post<-posthoc.ratebytree(fit.all)
fit.all.post

```

```

par(mfrow=c(3,3))
nulo<-mapply(phenogram, tree=trees, x=x, MoreArgs=list(ylim=range(x),
  ftype="off"))

```

```

dev.off()

```

```

##add mean brightness rates for differences among clades to last panel

```

```

ratebytree_data_sig2 <- as.data.frame(fit.all$multi.rate.model$sig2)
ratebytree_data_se <- as.data.frame(fit.all$multi.rate.model$SE.sig2)
ratebytree_data <- cbind(ratebytree_data_sig2, ratebytree_data_se)
colnames(ratebytree_data) <- c("mean", 'se')

```

```

ratebytree_data$clade <- c("Varanidae",
  "Viperidae",
  "Lacertidae",
  "Cordylidae",
  "Diplodactylidae",
  "Phrynosomatidae",
  "Tropiduridae")

```

```

g <- ggplot(data=ratebytree_data, aes(x=clade, y=mean, colour= clade,

```

```

fill=clade))+
  geom_bar(stat="identity")+
  geom_errorbar(aes(ymin=mean-se,ymax=mean+se),width=.2)+
  theme_bw()+theme(panel.grid = element_blank()+
  scale_color_manual(values=colors) +
  scale_fill_manual(values=colors) +
  labs(x="",y=expression(paste("Mean brightness evolution rate (
",sigma^2,"")))) +
  theme(legend.position = "none")

h <- ggarrange(b, d, f, e,
  labels = c("A", "B", "C", "D"),
  #heights = c(1,1.5,1),
  ncol = 2, nrow = 2,
  common.legend = T,
  align = "v")

ggarrange(h, g,
  labels = c("", "E"),
  nrow = 2, ncol=1,
  heights = c(2,1),
  align = "v")

#-----
#-----
#-----

#let's check the robustness of our evo rates by fitting different smoothing
parameters
lambda<-c(1,0.9,0.8,0.7,0.6,0.5,2,3,4,5)

start.time.lambda <- Sys.time()

#Varanidae
fits.varanidae<-list()
for(i in 1:length(lambda))
  fits.varanidae[[i]]<-
multirateBM(varanidae_tree,dorsaltrait.varanidae,lambda=lambda[i],
  n.iter=3, parallel=TRUE)

par(mfrow=c(5,2))
sig2.varanidae <- fit.ml.dorsum.varanidae$sig2
for(i in 1:length(fits.varanidae)){
  xylim<-range(c(sig2.varanidae,fits.varanidae[[i]]$sig2))
  plot(sig2.varanidae,fits.varanidae[[i]]$sig2,log="xy",xlim=xylim,
  ylim=xylim,bty="n",las=1,pch=21,cex=1.2,
  bg="grey",xlab=expression(paste("baseline ",sigma^2)),
  ylab=expression(paste("estimated ",sigma^2)),
  cex.axis=0.8)
  lines(xylim,xylim)
  if(i==1) mtext(expression(paste("a ",lambda,"= 1")),adj=0)
  else if(i==2) mtext(expression(paste("b ",lambda,"= 0.9")),adj=0)
  else if(i==3) mtext(expression(paste("c ",lambda,"= 0.8")),adj=0)
  else if(i==4) mtext(expression(paste("d ",lambda,"= 0.7")),adj=0)
  else if(i==5) mtext(expression(paste("e ",lambda,"= 0.6")),adj=0)
  else if(i==6) mtext(expression(paste("f ",lambda,"= 0.5")),adj=0)
  else if(i==7) mtext(expression(paste("g ",lambda,"= 2")),adj=0)
  else if(i==8) mtext(expression(paste("h ",lambda,"= 3")),adj=0)
  else if(i==9) mtext(expression(paste("i ",lambda,"= 4")),adj=0)
  else if(i==10) mtext(expression(paste("j ",lambda,"= 5")),adj=0)
}

par(mfrow=c(5,2))
plot(fits.varanidae[[1]],ftype="off",lwd=0.5,
  mar=c(1.1,1.1,2.1,1.1))
mtext(expression(paste("a ",lambda," = 1")),
  line=0,adj=0)
plot(fits.varanidae[[2]],ftype="off",lwd=0.5,
  mar=c(1.1,1.1,2.1,1.1))
mtext(expression(paste("b ",lambda," = 0.9")),
  line=0,adj=0)
plot(fits.varanidae[[3]],ftype="off",lwd=0.5,
  mar=c(1.1,1.1,2.1,1.1))
mtext(expression(paste("c ",lambda," = 0.8")),
  line=0,adj=0)
plot(fits.varanidae[[4]],ftype="off",lwd=0.5,
  mar=c(1.1,1.1,2.1,1.1))
mtext(expression(paste("d ",lambda," = 0.7")),
  line=0,adj=0)
plot(fits.varanidae[[5]],ftype="off",lwd=0.5,
  mar=c(1.1,1.1,2.1,1.1))

```

```

mtext(expression(paste("e ", lambda, " = 0.6")),
  line=0,adj=0)
plot(fits.varanidae[[6]],ftype="off",lwd=0.5,
  mar=c(1.1,1.1,2.1,1.1))
mtext(expression(paste("f ", lambda, " = 0.5")),
  line=0,adj=0)
plot(fits.varanidae[[7]],ftype="off",lwd=0.5,
  mar=c(1.1,1.1,2.1,1.1))
mtext(expression(paste("g ", lambda, " = 2")),
  line=0,adj=0)
plot(fits.varanidae[[8]],ftype="off",lwd=0.5,
  mar=c(1.1,1.1,2.1,1.1))
mtext(expression(paste("h ", lambda, " = 3")),
  line=0,adj=0)
plot(fits.varanidae[[9]],ftype="off",lwd=0.5,
  mar=c(1.1,1.1,2.1,1.1))
mtext(expression(paste("i ", lambda, " = 4")),
  line=0,adj=0)
plot(fits.varanidae[[10]],ftype="off",lwd=0.5,
  mar=c(1.1,1.1,2.1,1.1))
mtext(expression(paste("j ", lambda, " = 5")),
  line=0,adj=0)

#Viperidae
fits.viperidae<-list()
for(i in 1:length(lambda))
  fits.viperidae[[i]]<-
multirateBM(viper_tree,dorsaltrait.viper,lambda=lambda[i],
  n.iter=3, parallel=TRUE)

par(mfrow=c(5,2))
sig2.viperidae <- fit.ml.dorsum.viperidae$sig2
for(i in 1:length(fits.viperidae)){
  xylim<-range(c(sig2.viperidae,fits.viperidae[[i]]$sig2))
  plot(sig2.viperidae,fits.viperidae[[i]]$sig2,log="xy",xlim=xylim,
    ylim=xylim,bty="n",las=1,pch=21,cex=1.2,
    bg="grey",xlab=expression(paste("baseline ", sigma^2)),
    ylab=expression(paste("estimated ", sigma^2)),
    cex.axis=0.8)
  lines(xylim,xylim)
  if(i==1) mtext(expression(paste("a ", lambda, "= 1")),adj=0)
  else if(i==2) mtext(expression(paste("b ", lambda, "= 0.9")),adj=0)
  else if(i==3) mtext(expression(paste("c ", lambda, "= 0.8")),adj=0)
  else if(i==4) mtext(expression(paste("d ", lambda, "= 0.7")),adj=0)
  else if(i==5) mtext(expression(paste("e ", lambda, "= 0.6")),adj=0)
  else if(i==6) mtext(expression(paste("f ", lambda, "= 0.5")),adj=0)
  else if(i==7) mtext(expression(paste("g ", lambda, "= 2")),adj=0)
  else if(i==8) mtext(expression(paste("h ", lambda, "= 3")),adj=0)
  else if(i==9) mtext(expression(paste("i ", lambda, "= 4")),adj=0)
  else if(i==10) mtext(expression(paste("j ", lambda, "= 5")),adj=0)
}

par(mfrow=c(5,2))
plot(fits.viperidae[[1]],ftype="off",lwd=0.5,
  mar=c(1.1,1.1,2.1,1.1))
mtext(expression(paste("a ", lambda, " = 1")),
  line=0,adj=0)
plot(fits.viperidae[[2]],ftype="off",lwd=0.5,
  mar=c(1.1,1.1,2.1,1.1))
mtext(expression(paste("b ", lambda, " = 0.9")),
  line=0,adj=0)
plot(fits.viperidae[[3]],ftype="off",lwd=0.5,
  mar=c(1.1,1.1,2.1,1.1))
mtext(expression(paste("c ", lambda, " = 0.8")),
  line=0,adj=0)
plot(fits.viperidae[[4]],ftype="off",lwd=0.5,
  mar=c(1.1,1.1,2.1,1.1))
mtext(expression(paste("d ", lambda, " = 0.7")),
  line=0,adj=0)
plot(fits.viperidae[[5]],ftype="off",lwd=0.5,
  mar=c(1.1,1.1,2.1,1.1))
mtext(expression(paste("e ", lambda, " = 0.6")),
  line=0,adj=0)
plot(fits.viperidae[[6]],ftype="off",lwd=0.5,
  mar=c(1.1,1.1,2.1,1.1))
mtext(expression(paste("f ", lambda, " = 0.5")),
  line=0,adj=0)
plot(fits.viperidae[[7]],ftype="off",lwd=0.5,
  mar=c(1.1,1.1,2.1,1.1))
mtext(expression(paste("g ", lambda, " = 2")),

```

```

    line=0,adj=0)
plot(fits.viperidae[[8]],ftype="off",lwd=0.5,
     mar=c(1.1,1.1,2.1,1.1))
mtext(expression(paste("h " ,lambda," = 3")),
       line=0,adj=0)
plot(fits.viperidae[[9]],ftype="off",lwd=0.5,
     mar=c(1.1,1.1,2.1,1.1))
mtext(expression(paste("i " ,lambda," = 4")),
       line=0,adj=0)
plot(fits.viperidae[[10]],ftype="off",lwd=0.5,
     mar=c(1.1,1.1,2.1,1.1))
mtext(expression(paste("j " ,lambda," = 5")),
       line=0,adj=0)

#Tropiduridae
fits.tropiduridae<-list()
for(i in 1:length(lambda))
  fits.tropiduridae[[i]]<-
multirateBM(tropi_tree,dorsaltrait.tropi,lambda=lambda[i],
            n.iter=3, parallel=TRUE)

par(mfrow=c(5,2))
sig2.tropiduridae <- fit.ml.dorsum.tropiduridae$sig2
for(i in 1:length(fits.tropiduridae)){
  xlim<-range(c(sig2.tropiduridae,fits.tropiduridae[[i]]$sig2))
  plot(sig2.tropiduridae,fits.tropiduridae[[i]]$sig2,log="xy",xlim=xylim,
       ylim=xylim,bty="n",las=1,pch=21,cex=1.2,
       bg="grey",xlab=expression(paste("baseline ",sigma^2)),
       ylab=expression(paste("estimated ",sigma^2)),
       cex.axis=0.8)
  lines(xylim,xylim)
  if(i==1) mtext(expression(paste("a " ,lambda,"= 1")),adj=0)
  else if(i==2) mtext(expression(paste("b " ,lambda,"= 0.9")),adj=0)
  else if(i==3) mtext(expression(paste("c " ,lambda,"= 0.8")),adj=0)
  else if(i==4) mtext(expression(paste("d " ,lambda,"= 0.7")),adj=0)
  else if(i==5) mtext(expression(paste("e " ,lambda,"= 0.6")),adj=0)
  else if(i==6) mtext(expression(paste("f " ,lambda,"= 0.5")),adj=0)
  else if(i==7) mtext(expression(paste("g " ,lambda,"= 2")),adj=0)
  else if(i==8) mtext(expression(paste("h " ,lambda,"= 3")),adj=0)
  else if(i==9) mtext(expression(paste("i " ,lambda,"= 4")),adj=0)
  else if(i==10) mtext(expression(paste("j " ,lambda,"= 5")),adj=0)
}

par(mfrow=c(5,2))
plot(fits.tropiduridae[[1]],ftype="off",lwd=0.5,
     mar=c(1.1,1.1,2.1,1.1))
mtext(expression(paste("a " ,lambda," = 1")),
       line=0,adj=0)
plot(fits.tropiduridae[[2]],ftype="off",lwd=0.5,
     mar=c(1.1,1.1,2.1,1.1))
mtext(expression(paste("b " ,lambda," = 0.9")),
       line=0,adj=0)
plot(fits.tropiduridae[[3]],ftype="off",lwd=0.5,
     mar=c(1.1,1.1,2.1,1.1))
mtext(expression(paste("c " ,lambda," = 0.8")),
       line=0,adj=0)
plot(fits.tropiduridae[[4]],ftype="off",lwd=0.5,
     mar=c(1.1,1.1,2.1,1.1))
mtext(expression(paste("d " ,lambda," = 0.7")),
       line=0,adj=0)
plot(fits.tropiduridae[[5]],ftype="off",lwd=0.5,
     mar=c(1.1,1.1,2.1,1.1))
mtext(expression(paste("e " ,lambda," = 0.6")),
       line=0,adj=0)
plot(fits.tropiduridae[[6]],ftype="off",lwd=0.5,
     mar=c(1.1,1.1,2.1,1.1))
mtext(expression(paste("f " ,lambda," = 0.5")),
       line=0,adj=0)
plot(fits.tropiduridae[[7]],ftype="off",lwd=0.5,
     mar=c(1.1,1.1,2.1,1.1))
mtext(expression(paste("g " ,lambda," = 2")),
       line=0,adj=0)
plot(fits.tropiduridae[[8]],ftype="off",lwd=0.5,
     mar=c(1.1,1.1,2.1,1.1))
mtext(expression(paste("h " ,lambda," = 3")),
       line=0,adj=0)
plot(fits.tropiduridae[[9]],ftype="off",lwd=0.5,
     mar=c(1.1,1.1,2.1,1.1))
mtext(expression(paste("i " ,lambda," = 4")),
       line=0,adj=0)

```

```

plot(fits.tropiduridae[[10]],ftype="off",lwd=0.5,
     mar=c(1.1,1.1,2.1,1.1))
mtext(expression(paste("j" ),lambda," = 5")),
       line=0,adj=0)

#Diplodactylidae
fits.diplodactylidae<-list()
for(i in 1:length(lambda))
  fits.diplodactylidae[[i]]<-
  multirateBM(diplo_tree,dorsaltrait.diplo,lambda=lambda[i],
              n.iter=3, parallel=TRUE)

par(mfrow=c(5,2))
sig2.diplodactylidae <- fit.ml.dorsum.diplodactylidae$sig2
for(i in 1:length(fits.diplodactylidae)){
  ylim<-range(c(sig2.diplodactylidae,fits.diplodactylidae[[i]]$sig2))

plot(sig2.diplodactylidae,fits.diplodactylidae[[i]]$sig2,log="xy",xlim=xylim
,
     ylim=xylim,bty="n",las=1,pch=21,cex=1.2,
     bg="grey",xlab=expression(paste("baseline ",sigma^2)),
     ylab=expression(paste("estimated ",sigma^2)),
     cex.axis=0.8)
  lines(xylim,ylim)
  if(i==1) mtext(expression(paste("a" ),lambda,"= 1")),adj=0)
  else if(i==2) mtext(expression(paste("b" ),lambda,"= 0.9")),adj=0)
  else if(i==3) mtext(expression(paste("c" ),lambda,"= 0.8")),adj=0)
  else if(i==4) mtext(expression(paste("d" ),lambda,"= 0.7")),adj=0)
  else if(i==5) mtext(expression(paste("e" ),lambda,"= 0.6")),adj=0)
  else if(i==6) mtext(expression(paste("f" ),lambda,"= 0.5")),adj=0)
  else if(i==7) mtext(expression(paste("g" ),lambda,"= 2")),adj=0)
  else if(i==8) mtext(expression(paste("h" ),lambda,"= 3")),adj=0)
  else if(i==9) mtext(expression(paste("i" ),lambda,"= 4")),adj=0)
  else if(i==10) mtext(expression(paste("j" ),lambda,"= 5")),adj=0)
}

par(mfrow=c(5,2))
plot(fits.diplodactylidae[[1]],ftype="off",lwd=0.5,
     mar=c(1.1,1.1,2.1,1.1))
mtext(expression(paste("a" ),lambda," = 1")),
       line=0,adj=0)
plot(fits.diplodactylidae[[2]],ftype="off",lwd=0.5,
     mar=c(1.1,1.1,2.1,1.1))
mtext(expression(paste("b" ),lambda," = 0.9")),
       line=0,adj=0)
plot(fits.diplodactylidae[[3]],ftype="off",lwd=0.5,
     mar=c(1.1,1.1,2.1,1.1))
mtext(expression(paste("c" ),lambda," = 0.8")),
       line=0,adj=0)
plot(fits.diplodactylidae[[4]],ftype="off",lwd=0.5,
     mar=c(1.1,1.1,2.1,1.1))
mtext(expression(paste("d" ),lambda," = 0.7")),
       line=0,adj=0)
plot(fits.diplodactylidae[[5]],ftype="off",lwd=0.5,
     mar=c(1.1,1.1,2.1,1.1))
mtext(expression(paste("e" ),lambda," = 0.6")),
       line=0,adj=0)
plot(fits.diplodactylidae[[6]],ftype="off",lwd=0.5,
     mar=c(1.1,1.1,2.1,1.1))
mtext(expression(paste("f" ),lambda," = 0.5")),
       line=0,adj=0)
plot(fits.diplodactylidae[[7]],ftype="off",lwd=0.5,
     mar=c(1.1,1.1,2.1,1.1))
mtext(expression(paste("g" ),lambda," = 2")),
       line=0,adj=0)
plot(fits.diplodactylidae[[8]],ftype="off",lwd=0.5,
     mar=c(1.1,1.1,2.1,1.1))
mtext(expression(paste("h" ),lambda," = 3")),
       line=0,adj=0)
plot(fits.diplodactylidae[[9]],ftype="off",lwd=0.5,
     mar=c(1.1,1.1,2.1,1.1))
mtext(expression(paste("i" ),lambda," = 4")),
       line=0,adj=0)
plot(fits.diplodactylidae[[10]],ftype="off",lwd=0.5,
     mar=c(1.1,1.1,2.1,1.1))
mtext(expression(paste("j" ),lambda," = 5")),
       line=0,adj=0)

```

```

#Phrynosomatidae
fits.phrynosomatidae<-list()
for(i in 1:length(lambda))
  fits.phrynosomatidae[[i]]<-
multirateBM(phryno_tree,dorsaltrait.phryno,lambda=lambda[i],
              n.iter=3, parallel=TRUE)

par(mfrow=c(5,2))
sig2.phrynosomatidae <- fit.ml.dorsum.phrynosomatidae$sig2
for(i in 1:length(fits.phrynosomatidae)){
  xylim<-range(c(sig2.phrynosomatidae,fits.phrynosomatidae[[i]]$sig2))

plot(sig2.phrynosomatidae,fits.phrynosomatidae[[i]]$sig2,log="xy",xlim=xylim
,
      ylim=xylim,bty="n",las=1,pch=21,cex=1.2,
      bg="grey",xlab=expression(paste("baseline ",sigma^2)),
      ylab=expression(paste("estimated ",sigma^2)),
      cex.axis=0.8)
  lines(xylim,xylim)
  if(i==1) mtext(expression(paste("a" ,lambda,"= 1")),adj=0)
  else if(i==2) mtext(expression(paste("b" ,lambda,"= 0.9")),adj=0)
  else if(i==3) mtext(expression(paste("c" ,lambda,"= 0.8")),adj=0)
  else if(i==4) mtext(expression(paste("d" ,lambda,"= 0.7")),adj=0)
  else if(i==5) mtext(expression(paste("e" ,lambda,"= 0.6")),adj=0)
  else if(i==6) mtext(expression(paste("f" ,lambda,"= 0.5")),adj=0)
  else if(i==7) mtext(expression(paste("g" ,lambda,"= 2")),adj=0)
  else if(i==8) mtext(expression(paste("h" ,lambda,"= 3")),adj=0)
  else if(i==9) mtext(expression(paste("i" ,lambda,"= 4")),adj=0)
  else if(i==10) mtext(expression(paste("j" ,lambda,"= 5")),adj=0)
}

par(mfrow=c(5,2))
plot(fits.phrynosomatidae[[1]],ftype="off",lwd=0.5,
      mar=c(1.1,1.1,2.1,1.1))
mtext(expression(paste("a" ,lambda," = 1")),
        line=0,adj=0)
plot(fits.phrynosomatidae[[2]],ftype="off",lwd=0.5,
      mar=c(1.1,1.1,2.1,1.1))
mtext(expression(paste("b" ,lambda," = 0.9")),
        line=0,adj=0)
plot(fits.phrynosomatidae[[3]],ftype="off",lwd=0.5,
      mar=c(1.1,1.1,2.1,1.1))
mtext(expression(paste("c" ,lambda," = 0.8")),
        line=0,adj=0)
plot(fits.phrynosomatidae[[4]],ftype="off",lwd=0.5,
      mar=c(1.1,1.1,2.1,1.1))
mtext(expression(paste("d" ,lambda," = 0.7")),
        line=0,adj=0)
plot(fits.phrynosomatidae[[5]],ftype="off",lwd=0.5,
      mar=c(1.1,1.1,2.1,1.1))
mtext(expression(paste("e" ,lambda," = 0.6")),
        line=0,adj=0)
plot(fits.phrynosomatidae[[6]],ftype="off",lwd=0.5,
      mar=c(1.1,1.1,2.1,1.1))
mtext(expression(paste("f" ,lambda," = 0.5")),
        line=0,adj=0)
plot(fits.phrynosomatidae[[7]],ftype="off",lwd=0.5,
      mar=c(1.1,1.1,2.1,1.1))
mtext(expression(paste("g" ,lambda," = 2")),
        line=0,adj=0)
plot(fits.phrynosomatidae[[8]],ftype="off",lwd=0.5,
      mar=c(1.1,1.1,2.1,1.1))
mtext(expression(paste("h" ,lambda," = 3")),
        line=0,adj=0)
plot(fits.phrynosomatidae[[9]],ftype="off",lwd=0.5,
      mar=c(1.1,1.1,2.1,1.1))
mtext(expression(paste("i" ,lambda," = 4")),
        line=0,adj=0)
plot(fits.phrynosomatidae[[10]],ftype="off",lwd=0.5,
      mar=c(1.1,1.1,2.1,1.1))
mtext(expression(paste("j" ,lambda," = 5")),
        line=0,adj=0)

#Lacertidae
fits.lacertidae<-list()
for(i in 1:length(lambda))
  fits.lacertidae[[i]]<-
multirateBM(lacertid_tree,dorsaltrait.lacertid,lambda=lambda[i],
              n.iter=3, parallel=TRUE)

```

```

par(mfrow=c(5,2))
sig2.lacertidae <- fit.ml.dorsum.lacertidae$sig2
for(i in 1:length(fits.lacertidae)){
  xyylim<-range(c(sig2.lacertidae, fits.lacertidae[[i]]$sig2))
  plot(sig2.lacertidae, fits.lacertidae[[i]]$sig2, log="xy", xlim=xyylim,
       ylim=xyylim, bty="n", las=1, pch=21, cex=1.2,
       bg="grey", xlab=expression(paste("baseline ", sigma^2)),
       ylab=expression(paste("estimated ", sigma^2)),
       cex.axis=0.8)
  lines(xyylim, xyylim)
  if(i==1) mtext(expression(paste("a ", lambda, "= 1")), adj=0)
  else if(i==2) mtext(expression(paste("b ", lambda, "= 0.9")), adj=0)
  else if(i==3) mtext(expression(paste("c ", lambda, "= 0.8")), adj=0)
  else if(i==4) mtext(expression(paste("d ", lambda, "= 0.7")), adj=0)
  else if(i==5) mtext(expression(paste("e ", lambda, "= 0.6")), adj=0)
  else if(i==6) mtext(expression(paste("f ", lambda, "= 0.5")), adj=0)
  else if(i==7) mtext(expression(paste("g ", lambda, "= 2")), adj=0)
  else if(i==8) mtext(expression(paste("h ", lambda, "= 3")), adj=0)
  else if(i==9) mtext(expression(paste("i ", lambda, "= 4")), adj=0)
  else if(i==10) mtext(expression(paste("j ", lambda, "= 5")), adj=0)
}

```

```

par(mfrow=c(5,2))
plot(fits.lacertidae[[1]], ftype="off", lwd=0.5,
     mar=c(1.1, 1.1, 2.1, 1.1))
mtext(expression(paste("a ", lambda, "= 1")),
       line=0, adj=0)
plot(fits.lacertidae[[2]], ftype="off", lwd=0.5,
     mar=c(1.1, 1.1, 2.1, 1.1))
mtext(expression(paste("b ", lambda, "= 0.9")),
       line=0, adj=0)
plot(fits.lacertidae[[3]], ftype="off", lwd=0.5,
     mar=c(1.1, 1.1, 2.1, 1.1))
mtext(expression(paste("c ", lambda, "= 0.8")),
       line=0, adj=0)
plot(fits.lacertidae[[4]], ftype="off", lwd=0.5,
     mar=c(1.1, 1.1, 2.1, 1.1))
mtext(expression(paste("d ", lambda, "= 0.7")),
       line=0, adj=0)
plot(fits.lacertidae[[5]], ftype="off", lwd=0.5,
     mar=c(1.1, 1.1, 2.1, 1.1))
mtext(expression(paste("e ", lambda, "= 0.6")),
       line=0, adj=0)
plot(fits.lacertidae[[6]], ftype="off", lwd=0.5,
     mar=c(1.1, 1.1, 2.1, 1.1))
mtext(expression(paste("f ", lambda, "= 0.5")),
       line=0, adj=0)
plot(fits.lacertidae[[7]], ftype="off", lwd=0.5,
     mar=c(1.1, 1.1, 2.1, 1.1))
mtext(expression(paste("g ", lambda, "= 2")),
       line=0, adj=0)
plot(fits.lacertidae[[8]], ftype="off", lwd=0.5,
     mar=c(1.1, 1.1, 2.1, 1.1))
mtext(expression(paste("h ", lambda, "= 3")),
       line=0, adj=0)
plot(fits.lacertidae[[9]], ftype="off", lwd=0.5,
     mar=c(1.1, 1.1, 2.1, 1.1))
mtext(expression(paste("i ", lambda, "= 4")),
       line=0, adj=0)
plot(fits.lacertidae[[10]], ftype="off", lwd=0.5,
     mar=c(1.1, 1.1, 2.1, 1.1))
mtext(expression(paste("j ", lambda, "= 5")),
       line=0, adj=0)

```

#Cordylidae

```

fits.cordylidae<-list()
for(i in 1:length(lambda))
  fits.cordylidae[[i]]<-
  multirateBM(cordylid_tree, dorsaltrait.cordylid, lambda=lambda[i],
              n.iter=3, parallel=TRUE)

```

```

par(mfrow=c(5,2))
sig2.cordylidae <- fit.ml.dorsum.cordylidae$sig2
for(i in 1:length(fits.cordylidae)){
  xyylim<-range(c(sig2.cordylidae, fits.cordylidae[[i]]$sig2))
  plot(sig2.cordylidae, fits.cordylidae[[i]]$sig2, log="xy", xlim=xyylim,
       ylim=xyylim, bty="n", las=1, pch=21, cex=1.2,
       bg="grey", xlab=expression(paste("baseline ", sigma^2)),
       ylab=expression(paste("estimated ", sigma^2)),
       cex.axis=0.8)

```

```

lines(xylim,xyylim)
if(i==1) mtext(expression(paste("a" ,lambda,"= 1")),adj=0)
else if(i==2) mtext(expression(paste("b" ,lambda,"= 0.9")),adj=0)
else if(i==3) mtext(expression(paste("c" ,lambda,"= 0.8")),adj=0)
else if(i==4) mtext(expression(paste("d" ,lambda,"= 0.7")),adj=0)
else if(i==5) mtext(expression(paste("e" ,lambda,"= 0.6")),adj=0)
else if(i==6) mtext(expression(paste("f" ,lambda,"= 0.5")),adj=0)
else if(i==7) mtext(expression(paste("g" ,lambda,"= 2")),adj=0)
else if(i==8) mtext(expression(paste("h" ,lambda,"= 3")),adj=0)
else if(i==9) mtext(expression(paste("i" ,lambda,"= 4")),adj=0)
else if(i==10) mtext(expression(paste("j" ,lambda,"= 5")),adj=0)
}

par(mfrow=c(5,2))
plot(fits.cordylidae[[1]],ftype="off",lwd=0.5,
     mar=c(1.1,1.1,2.1,1.1))
mtext(expression(paste("a" ,lambda,"= 1")),
       line=0,adj=0)
plot(fits.cordylidae[[2]],ftype="off",lwd=0.5,
     mar=c(1.1,1.1,2.1,1.1))
mtext(expression(paste("b" ,lambda,"= 0.9")),
       line=0,adj=0)
plot(fits.cordylidae[[3]],ftype="off",lwd=0.5,
     mar=c(1.1,1.1,2.1,1.1))
mtext(expression(paste("c" ,lambda,"= 0.8")),
       line=0,adj=0)
plot(fits.cordylidae[[4]],ftype="off",lwd=0.5,
     mar=c(1.1,1.1,2.1,1.1))
mtext(expression(paste("d" ,lambda,"= 0.7")),
       line=0,adj=0)
plot(fits.cordylidae[[5]],ftype="off",lwd=0.5,
     mar=c(1.1,1.1,2.1,1.1))
mtext(expression(paste("e" ,lambda,"= 0.6")),
       line=0,adj=0)
plot(fits.cordylidae[[6]],ftype="off",lwd=0.5,
     mar=c(1.1,1.1,2.1,1.1))
mtext(expression(paste("f" ,lambda,"= 0.5")),
       line=0,adj=0)
plot(fits.cordylidae[[7]],ftype="off",lwd=0.5,
     mar=c(1.1,1.1,2.1,1.1))
mtext(expression(paste("g" ,lambda,"= 2")),
       line=0,adj=0)
plot(fits.cordylidae[[8]],ftype="off",lwd=0.5,
     mar=c(1.1,1.1,2.1,1.1))
mtext(expression(paste("h" ,lambda,"= 3")),
       line=0,adj=0)
plot(fits.cordylidae[[9]],ftype="off",lwd=0.5,
     mar=c(1.1,1.1,2.1,1.1))
mtext(expression(paste("i" ,lambda,"= 4")),
       line=0,adj=0)
plot(fits.cordylidae[[10]],ftype="off",lwd=0.5,
     mar=c(1.1,1.1,2.1,1.1))
mtext(expression(paste("j" ,lambda,"= 5")),
       line=0,adj=0)

end.time.lambda <- Sys.time()
time.taken.lambda <- end.time.lambda - start.time.lambda
time.taken.lambda # 10.06139 hours

#-----
#-----
#-----

#And now we are going to evaluate the correlation between brightness
evolution rates and delta-O-18

evo_rates_1 <- NULL
for(i in 1:nrow(evo_rates)){
  res.value.paleo <- NULL
  value <- abs(evo_rates[i,]$time_adj)
  for(j in 1:nrow(paleo_climate)){
    value.paleo <- abs(paleo_climate[j,]$age-value)
    res.value.paleo <- c(res.value.paleo,value.paleo)
  }
  diff <-
which(as.numeric(res.value.paleo)==min(as.numeric(res.value.paleo)))
  res <-
cbind(evo_rates[i,],paleo_climate[diff,]$age,paleo_climate[diff,]$d18O)
  evo_rates_1 <- rbind(evo_rates_1,res)
}
head(evo_rates_1)

```

```

names(evo_rates_1) <- c("SIG2","time","time_adj","clade","age","d180")

combined_evo_rates <- evo_rates_1 %>%
  group_by(time_adj) %>%
  summarise(SIG2=mean(SIG2),
            d180=mean(d180),
            clade=clade)

evo_plot_Varaniidae <-
ggscatter(subset(combined_evo_rates,clade=="Varaniidae"), x = "d180", y =
"SIG2",
          add = "none", conf.int = TRUE,
          cor.coef = TRUE, cor.method = "pearson",
          xlab = expression(paste(" ",delta^18,"0")),
          ylab = expression(paste("Brightness evolution
rate ( ",sigma^2,")")),
          title = "Varaniidae")

evo_plot_Viperidae <-
ggscatter(subset(combined_evo_rates,clade=="Viperidae"), x = "d180", y =
"SIG2",
          add = "none", conf.int = TRUE,
          cor.coef = TRUE, cor.method = "spearman",
          xlab = expression(paste("
",delta^18,"0")),
          ylab = expression(paste("Brightness
evolution rate ( ",sigma^2,")")),
          title = "Viperidae")

evo_plot_Cordylidae <-
ggscatter(subset(combined_evo_rates,clade=="Cordylidae"), x = "d180", y =
"SIG2",
          add = "none", conf.int = TRUE,
          cor.coef = TRUE, cor.method = "pearson",
          xlab = expression(paste("
",delta^18,"0")), ylab = expression(paste("Brightness evolution rate (
",sigma^2,")")),title = "Cordylidae")
evo_plot_Diplodactylidae <-
ggscatter(subset(combined_evo_rates,clade=="Diplodactylidae"), x = "d180",
y = "SIG2",
          add = "none", conf.int = TRUE,
          cor.coef = TRUE, cor.method = "pearson",
          xlab = expression(paste("
",delta^18,"0")), ylab = expression(paste("Brightness evolution rate (
",sigma^2,")")),title = "Diplodactylidae")
evo_plot_Lacertidae <-
ggscatter(subset(combined_evo_rates,clade=="Lacertidae"), x = "d180", y =
"SIG2",
          add = "none", conf.int = TRUE,
          cor.coef = TRUE, cor.method = "spearman",
          xlab = expression(paste("
",delta^18,"0")), ylab = expression(paste("Brightness evolution rate (
",sigma^2,")")),title = "Lacertidae")
evo_plot_Phrynosomatidae <-
ggscatter(subset(combined_evo_rates,clade=="Phrynosomatidae"), x = "d180",
y = "SIG2",
          add = "none", conf.int = TRUE,
          cor.coef = TRUE, cor.method = "spearman",
          xlab = expression(paste("
",delta^18,"0")), ylab = expression(paste("Brightness evolution rate (
",sigma^2,")")),title = "Phrynosomatidae")
evo_plot_Tropiduridae <-
ggscatter(subset(combined_evo_rates,clade=="Tropiduridae"), x = "d180", y =
"SIG2",
          add = "none", conf.int = TRUE,
          cor.coef = TRUE, cor.method = "pearson",
          xlab = expression(paste("
",delta^18,"0")), ylab = expression(paste("Brightness evolution rate (
",sigma^2,")")),title = "Tropiduridae")

ggarrange(evo_plot_Cordylidae,
          evo_plot_Diplodactylidae,
          evo_plot_Lacertidae,
          evo_plot_Phrynosomatidae,
          evo_plot_Tropiduridae,
          evo_plot_Varaniidae,
          evo_plot_Viperidae,
          labels = c("a", "b", "c", "d", "e", "f", "g"),
          align = "v")

```

```

#-----
-----

```

```

#-----
#-----

#Let's now test whether climate change through time has shaped the rate of
evolution of color brightness using multi-regime models

#Cordylidae
data.test_cordylid <-
data.frame(Genus_species=rownames(data_squamata_Cordylidae),Veg=data_squamata_Cordylidae$Habitat
,
           X=data_squamata_Cordylidae$Dorsal)

rownames(data.test_cordylid) <- data.test_cordylid$Genus_species

vegtrait.cordylid <- setNames(data.test_cordylid$Veg,
data.test_cordylid$Genus_species)

## perform & plot stochastic maps
smap.tree.cordylid<-make.simap(cordylid_tree,vegtrait.cordylid)
plot(smap.tree.cordylid,type="fan",fsize=0.8,ftype="i")

fitBM_cordylid<-
OUwie(smap.tree.cordylid,data.test_cordylid,model="BM1",simmap.tree=TRUE)
## single rate
fitBMS_cordylid<-
OUwie(smap.tree.cordylid,data.test_cordylid,model="BMS",simmap.tree=TRUE)
## multiple rates
fitOUM_cordylid<-
OUwie(smap.tree.cordylid,data.test_cordylid,model="OUM",simmap.tree=TRUE)
## multiple optima

aic_cordylid<-
setNames(c(fitBM_cordylid$AIC,fitBMS_cordylid$AIC,fitOUM_cordylid$AIC),
c("BM1","BMS","OUM"))
aic_cordylid
aic.w(aic_cordylid)

# -----
# -----

#Diplodactylidae
data.test_diplo <-
data.frame(Genus_species=rownames(data_squamata_Diplodactylidae),Veg=data_squamata_Diplodactylidae$Habitat
,
           X=data_squamata_Diplodactylidae$Dorsal)

rownames(data.test_diplo) <- data.test_diplo$Genus_species

vegtrait.diplo <- setNames(data.test_diplo$Veg,
data.test_diplo$Genus_species)

## perform & plot stochastic maps
smap.tree.diplo<-make.simap(diplo_tree,vegtrait.diplo)
plot(smap.tree.diplo,type="fan",fsize=0.8,ftype="i")

fitBM_diplo<-
OUwie(smap.tree.diplo,data.test_diplo,model="BM1",simmap.tree=TRUE) ##
single rate
fitBMS_diplo<-
OUwie(smap.tree.diplo,data.test_diplo,model="BMS",simmap.tree=TRUE) ##
multiple rates
fitOUM_diplo<-
OUwie(smap.tree.diplo,data.test_diplo,model="OUM",simmap.tree=TRUE) ##
multiple optima

aic_diplo<-setNames(c(fitBM_diplo$AIC,fitBMS_diplo$AIC,fitOUM_diplo$AIC),
c("BM1","BMS","OUM"))
aic_diplo
aic.w(aic_diplo)

# -----
# -----

#Lacertidae
data.test_lacertid <-
data.frame(Genus_species=rownames(data_squamata_Lacertidae),Veg=data_squamata_Lacertidae$Habitat
,
           X=data_squamata_Lacertidae$Dorsal)

```

```

rownames(data.test_lacertid) <- data.test_lacertid$Genus_species

vegtrait.lacertid <- setNames(data.test_lacertid$Veg,
data.test_lacertid$Genus_species)

## perform & plot stochastic maps
smap.tree.lacertid<-make.simap(lacertid_tree,vegtrait.lacertid)
plot(smap.tree.lacertid,type="fan",fsize=0.8,ftype="i")

fitBM_lacertid<-
OUwie(smap.tree.lacertid,data.test_lacertid,model="BM1",simmap.tree=TRUE)
## single rate
fitBMS_lacertid<-
OUwie(smap.tree.lacertid,data.test_lacertid,model="BMS",simmap.tree=TRUE)
## multiple rates
fitOUM_lacertid<-
OUwie(smap.tree.lacertid,data.test_lacertid,model="OUM",simmap.tree=TRUE)
## multiple optima

aic_lacertid<-
setNames(c(fitBM_lacertid$AIC,fitBMS_lacertid$AIC,fitOUM_lacertid$AIC),
c("BM1","BMS","OUM"))
aic_lacertid
aic.w(aic_lacertid)

# -----
# -----

#Phrynosomatidae
data.test_phryno <-
data.frame(Genus_species=rownames(data_squamata_Phrynosomatidae),Veg=data_squamata_Phrynosomatidae$Habitat
,
X=data_squamata_Phrynosomatidae$Dorsal)

rownames(data.test_phryno) <- data.test_phryno$Genus_species

vegtrait.phryno <- setNames(data.test_phryno$Veg,
data.test_phryno$Genus_species)

## perform & plot stochastic maps
smap.tree.phryno<-make.simap(phryno_tree,vegtrait.phryno)
plot(smap.tree.phryno,type="fan",fsize=0.8,ftype="i")

fitBM_phryno<-
OUwie(smap.tree.phryno,data.test_phryno,model="BM1",simmap.tree=TRUE) ##
single rate
fitBMS_phryno<-
OUwie(smap.tree.phryno,data.test_phryno,model="BMS",simmap.tree=TRUE) ##
multiple rates
fitOUM_phryno<-
OUwie(smap.tree.phryno,data.test_phryno,model="OUM",simmap.tree=TRUE) ##
multiple optima

aic_phryno<-
setNames(c(fitBM_phryno$AIC,fitBMS_phryno$AIC,fitOUM_phryno$AIC),
c("BM1","BMS","OUM"))
aic_phryno
aic.w(aic_phryno)

# -----
# -----

#Tropiduridae
data.test_tropi <-
data.frame(Genus_species=rownames(data_squamata_Tropiduridae),Veg=data_squamata_Tropiduridae$Habitat
,
X=data_squamata_Tropiduridae$Dorsal)

rownames(data.test_tropi) <- data.test_tropi$Genus_species

vegtrait.tropi <- setNames(data.test_tropi$Veg,
data.test_tropi$Genus_species)

## perform & plot stochastic maps
smap.tree.tropi<-make.simap(tropi_tree,vegtrait.tropi)
plot(smap.tree.tropi,type="fan",fsize=0.8,ftype="i")

```

```

fitBM_tropi<-
OUwie(smmap.tree.tropi,data.test_tropi,model="BM1",simmap.tree=TRUE) ##
single rate
fitBMS_tropi<-
OUwie(smmap.tree.tropi,data.test_tropi,model="BMS",simmap.tree=TRUE) ##
multiple rates
fitOUM_tropi<-
OUwie(smmap.tree.tropi,data.test_tropi,model="OUM",simmap.tree=TRUE) ##
multiple optima

aic_tropi<-setNames(c(fitBM_tropi$AIC,fitBMS_tropi$AIC,fitOUM_tropi$AIC),
                    c("BM1","BMS","OUM"))
aic_tropi

aic.w(aic_tropi)

# -----
#-----

#Varanidae
data.test_varanidae <-
data.frame(Genus_species=rownames(data_squamata_Varandae),Veg=data_squamata_Varandae$Habitat
,
           X=data_squamata_Varandae$Dorsal)

rownames(data.test_varanidae) <- data.test_varanidae$Genus_species

vegtrait.varandae <- setNames(data.test_varandae$Veg,
data.test_varandae$Genus_species)

## perform & plot stochastic maps
smmap.tree.varandae<-make.simmap(varandae_tree,vegtrait.varandae)
plot(smmap.tree.varandae,type="fan",fsize=0.8,ftype="i")

fitBM_varandae<-
OUwie(smmap.tree.varandae,data.test_varandae,model="BM1",simmap.tree=TRUE)
## single rate
fitBMS_varandae<-
OUwie(smmap.tree.varandae,data.test_varandae,model="BMS",simmap.tree=TRUE)
## multiple rates
fitOUM_varandae<-
OUwie(smmap.tree.varandae,data.test_varandae,model="OUM",simmap.tree=TRUE)
## multiple optima

aic_varandae<-
setNames(c(fitBM_varandae$AIC,fitBMS_varandae$AIC,fitOUM_varandae$AIC),
         c("BM1","BMS","OUM"))
aic_varandae

aic.w(aic_varandae)

# -----
#-----

#Viperidae
data.test_viper <-
data.frame(Genus_species=rownames(data_squamata_Viperidae),Veg=data_squamata_Viperidae$Habitat
,
           X=data_squamata_Viperidae$Dorsal)

rownames(data.test_viper) <- data.test_viper$Genus_species

vegtrait.viper <- setNames(data.test_viper$Veg,
data.test_viper$Genus_species)

## perform & plot stochastic maps
smmap.tree.viper<-make.simmap(viper_tree,vegtrait.viper)
plot(smmap.tree.viper,type="fan",fsize=0.8,ftype="i")

fitBM_viper<-
OUwie(smmap.tree.viper,data.test_viper,model="BM1",simmap.tree=TRUE) ##
single rate
fitBMS_viper<-
OUwie(smmap.tree.viper,data.test_viper,model="BMS",simmap.tree=TRUE) ##
multiple rates
fitOUM_viper<-
OUwie(smmap.tree.viper,data.test_viper,model="OUM",simmap.tree=TRUE) ##
multiple optima

```

```

aic_viper<-setNames(c(fitBM_viper$AIC,fitBMS_viper$AIC,fitOUM_viper$AIC),
                    c("BM1","BMS","OUM"))
aic_viper

aic.w(aic_viper)

# As OU models perform better we continue with these
fitOUM_list <-
list(fitOUM_cordylid,fitOUM_diplo,fitOUM_lacertid,fitOUM_phryno,fitOUM_tropi,fitOUM_varanidae,fitOUM_viper
)
names_vector <- c("Cordylidae","Diplodactylidae","Lacertidae",
                  "Phrynosomatidae","Tropiduridae","Varanidae","Viperidae")
total.res <- NULL
for(i in 1:length(fitOUM_list)){
  res <- as.data.frame(fitOUM_list[[i]]$theta)
  res <- cbind(res,clade=paste(names_vector[i]))

  total.res <- rbind(total.res,res)
}
total.res <-
cbind(total.res,Habitat=c("Open","OpenClosed",rep(c("Closed","Open","OpenClosed"),6))
)

colors_evorates <- c("Closed" = "darkblue",
                    "OpenClosed" = "chartreuse4",
                    "Open" = "darkgoldenrod2")

total.res$Habitat <- factor(total.res$Habitat,levels = c("Open",
"OpenClosed", "Closed"))
total.res$SIG2 <- total.res$V1

ggplot(data=total.res,aes(x=Habitat,y=SIG2,colour= Habitat, fill=Habitat))+
  geom_bar(stat="identity")+
  geom_errorbar(aes(ymin=SIG2-se,ymax=SIG2+se),width=.2)+
  theme_bw()+theme(panel.grid = element_blank())+
  scale_color_manual(values=colors_evorates) +
  scale_fill_manual(values=colors_evorates) +
  theme(legend.position = "none") +
  facet_wrap(~clade,scales = "free") +
  scale_x_discrete(name="Vegetation structure") +
  scale_y_continuous(name = expression(paste("Brightness evolution rate (
",sigma^2,")"))) +
  coord_cartesian(ylim=c(32,51))

#-----
#-----
#-----

#For fun, let's plot our tree with the geoscalePhylo function in "strap"
squamata_tree_red$root.time <- max(nodeHeights(squamata_tree_red))
geoscalePhylo(squamata_tree_red,cex.age = 0.3, cex.ts = 0.3, cex.tip =
0.3,width = 0.0001, units = c("Period", "Epoch", "Age"),)
...

```{COMMENT 3}
#Cordylidae
data.test_cordylid <-
data.frame(Genus_species=rownames(data_squamata_Cordylidae),Veg=data_squamata_Cordylidae$Habitat
,
 X=data_squamata_Cordylidae$Dorsal)

rownames(data.test_cordylid) <- data.test_cordylid$Genus_species

vegtrait.cordylid <- setNames(data.test_cordylid$Veg,
data.test_cordylid$Genus_species)

perform & plot stochastic maps
smap.tree.cordylid<-make.simmap(cordylid_tree,vegtrait.cordylid)
plot(smap.tree.cordylid,type="fan",fsize=0.8,ftype="i")

fitBM_cordylid<-
OUwie(smap.tree.cordylid,data.test_cordylid,model="BM1",simmap.tree=TRUE)
single rate
fitBMS_cordylid<-
OUwie(smap.tree.cordylid,data.test_cordylid,model="BMS",simmap.tree=TRUE)
multiple rates
fitOU1_cordylid<-

```

```
OUwie(smmap.tree.cordylid,data.test_cordylid,model="OU1",simmap.tree=TRUE)
multiple optima
fitOUM_cordylid<-
OUwie(smmap.tree.cordylid,data.test_cordylid,model="OUM",simmap.tree=TRUE)
multiple optima

aic_cordylid<-
setNames(c(fitBM_cordylid$AIC,fitBMS_cordylid$AIC,fitOU1_cordylid$AIC,fitOUM_cordylid$AIC)
,
 c("BM1","BMS","OU1", "OUM"))
aic_cordylid

aic.w(aic_cordylid)
``,``
```